# Mixotrophy emerges as an optimal strategy in mature waters of the Amazon River plume
Ana Fernández-Carrera [1,4] ✉, Noémie Choisnard[1], Dirk Wodarg[1], Iris Liskow[1], Ajit Subramaniam [2], Joseph P. Montoya [3], Maren Voss [1] & Natalie Loick-Wilde [1] ✉

Phytoplankton, namely diatoms and cyanobacteria, combine photoautotrophy and the uptake of dissolved organic matter (osmotrophy) for a mixotrophic living. All other photosynthetic protists, except diatoms, are potentially phagotrophs, and currently classified as mixoplankton. This functional group occupies a unique position between autotrophs and heterotrophs in planktonic food webs, producing a greater carbon stock and higher-quality food for metazoans than phytoplankton do. However, field studies remain challenging due to the difficulty of distinguishing their sole activity in seston containing a mixture of all functional groups. During April/May 2018 and 2021, we examined seston using compound-specific stable nitrogen isotope analysis of amino acids to determine its trophic dynamics along the Amazon River plume. Based on the comparison of nitrogen isotopes in glutamic acid and alanine with phenylalanine, we found a dominance of mixotrophs in the Outer Plume Margin, a region of mature waters around 27 days old. Mixotrophy appears to be the optimal growth strategy in these heterogeneous margins as part of the succession of microalgae functional diversity along the plume. Our study highlights the urgent need to study mixotrophs and mixoplankton in situ within a multidisciplinary framework, pioneering the use of amino acid nitrogen isotopes in field research in this area.

The traditional dichotomy between plant-like (i.e., phytoplankton) and animal-like (i.e., zooplankton) organisms structuring the base of pelagic food webs has been challenged by increasing evidence that most 'phytoplankton' are in fact mixotrophs, that is, they combine photoautotrophy and heterotrophy to make a living[1]. On the one hand, eukaryotic and prokaryotic photoautotrophs are able to use dissolved organic matter to some extent to support their nutritional needs[2–5], i.e., they are osmotrophs. However, if osmotrophy is used for acquiring organic compounds or recovering leaked metabolites is still unclear[6]. On the other hand, most unicellular eukaryotes, with the exception of diatoms, also exploit phagotrophy on varying degrees, thus constituting an additional functional group recently recognized and named mixoplankton[7]. Hence, four functional groups are currently defined in unicellular planktonic organisms: bacterioplankton, phytoplankton, mixoplankton, and protozooplankton. The impact of mixotrophs that rely on a combination of osmotrophy and photoautotrophy on biogeochemical cycling differs markedly from that of mixoplankton that use phagotrophy as

well. Osmotrophy does not remove preys, competitors or other grazers from the food web affecting its structure, whereas phagotrophy does[5]. Mixoplankton do not work in binary fashion, switching between pure photoautotrophy and pure heterotrophy. In most cases, they couple both nutritional modes to maximize their growth efficiency[8]. Heterotrophy, via osmotrophy and/or phagotrophy, can support growth directly or supply limiting nutrients to photosynthesis, which in turn compensates respiratory losses due to the former, allowing the increase in biomass of the mixotrophic stock relative to the autotrophic[9,10]. In addition, mixoplankton provide better quality food to metazoans[11], and, in seasonally matured systems (i.e., nutrient poor) where they dominate the community, they drive the accumulation of refractory dissolved organic matter, which directly contributes to the ability of the carbon pump to mitigate atmospheric carbon dioxide[9]. Overall, their better performance seems to lead to a higher transfer of mixotrophic carbon and energy than that of pure autotrophs[8]. Yet, laboratory experiments also suggest that photoheterotrophy is the

[1]Department of Biological Oceanography, Leibniz Institute for Baltic Sea Research Warnemuende, Rostock, Germany. [2]Lamont-Doherty Earth Observatory, Columbia University, Palisades, NY, USA. [3]School of Biological Sciences, Georgia Institute of Technology, Atlanta, GA, USA. [4]Present address: Institute of Oceanography and Global Change. Universidad de Las Palmas de Gran Canaria, Taliarte, Spain. ✉e-mail: ana.carrera@ulpgc.es; natalie.loick-wilde@io-warnemuende.de

dominant mode in some mixoplankton, resulting in no enhancement of primary production or compensation of respiratory losses, as the photosynthetic system is used as mere energy provider rather than carbon fixer[12]. Furthermore, some groups seem to provide poor quality food affecting the overall performance of their grazers[13]. Thus, it remains uncertain whether mixoplankton universally implies an increase or a decrease in mass (e.g., carbon and nitrogen) and energy transfer to upper trophic levels in comparison to phytoplankton. Therefore, it is imperative to investigate their occurrence and activity in natural systems. The development of gene-based predictive models for identifying the potential activity of mixoplankton in the field is currently being explored[14,15], as there is no specific molecular marker for directly quantifying heterotrophy. In this regard, the amino acid nitrogen stable isotope analysis (or CSIA-AA) offers a complementary tool for identifying the imprint of osmotrophy and phagotrophy from microalgae-derived field samples by combining the differential fractionation in nitrogen (N) stable isotopes of the trophic amino acids glutamic acid or alanine with the source amino acid phenylalanine[16–19].

The seminal work of McClelland and Montoya[20] demonstrated the potential of nitrogen isotopes in amino acids to track food web dynamics. This approach eliminates many of the problems associated with the traditional bulk nitrogen isotope approaches concerning the origin, movement, and transformation of nitrogen in the upper water column as well as the potential decoupling of bulk $\delta^{15}N$ signatures due to the different N turnover times (life spans) of autotrophs and consumers[21,22]. They showed that trophic and source amino acids exhibit distinct nitrogen isotopic behaviors: trophic amino acids, such as glutamic acid (Glu) or alanine (Ala), become enriched in $\delta^{15}N$ due to metabolic processing with each trophic transfer, while source amino acids, such as phenylalanine (Phe), retain their isotopic signatures from primary producers[19,23]. Building on this foundation, subsequent studies, notably by Chikaraishi and coworkers, formalized the first equations that allow precise trophic position (TP) calculations[24], making CSIA-AA a widely accepted approach in marine ecology and across biological kingdoms[25]. The graphical analysis of trophic position is done by comparing the $\delta^{15}N$ values of a trophic amino acid (e.g., Glu or Ala) with those of a source amino acid (e.g., Phe). In this trophic space, it is possible to define trophoclines (Fig. 1), that is, lines representing the same trophic position which result from the combination of $\delta^{15}N$ values of the two amino acids along the aforementioned equations[24]. Trophoclines can be labeled as TP_{Glu} and TP_{Ala} for representing the trophic amino acid, glutamic acid +

glutamine and alanine, respectively. The potential of CSIA-AA to address complexity in natural systems is evident in its ability to track both nitrogen assimilation pathways and metabolic processes associated with mixotrophy. Yamaguchi et al.[16] compared the nitrogen isotope signatures of the amino acids of different groups of microbes (*Archaea*, fungi, and bacteria) growing on ammonium or casamino acids. Those growing solely on ammonium as dissolved inorganic nitrogen (DIN) source were capable of de novo synthesis of their own amino acids, hence presented a difference in the nitrogen isotopes of Glu and Ala relative to Phe similar to that of autotrophs (Fig. 1A, B), and overlapped with the literature autotrophic microalgae end member around TP_{Glu} and TP_{Ala} 1.0. By contrast, the authors showed that the same microbes growing on dissolved organic nitrogen (DON) from a casamino acids medium (i.e., as osmotrophs) presented a nitrogen isotope enrichment of Glu similar to that of herbivores, clearly distributing around the TP_{Glu} 2.0 (Fig. 1A). That is, their TP based on amino acid nitrogen isotopes was consistent with herbivores. However, the impact on nitrogen isotope enrichment in Ala was not as clear as that of Glu with data distributing between the trophocline of autotrophs (TP_{Ala} 1.0, Fig. 1B) to that of herbivores (TP_{Ala} 2.0, Fig. 1B). The authors suggested that nitrogen isotopes in Glu integrated the heterotrophic processes related to osmotrophy better than nitrogen isotopes in other amino acids like Ala. By contrast, the unique role of Ala as integrator of protistan phagotrophic steps in the food web, invisible in nitrogen isotopes in Glu, was shown by Gutierrez-Rodriguez et al.[17] and Décima et al.[18] in diverse protozooplankton cultures (*Oxyrrhis marina*, *Heterocapsa triquetra*, and *Favella* spp.). When nitrogen from phagocytosis as feeding strategy is the sole nitrogen source of protists, that is, when they are the herbivorous trophic step in a food web, nitrogen isotopes in Glu do not enrich proportionally to this trophic step, hence TP_{Glu} is located also below 1.5 close to the autotrophic end member (Fig. 1A). By contrast, TP_{Ala} of protozooplankton uniformly distributes around 2.0, clearly reflecting the trophic mode of these grazers (Fig. 1B). Hence, given the distinct effect of osmotrophy and phagotrophy on their nitrogen isotopes, the combination of glutamic acid and alanine nitrogen isotopic signatures has the potential to detect mixoplankton activity in natural samples.

The Amazon River plume provides a unique natural laboratory, where it is possible to study different stages of microalgae functional and community structure succession as the plume matures from nutrient-replete to nutrient-deplete conditions. It spreads in the Western Tropical Atlantic Ocean, where the energetic surface circulation plays a key role in the

**Fig. 1 | Literature end members of nitrogen isotopes in phenylalanine, glutamic acid and alanine.**
**A** $\delta^{15}N_{Phe}$ (phenylalanine) vs $\delta^{15}N_{Glu}$ (glutamic acid + glutamine) and **B** $\delta^{15}N_{Phe}$ (phenylalanine) vs $\delta^{15}N_{Ala}$ (alanine) of the literature end members[16,18,24,65,66]. Symbols represent the organisms: cultured fungi (square), cultured bacteria (diamond), cultured *Archaea* (triangle), cultured protozooplankton (crossed circle), and cultured microalgae (circle). Colors represent the nitrogen source: dissolved inorganic nitrogen (green, DIN), dissolved organic nitrogen (purple, DON), nitrogen from phagocytosis (pink, N from preys). Trophoclines (dashed and dotted gray lines) with a slope of 1.0 and y-intercepts of 3.4‰, 7.2‰, and 11.0‰ for $\delta^{15}N_{Glu}$ and of 3.2‰, 6.0‰, and 8.9‰ for $\delta^{15}N_{Ala}$, respectively, represent different trophic positions (TPs = 1.0, 1.5, 2.0) according to Chikaraishi et al.[24]. Data are provided in Supplementary Data 1 for a total of $n = 46$ individual samples collected from literature.

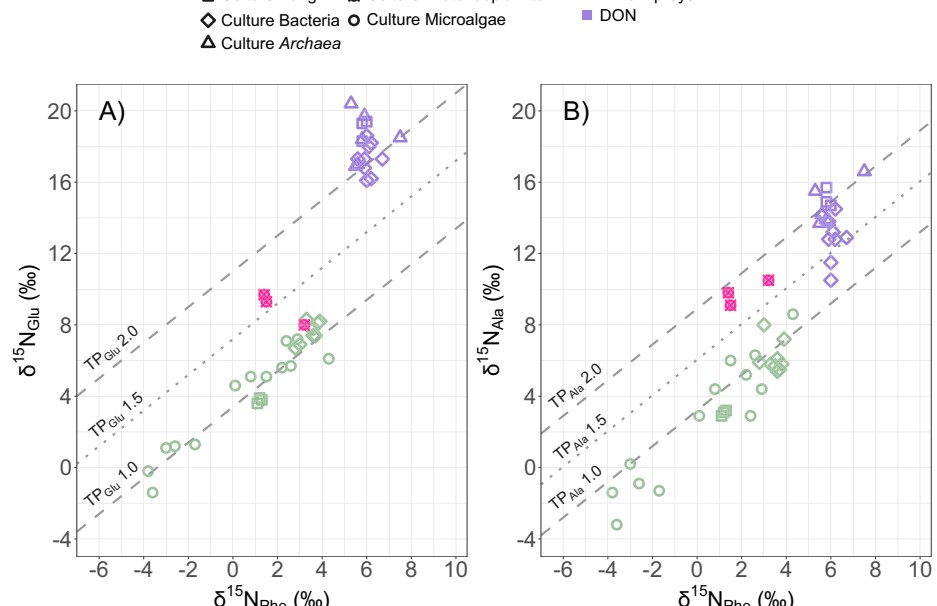

transport of heat, salt, and water from the Southern to the Northern Hemisphere[26]. In this complex region, the Amazon River discharges about 50% of all annual freshwater input into the tropical Atlantic, contributing up to 20% of annual fluvial freshwater into the global ocean[27,28]. The magnitude of the Amazon outflow changes markedly through the year, with a maximum in May/June and a minimum in October/November[29]. The interaction of the river discharge with the strong tidal currents (>1 m s⁻¹), wind stress and the energetic North Brazil current results in complex circulation patterns on the shelf over the year[29]. The Amazon River plume displays a strong seasonal variability, and spreads as a buoyant lens of fresher water at the surface, reaching hundreds of km to the north and the east during the peak outflow season. This freshwater lens acts as an effective barrier to vertical mixing[30], so the plume works as a semi-enclosed system slowly exchanging material with the adjacent environment. As a result of this extension and complexity, at any given instant, the plume is composed of a mixture of waters of different age and salinity[30], potentially reflecting different stages of microalgae succession. The strong dynamics along the plume and its large extension define distinct planktonic habitats capturing the aging process of the plume[31], where the mature systems optimal for mixoplankton can emerge[9].

For the purpose of this study, we will refer to the pigmented community containing both phytoplankton and mixoplankton as 'microalgae'. The organisms will also be named based on their dominant activity identified using our amino acid nitrogen isotope approach. Therefore, when autotrophy is the dominant mode in seston estimated using glutamic acid and phenylalanine nitrogen isotopes (TP$_{Glu}$ 1.0), the organisms will be referred to as 'autotrophs', whereas those with a TP$_{Glu}$ of 1.5 will be referred to as 'mixotrophs', reflecting osmotrophy in addition to autotrophy. The trophic position estimated using alanine and phenylalanine nitrogen isotopes (TP$_{Ala}$) will be used for detecting the activity of phagotrophs, where a TP$_{Ala}$ of 1.5 would reflect an equal contribution of autotrophy and

phagotrophy to the samples. The term mixoplankton will be applied to refer to organisms fulfilling the new paradigm (i.e., coupling osmo-phago-photoautotrophy), while phytoplankton refers only to photosynthetic cyanobacteria and diatoms. All the relevant terms to this manuscript are summarized in the Glossary in Supplementary Table 1 to make the document easier to read. Additionally, it should be noted that our trophic position estimates are based on nitrogen. Therefore, any heterotrophic reworking of glutamic acid or alanine, that leads to a greater enrichment of nitrogen isotopes in these amino acids than expected for autotrophs, reflects a trophic shift in terms of nitrogen, i.e., a change in trophic position.

In this study, we apply a multidisciplinary approach to explore the environmental drivers of the dominant trophic mode at the base of the planktonic food web (autotrophy, osmotrophy and/or phagotrophy) in the varying habitats along the Amazon River plume from the estuary to the northern end of the plume at 15°N in two cruises in 2018 and 2021. Our approach by the use of nitrogen stable isotopes of individual amino acids reveals widespread osmo-phago-photoautotrophy in mature waters of the plume, where the potential accumulation of refractory dissolved organic matter will likely contribute to the sink of carbon dioxide found in the margins[32], and also confirms that this flexibility in microalgae nutrition is more ubiquitous in the global ocean than traditionally considered.

## Results
### Environmental variables
The Amazon River plume reached as far as 15°N during both cruises as shown by the extension of sea surface salinities below 35 derived from the Copernicus Global Ocean Physics Analysis and Forecast product[33] (Fig. 2). Our samples covered six different habitats along the plume[31], from the freshest plume in the Riverine (RI) to the oldest stages in the margins (OPM and WPM) and the Modified Oceanic water (MOW) habitats. Samples were also taken in the habitat without influence of the plume named as Oceanic

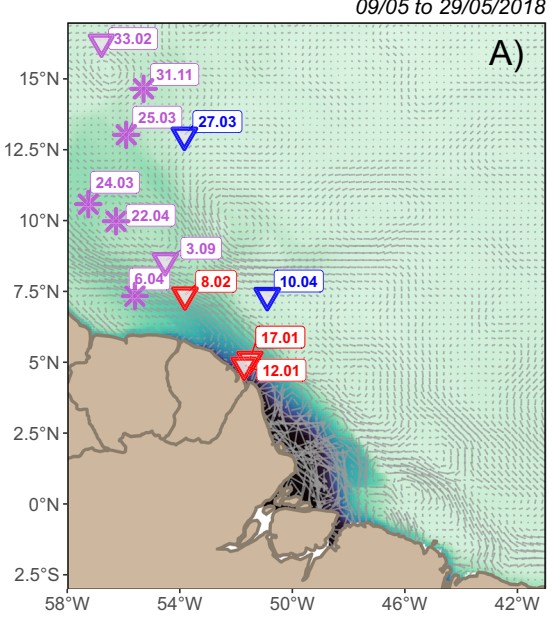
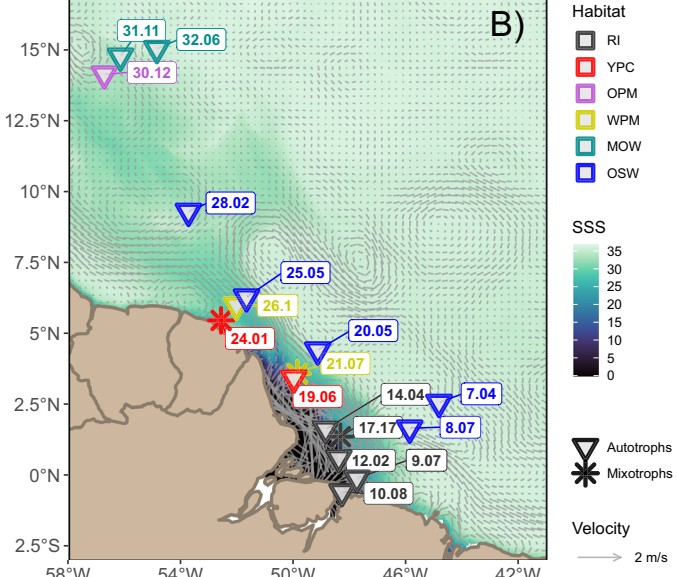

**Fig. 2 | Map of stations sampled at surface along the Amazon River plume and their dominant trophic position.** Stations sampled for compound-specific isotope analysis of amino acids of seston in surface waters along the Amazon River plume over sea surface salinity (SSS) contours and geostrophic velocities for the 2018 **A** and 2021 **B** cruises. Shape of the symbols represent the dominant trophic position (TP$_{Glu}$) in seston based on the pair glutamic acid + glutamine and phenylalanine, with inverted triangle for autotrophs and asterisk for mixotrophs. Both size fractions presented the same TP$_{Glu}$ in all stations, except station 17.17 in 2021, where the large size fraction (>3 μm) was mixotrophic and the small (0.2−3 μm) autotrophic. A total of n = 46 individual samples were collected at surface in 29 stations. Colors of the

symbols represent the habitat definition according to Pham et al.[31] ordered by apparent age: Riverine Input (RI, dark gray), Young Plume Core (YPC, red), Outer Plume Margin (OPM, purple), Western Plume Margin (WPM, yellow), modified Oceanic Water (MOW, dark cyan) and Oceanic Water (OSW, blue). The extent of the Amazon River plume is illustrated by the median of the sea surface salinity at each grid point during the timing of each cruise (09 to 29/05/2018 and 21/04 to 13/05/2021) of the daily surface salinity provided by the Copernicus Global Ocean Physics Analysis and Forecast[87]. Arrows represent the speed and direction of the median of the geostrophic velocity throughout each cruise as provided by the Copernicus Global Ocean Gridded L4 Sea Surface Heights[88].

https://doi.org/10.1038/s42003-026-09893-4                                                                          **Article**

waters (OSW). In the Amazon shelf close to the river mouth, the geostrophic velocities revealed an intricate and strong circulation as the riverine waters spread at surface in this energetic region[33] (Fig. 2). Additionally, some differences in the strength of the North Brazil current and the variety of eddy-like structures appeared north of 7.5°N affecting the overall distribution of the plume in the region in each cruise.

The physicochemical variables measured in these six habitats reflected the changing and dynamic characteristics of the Amazon River plume (Supplementary Data 2). Sea surface temperature was warmer in the estuary, where the lower salinity surface waters (<6) were found. Moving to the north, surface temperature decreased to reach a minimum towards the older, saltier regions of the plume (OPM and MOW). The base of the mixed layer in the area under the influence of the plume ranged from 8 to 40 m, shallower than that in the oligotrophic oceanic waters, where it exceeded 70 m (Supplementary Data 2). These shallower mixed layers were related to higher stability of the water column, shown by values of the maximum of buoyancy frequency ($N^2$) above 0.02 s$^{-2}$, which in most stations was found at base of the plume. This suggests a slow vertical exchange between the plume and oceanic waters beneath, which intensified towards the plume margins, where vertical stability decreased (lower $N^2$) and surface salinity increased. As expected, surface dissolved oxygen concentration decreased with increasing salinity, but in most stations, it was consistently below the saturation concentration estimated using temperature and salinity[34] (Supplementary Data 2). The concentration of chlorophyll *a* at the surface exceeded 0.5 µg L$^{-1}$ in the habitats representing fresher parts of the plume (RI, YPC, and WPM), and decreased towards older sections of the plume and the oceanic realm. The elevated nutrient load carried by the Amazon and Pará Rivers is evident in the elevated concentrations at surface of nitrate and silicate found in the brackish estuary (RI) compared to other habitats, with a sharp decrease at stations on the shelf (YPC), although values remained relatively high at around 2 µM (Supplementary Data 2).

## Microalgae community structure

The hierarchical cluster analysis (HCA) performed on the relative contribution of the different groups of microalgae to total chlorophyll *a* revealed the existence of four different microalgae communities along the Amazon River plume during our cruises (Fig. 3C). The clustering tendency of the HCA was validated by a principal component analysis (PCA), which evidenced the separation of clusters along the first two dimensions, supporting that the HCA did not define random clusters (Fig. 3A, B). While diatoms dominated in most parts of the mouth, they were also accompanied by a relatively high abundance of presumably filamentous freshwater cyanobacteria in the oligohaline section of the plume (salinity <5.0, community 2, RI stations). Additionally, two stations within the mouth exhibited a distinct structure dominated by cryptophytes, with a complete absence of pico-cyanobacteria (community 1). Along the margin habitats, representing older plume waters, we observed a transition to a community dominated by haptophytes, where filamentous cyanobacteria (presumably *Trichodesmium*) were also relatively abundant (community 3). The community in sections of the mature plume and the ocean domain resembled one of oligotrophic waters, dominated by pico-cyanobacteria (i.e., *Synechococcus* and *Prochlorococcus*) and haptophytes (community 4). This community also showed a low percent contribution of *Trichodesmium* or diatoms, presumably associated with symbiotic diazotrophs (DDAs), in the stations affected by the plume. However, the PlanktoScope image set, showing the symbionts in some images, suggests that these two phytoplankton groups do not fully coexist in similar abundances. Instead, they trade dominance, that is, either DDAs (in station 32.06 in MOW) or *Trichodesmium* (in station 30.12 in OPM) were in greater abundances relative to the other group. Dinoflagellates represented a small percent in most of the samples with the exception of stations 8.02 in 2018 and 24.01 in 2021. It is notable that each microalgae community consisted of not only a single dominant habitat, but also samples from the habitat that preceded and succeeded it, highlighting the gradual transition that occurs along the plume as it ages.

## Amino acids nitrogen isotopes and trophic mode of seston along the Amazon River plume

Most of our seston samples fell around the autotroph trophocline (TP$_{Glu}$ 1.0), pointing to a clear dominance of autotrophy along the Amazon River plume and oceanic realm in both 2018 and 2021 (Fig. 4A). However, we also found mixotrophs (TP$_{Glu}$ 1.5) at three stations in 2021 (in the large size fraction of 17.17 and both size fractions of 21.07 and 24.01), and at five stations in mature plume waters during 2018 (in the total seston of stations 6.04, 22.04, 24.03, 25.03 and 31.11), suggesting a combination of autotrophy and osmotrophy, mostly in the Outer Plume Margin habitat (Fig. 4A).

TP$_{Ala}$ confirmed a dominance of autotrophic samples along the plume (Fig. 4B), with only 10 samples falling around or above TP$_{Ala}$ 1.5, suggesting a contribution of active protistan phagotrophs to the signatures of our seston samples (Fig. 4B). Five of these samples with deviations of TP$_{Ala}$ from that of autotrophs, coincided with stations where predominantly osmotrophs (TP$_{Glu}$ 1.5) were found (stations 6.04 and 24.03 during 2018, and the nano +micro size fraction in stations 17.17, 21.07 and 24.01 during 2021), while the other five corresponded to stations with a clear autotrophic TP$_{Glu}$ (both size fractions in station 20.05 and the pico-fraction in stations 8.07, 19.06, and 26.1 during 2021, mainly in the oceanic habitat). By contrast with osmotrophy, the imprint of phagotrophy was found in almost every habitat sampled along the Amazon River plume (Fig. 4B).

TP$_{Glu}$ and TP$_{Ala}$ did not show any linear correlation with the environmental drivers found along the plume (data not shown), hence, we applied a supervised machine learning algorithm named C5.0 for exploring the variables predicting trophic position, which is capable of handling non-linear relations. The C5.0 decision tree model[35] selected the depth of the mixed layer, surface oxygen concentration, the absolute value of the maximum buoyancy frequency at each station, and surface chlorophyll *a* concentration as the most relevant variables for predicting the trophic mode of seston based on TP$_{Glu}$ among all the environmental variables chosen (Fig. 5). The relative attribute usage, or the percent of cases included in each branch of the classification tree for each classifier, was 100% of cases for mixed layer depth, 73.9% for surface oxygen concentration, 52.2% for the maximum of buoyancy frequency, and 21.7% for surface chlorophyll *a*. The model only misclassified the large size fraction of station 17.17 in 2021 as Dominant Autotroph, rather than Dominant Mixotroph, for an overall error rate of 2.2%. Applying the same approach to TP$_{Ala}$ generated no pattern or explanatory variables. For the scope of this study, we thus focused on the canonical TP$_{Glu}$, which accounts for osmotrophy and photo-autotrophy, for discussing the main trophic position of our seston samples along the Amazon River plume.

## Discussion

The dynamic nature of the Amazon River plume is evident in the daily evolution of sea surface salinity and geostrophic velocities provided by the Copernicus Marine Service[33], which illustrate why distinct habitats emerge as the plume ages. During our cruises, the plume extended as far as 15°N, delineating three habitats in 2018 and six in 2021 (Fig. 2). These planktonic habitats, identified by Pham et al.[31] following the approach proposed by Weber et al.[36], were based on physicochemical drivers and capture the age of the plume better than single proxies like salinity or distance to the Amazon shelf. This age accounts for the history of the waters as they travel along the path of the plume, and determines the direction of the microalgae succession as the habitats suitable for each group emerge[37]. The apparent age of plume waters (Supplementary Fig. 1), estimated by concurrent radium isotopes (Supplementary Note 1) measured on CTD samples during our 2018 cruise[38] and an additional cruise in 2019[39], both included in Pham et al.[31], showed that the four habitats sampled in these surveys captured the aging gradient as follows: Young Plume Core (YPC, apparent age 12.9 ± 6.8 days, $n = 8$), Old Plume Core (OPC, 14.1 ± 6.8 days, $n = 9$), Outer Plume Margin (OPM, 26.6 ± 4.3 days, $n = 23$) and Western Plume Margin (WPM, 31.7 ± 5.8 days, $n = 9$). The evolution of age between these habitats seems gradual rather than abrupt, therefore, habitats with different physicochemical forcing may contain waters of similar age as a result of the varied

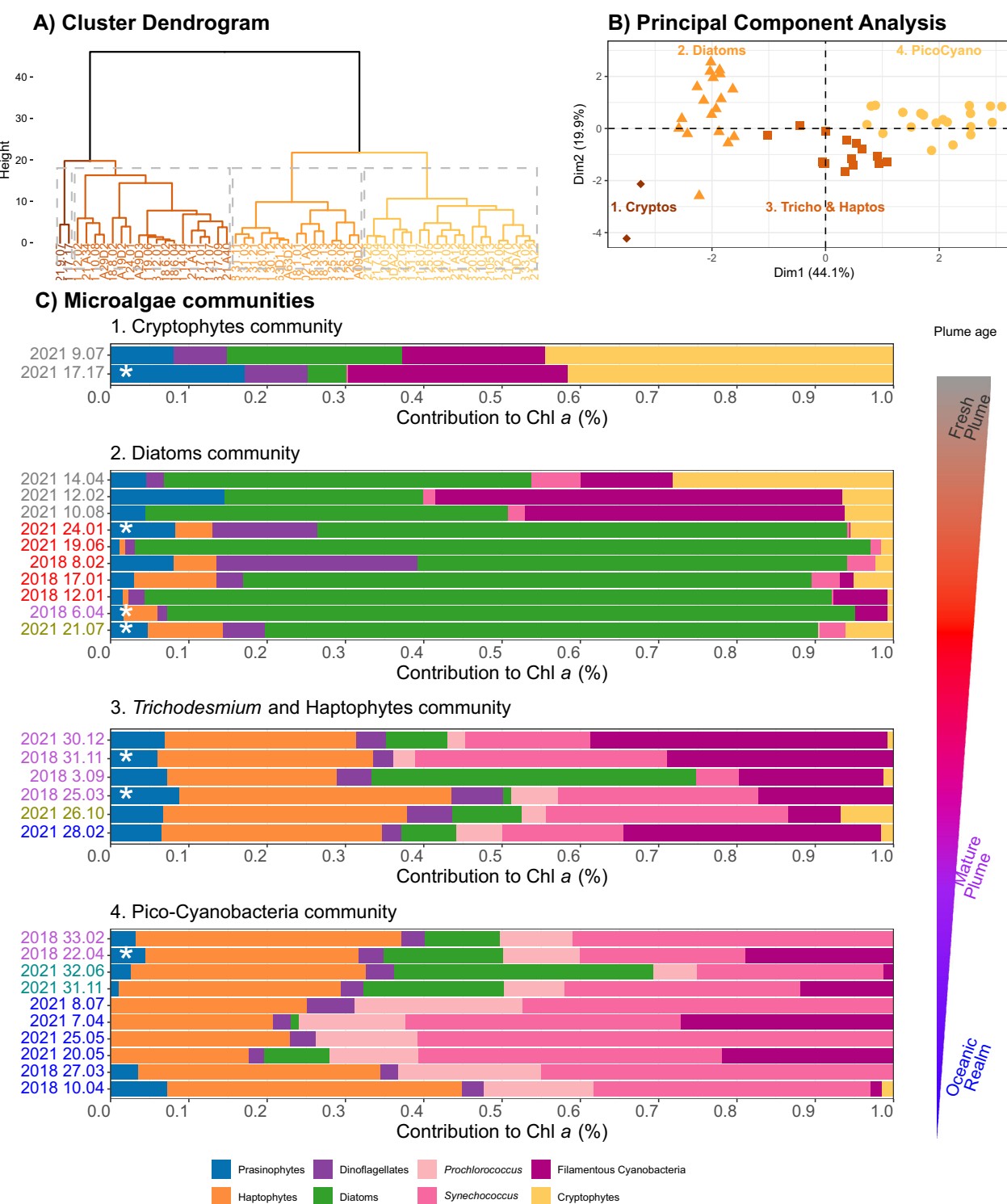

**Fig. 3 | Microalgae communities at surface along the Amazon River plume.**
Structure of the pigmented plankton community derived by chemotaxonomy based on specific pigment markers during 2018 and 2021 cruises along the Amazon River plume. Pigments were collected at surface at 55 stations for a total of *n* = 55 individual samples. **A** Four distinct communities were identified by hierarchical cluster analysis and Manhattan distances as shown in the dendrogram (*n* = 55). **B** The separation of the clusters is not random, as confirmed by the principal component analysis (PCA), where the points are colored according to the clusters in the dendrogram, and it is possible to see clear clusters distributing along the first two principal components (Dim 1 and Dim2, *n* = 55). **C** Stacked bars represent the

relative contribution of each group to total chlorophyll *a* in our stations for CSIA (*n* = 28, note that the sample in st 24.03 in 2018 was missed). Colors of the sample name in the stacked bars plot represent the different habitats defined by Pham et al.[31] ordered by apparent age: Riverine Input (RI, gray), Young Plume Core (YPC, red), Outer Plume Margin (OPM, purple), Western Plume Margin (WPM, dark yellow), modified Oceanic Water (MOW, dark cyan) and Oceanic Water (OSW, blue). The stations where mixotrophy (TP$_{Glu}$ 1.5) was dominant are marked with a white asterisk. The gradient in the triangle illustrates the potential decreasing impact on the communities of the physicochemical forcing driven by the plume as the water ages.

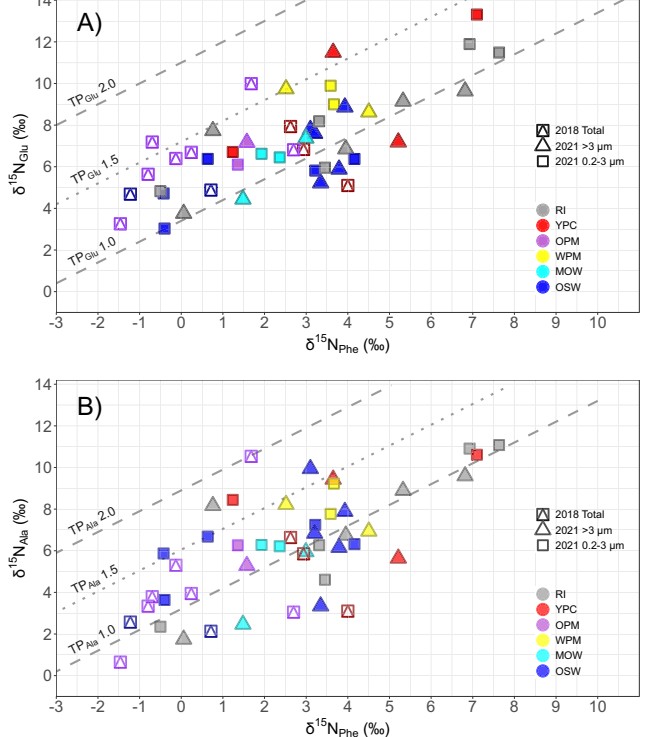

**Fig. 4 | Trophic position of the seston sampled at surface along the Amazon River plume based on Glu and Ala relative to Phe. A** $\delta^{15}N_{Phe}$ (phenylalanine) vs $\delta^{15}N_{Glu}$ (glutamic acid + glutamine) of the seston samples. **B** $\delta^{15}N_{Phe}$ (phenylalanine) vs $\delta^{15}N_{Ala}$ (alanine) of the seston samples. All values expressed in delta notation ($\delta^{15}N$ ‰, relative to atmospheric $N_2$). In 2018, total particles (combined square and triangle) were collected, while in 2021, seston was separated into two size fractions (triangles for >3 μm, squares for 0.2 − 3 μm). A total of *n* = 46 individual samples were collected at surface in 29 stations along the Amazon River plume. Triple analytical measurements were done for each sample, with a reproducibility for individual amino acids typically better than 1‰. The gray dashed and dotted lines represent the trophoclines like in Fig. 1A, B. Colors represent the different habitats defined by Pham et al.[31] ordered by apparent age: Riverine Input (RI, gray), Young Plume Core (YPC, red), Outer Plume Margin (OPM, purple), Western Plume Margin (WPM, yellow), modified Oceanic Water (MOW, cyan) and Oceanic Water (OSW, blue).

gradients in hydrological and biogeochemical properties of the plume[31]. Two additional plume habitats were sampled in 2021, but not in 2018 or 2019. They seem to represent the youngest (Riverine Input, RI) and the oldest (modified Oceanic Water, MOW) plume waters. Seston in RI exhibited bulk $\delta^{13}C$ values below −27‰ (Supplementary Fig. 2B), which are consistent with previous studies reporting suspended terrestrial material between −34 and −27‰ in the Amazon shelf[40−42]. Hence, these waters containing fluvial material are younger than those in YPC, where seston $\delta^{13}C$ up to −19‰ points to a clear shift towards marine particles[43]. Lastly, the physicochemical similarities between MOW and OSW (Oceanic Water without influence of the plume), discussed elsewhere[31], suggest that MOW represents the oldest waters of the plume, above the average apparent age of 32 days in WPM.

In May 2018 and April–May 2021, we sampled surface seston along the Amazon River plume as a total sample (>0.2 μm) in 2018 and size fractionated into picoseston (0.2−3 μm) and the combination of nano- and microseston (3−200 μm) in 2021. The bulk $\delta^{13}C$, C:N ratio, and the relationship between particulate carbon and chlorophyll *a* suggest that the seston was clearly dominated by microalgae (Supplementary Fig. 2, Supplementary Note 2), thus representing the dominant trophic position (TP) at the base of the food web. Here an exact $TP_{Glu}$ or $TP_{Ala}$ of 1.0 indicates pure

autotrophy[44], while deviations between 1.0 and 1.4 are interpreted as a predominance of autotrophy. The canonical $TP_{Glu}$ is derived from the comparison of nitrogen stable isotope values between the source amino acid phenylalanine (Phe), which barely changes with trophic transfers, and the trophic amino acid glutamic acid (Glu), which integrates the effect of trophic steps[19,24,45,46]. This seamless integration of trophic transfers in the isotopes of Glu at lower trophic levels is due to the metabolic pathways in heterotrophs, where Glu is central to the metabolism of nitrogenous compounds in organisms, acting as a precursor to most amino acids in reactions involving deamination or transamination, e.g., to alanine (Ala), proline or asparagine[23]. Consequently, organisms combining both autotrophic and heterotrophic nutrition should occupy an intermediate position between $TP_{Glu}$ 1.0 and $TP_{Glu}$ 2.0 (pure herbivory), as it was previously observed in studies of mixotrophic foraminifera and corals[47,48]. Therefore, when mixotrophy dominates the community, the $TP_{Glu}$ of microalgae-derived seston will distribute around the trophocline of $TP_{Glu}$ 1.5. Indeed, we identify this imprint of mixotrophy in our Amazon River plume samples. Interestingly, it was prevalent in mature sections of the Amazon River plume in the Outer Plume Margin (Figs. 2 and 4A), a habitat representing waters with an average apparent age of 27 days, and comprising a large extension of the plume, where a substantial sink of atmospheric $CO_2$ was previously described[32]. In this habitat, mixotrophs seem to find the optimal environmental settings required for taking over pure autotrophs and heterotrophs[9]. Elsewhere along the Amazon River plume, our samples predominantly reflected the dominance of autotrophy, clustering around the $TP_{Glu}$ 1.0 trophocline, and falling below the range defined for mixotrophs.

In the intricate dynamic environment of the plume, where waters of various ages coexist throughout the year[30], and the plume ages delimiting distinct biogeochemical habitats[31,36], it is likely that non-linear interactions among multiple physicochemical drivers influence the dominant nutrition (autotrophy vs mixotrophy) at the base of food webs at a given time. The most relevant predictors for $TP_{Glu}$, identified by the C5.0 classification model[35], include, in order of importance, the depth of the mixed layer (m), the concentration of oxygen at surface (mM), the absolute maximum of buoyancy frequency ($s^{-2}$), and the concentration of chlorophyll *a* (μg $L^{-1}$) at surface (Fig. 5). Mixotrophy was predominantly found at stations with relatively shallow mixed layers (≤37 m), in areas where the plume had a thickness between 5 and 35 m (Supplementary Data 2). The concentration of dissolved oxygen serves as a proxy for dominant heterotrophic processes, as heterotrophy consumes more oxygen than is produced by oxygenic photosynthesis[49]. Mixotrophy-rich regions presented oxic but undersaturated surface waters (Supplementary Data 2), likely due to photoheterotrophy, which has been suggested as a growth strategy for these organisms[12]. If photoheterotrophy dominates, mixoplankton may use the photosynthetic apparatus for energy production rather than carbon fixation, thus contributing to oxygen consumption. Moreover, mixotrophy dominance was observed at stations with higher chlorophyll *a* relative to similar locations (>0.171 μg $L^{-1}$), likely reflecting the production of a larger carbon stock by mixoplankton relative to phytoplankton suggested by Mitra et al.[50], which could potentially drive the accumulation of refractory dissolved organic matter over time contributing to mitigation of atmospheric carbon dioxide by the biological carbon pump[9].

Interestingly, our stable isotopes approach shows a dominance of mixotrophy at stations 6.04 and 24.03 during 2018, with waters close to oxygen saturation and a relatively less stable water column (Brunt Väisälä $N^2 \leq 0.006$ $s^{-2}$, Supplementary Data 2). The pigmented community at station 6.04 consisted mainly of diatoms with minor contributions from cryptophytes, prasinophytes, haptophytes, dinoflagellates, and *Trichodesmium* (Fig. 3C). Although all of the minor groups in this station, except *Trichodesmium*, are known members of the mixoplankton combining osmo-phago-photoautotrophy[1,50], it is very unlikely that they were able to override the potential dominance of diatoms in the nitrogen signatures of Glu and Ala of the whole sample, because the sum of all of them accounted for only 10% of the chlorophyll *a*. Rather, it seems that diatom osmotrophy of ambient organics, previously described as relevant in other systems[3,15],

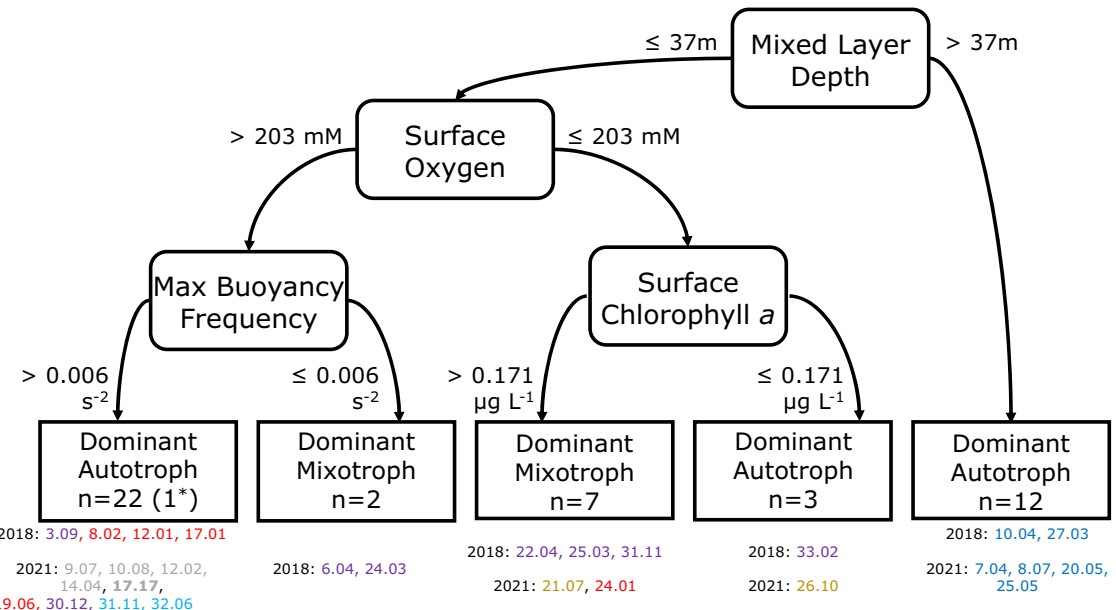

**Fig. 5 | Predictors of the dominant trophic position in seston at surface along the Amazon River plume.** Classification tree based on C5.0 showing the most likely predictors of the dominant trophic mode of seston based on $TP_{Glu}$ and the thresholds defined by the model for each of them. Note the model misclassified one sample (17.17 mixotroph was classified as autotroph, accounting for a 2.2% error).

The stations classified in each branch are shown below the tree colored by the different habitats defined by Pham et al.[31] ordered by apparent age: Riverine Input (RI, gray), Young Plume Core (YPC, red), Outer Plume Margin (OPM, purple), Western Plume Margin (WPM, yellow), modified Oceanic Water (MOW, cyan) and Oceanic Water (OSW, blue).

was responsible for the signature of this sample. This suggests that under certain environmental settings osmotrophy actively contributes to meeting nutrient needs of microalgae, rather than simply recovering leaked metabolites[51].

The functional diversity of plankton identified and quantified using nitrogen stable isotopes of amino acids (CSIA-AA) can be complemented by microalgae community structure measures using the matrix factorization procedure named ChemTax[52,53]. Despite a number of caveats discussed extensively in the literature[54–56], ChemTax is a simple method for evaluating the community structure of a mixture of pigmented pico-, nano- and microplankton through a single analysis, and detecting the presence of protist combining osmo-phago-photoautotrophy, such as dinoflagellates, prasinophytes, haptophytes or cryptophytes[1,57]. This taxonomical classification method assumes that all microalgae require pigment assemblages for harvesting/dissipating light for photosynthesis, with some of them exclusive for each group[58]. During boreal springs of 2018 and 2021, we identified four different microalgae communities along the Amazon River plume by chemotaxonomy based on pigment markers. Each community included microalgae not only from one dominant planktonic habitat type but from habitats representing the precursor or the subsequent stage of this dominant habitat, illustrating the gradual transition between habitats and underscoring the role of river plume age structuring the microalgae communities along the Amazon River plume[59–61]. In RI, we observed two distinct communities: one dominated by cryptophytes (community 1) and the other by diatoms (community 2). We hypothesize that these communities represent different microalgae assemblages originating from the Amazon or neighboring Pará river, as indicated by the consistently fluvial bulk $\delta^{13}C$ values of these samples (Supplementary Fig. 2B). Given the intricacy of swirling patterns in the estuary, suggested by the geostrophic velocities, it is possible that the communities were eventually trapped and transported unaltered within coherent eddy-like structures. Nevertheless, both communities in the RI harbor varying amounts of filamentous cyanobacteria, likely *Nostoc*, *Anabaena*, or *Oscillatoria*, known to be abundant in the Amazon River basin[62,63], which could still be viable in these brackish waters, with salinities below 6.

The diatom-dominated community (community 2) was primarily distributed in the pathway of the North Brazil current, where younger

waters swiftly travel along the coast, reaching as far as 8°N[30], and still containing ample nitrate and silicate to support a non-diazotrophic diverse diatom community[64]. Adjacent to this rapid advective pathway conveying the young plume, older saltier plume waters are typically found in the northern sections[30], in the regions of our study included in the plume margin habitats OPM and WPM. Overall, the plume waters undergo slow exchange with adjacent oceanic waters, acting as a barrier to vertical mixing[30], thus resembling a semi-enclosed system where the different microalgae communities reflect different stages of microalgae succession[37,64]. While diatoms dominate in younger waters, once the riverine nitrate is exhausted, residual phosphorus seems enough to sustain *Trichodesmium* along with a larger contribution of haptophytes and *Synechococcus* (community 3). However, as the succession progresses, a community more akin to the nitrogen-depleted oceanic end emerges, dominated by picoplankton (*Synechococcus* and *Prochlorococcus*), with diatoms and *Trichodesmium* also present (community 4). Notably, previous studies in the region[59,60] and our own image set at stations 31.11 and 32.06 in 2021, indicate that diatom diazotroph associations and *Trichodesmium* do not fully coexist, but tend to occupy different sections of the plume with little overlap.

As shown, prasinophytes, haptophytes, dinoflagellates and cryptophytes were present in all microalgae communities. These functional groups of unicellular eukaryotes are known to contained mixoplankton species, coupling osmo-phago-photoautotrophy[1,57]. Therefore, in locations were mixotrophy was the main mode of nutrition in the community, these organisms will drive the dominant trophic position, where the consumption of other microorganisms by phagotrophy coupled with osmotrophy will unambiguously impact the signatures of Ala and Glu.

The heterotrophic reworking of amino acids acquired by osmotrophy or phagotrophy seems to reflect distinctively on the nitrogen isotope signatures of Glu and Ala, producing deviations from the signatures of autotrophs, and capturing the trophic shift[16–18]. To define the ranges of variation in trophic position associated to each feeding strategy of unicellular plankton, we conducted a literature review of the different nitrogen isotopic end members (Fig. 1, Supplementary Data 1): i) microalgae, fungi, *Archaea* and bacteria growing autotrophically on inorganic nitrogen sources (DIN)[16,24,65,66]; ii) fungi, *Archaea* and bacteria growing heterotrophically on an organic nitrogen source (DON)[16]; and iii) strict protist phagotrophs

preying autotrophic microalgae (N from preys)[18]. The canonical $TP_{Glu}$ was used for discussing the main trophic position in the sections above. However, $TP_{Glu}$ only accounts for mixotrophs combining osmotrophy and photoautotrophy. To determine whether these are also mixoplankton or not, a comparison with $TP_{Ala}$ is required. This way it would be possible to resolve the contribution of nitrogen derived from the phagocytosis of preys, hence allowing the subsequent identification of the trophic mode of mixoplankton combining osmo-phago-photoautotrophy.

If we compare the $TP_{Glu}$ and $TP_{Ala}$ of our seston samples (Fig. 4) with the literature end members (Fig. 1), we should be able to resolve the predominance of photoautotrophy, osmo-photoautotrophy, phago-photoautotrophy and osmo-phago-photoautotrophy in the community. Five of our mixotrophic seston samples showed a relatively balanced contribution of DIN and DON in their $TP_{Glu}$, which is raised to 1.5 (Supplementary Fig. 3A). However, their $TP_{Ala}$ was still around 1.0, undistinguishable from the autotrophic end member in literature (microalgae and microbes) or our own autotrophic seston (Supplementary Fig. 3B). This suggests these five $TP_{Glu}$ mixotrophic samples were mixotrophs combining only osmotrophy and photoautotrophy. By contrast, five of our autotrophic $TP_{Glu}$ (1.0) samples, pointing to the absence of DON incorporation (Supplementary Fig. 4A), presented a clear distribution around $TP_{Ala}$ 1.5. This suggests a combined contribution of the uptake of DIN and nitrogen derived from the phagocytosis of preys to their nitrogen isotopes (Supplementary Fig. 4B). The last five of our ARP seston samples around $TP_{Glu}$ 1.5 (Supplementary Figs. 5A, 6A), also distributed around $TP_{Ala}$ 1.5 (Supplementary Figs. 5B, 6B), which suggests a contribution of autotrophy (DIN), osmotrophy (DON) and phagocytosis (nitrogen from preys) to their signatures, that is, a predominance of mixoplankton in our samples. However, given the lack of specific data on mixoplankton in literature, future studies addressing the impact of the incorporation of DON and N from phagocytosis on the nitrogen isotopes of different trophic amino acids in mixoplankton are still required to fully validate and extend the application of this amino acid nitrogen isotope approach in situ. For example, it is essential to determine the trophic discrimination factor of Ala and Glu in mixoplankton in order to accurately define TP based on the combination of these two amino acids, instead of only one, which is the most common practice. As shown above, the reason is that Glu only accounts for osmotrophy, and the impact on Ala was not specifically defined for mixoplankton. Nevertheless, our study demonstrates the potential usefulness of combining the nitrogen stable isotopes from different amino acids in seston to detect mixoplankton activity in the field, provided that samples primarily reflect the signature of microalgae.

In summary, the Amazon River plume encompasses a complex and dynamic array of habitats as it moves and ages in the Western Tropical Atlantic Ocean. The interplay of varying physicochemical drivers intricately regulates plankton structure and trophic mode, ultimately influencing the mass and energy available to upper trophic levels. In the present study, we observed a predominance of mixotrophs in the mature waters of the Outer Plume Margin, a habitat shaped by specific environmental conditions[31] comprising an average apparent age of 27 days. In this dynamic system, it is likely that the microalgae succession, occurring after the depletion of river-derived nitrogen by diatoms, favored osmo-phago-photoautotrophy as the optimal growth strategy in mature plume waters. Our results underscore the urgent need to study the pathways of mixotrophic mass and energy in the field applying complementary approaches[67]. It seems that trophic transfer as well as mitigation of atmospheric $CO_2$ may be more efficient when mixoplankton dominates through the production of a larger, better quality biomass[10,11] as well as driving the accumulation of refractory dissolved organic matter[9], which may contribute to the sink of $CO_2$ associated with the plume[32].

Although our findings are consistent overall, there are several uncertainties and future directions that deserve consideration. The application of compound-specific nitrogen isotope analysis of amino acids in marine ecology is relatively recent. Over the last 20 years, the field has grown

steadily, driven by a community of critically minded scientists working to validate this approach and extend its application to solve complex questions not only in marine ecology but also in geochemistry and paleoceanography[68–70]. However, despite this progress, the CSIA-AA community has not yet paid attention to mixoplankton, a group recently recognized as ecologically relevant in aquatic systems[7,67,71]. Most of the microalgae investigated to date have been prokaryotic or eukaryotic species cultured autotrophically in laboratories and regarded simply as 'photoautotrophs' or $TP_{Glu}$ 1.0 organisms growing mainly on inorganic nutrients. Given this gap, the approach we propose for detecting mixoplankton (i.e., using the combination of glutamic acid and alanine nitrogen isotopic signatures) is largely conceptual. It is based on the strict regulation of amino acid metabolism and on the consistent patterns of $^{15}N$ enrichment observed across trophic levels in the literature (Fig. 1). Yet, there remains a lack of understanding regarding the specific nitrogen isotope fractionation associated with the autotrophic and heterotrophic metabolic routes of mixoplankton, and how these processes control the isotopic differences of trophic amino acids relative to the quality of their nitrogen sources. For instance, does nitrogen incorporation by phagotrophy in mixoplankton mask the nitrogen isotopic signature of glutamic acid in the same way it does in protozooplankton? To properly establish the base of food webs in CSIA-AA studies, it is urgent to investigate the full range of nutritional strategies employed by mixoplankton and their impact on amino acid nitrogen isotope fractionation. It is also essential to determine the trophic discrimination factors of alanine and glutamic acid in mixoplankton to enable accurate trophic position calculations based on nitrogen isotopes from both amino acids, rather than relying on the nitrogen isotopic signature of a single trophic amino acid. This would help make the protistan trophic steps more visible through the food web[72]. Additionally, it is imperative to assess whether the transfer of mass and energy from mixoplankton is indeed more efficient than from phytoplankton or not. Finally, a specific characterization of the preferential metabolic routes for prey-derived amino acids in both protozooplankton and mixoplankton would be extremely valuable. Such work would establish the foundation for understanding the differential nitrogen isotope enrichment of alanine and glutamic acid observed in previous studies[16–18], and open the application of CSIA-AA in field studies related to mixoplankton.

## Methods
### Sampling and hydrography
To investigate how the changing environments along the Amazon River plume (ARP) shape the structure of the plankton food web at surface, we conducted two cruises along the plume at the beginning of the peak discharge season in May 2018 and April–May 2021. Sampling took place during expeditions EN614 on board RV Endeavor[73] and M174 on board RV Meteor[74]. In 2018, the Endeavor departed from Bridgetown (Barbados) on May 8th and arrived in San Juan (Puerto Rico) on June 1st. In this survey, a total of 67 CTD casts were carried out at 19 stations covering different regions and age stages of the ARP. In 2021, the Meteor departed from Las Palmas on April 12th and arrived in Emden (Germany) on May 30th. The survey along the ARP began on April 21st and ended on May 13th, and a total of 114 CTD casts were carried out at 23 stations from the mouth of the Amazon and Pará rivers to the waters east Barbados to cover the largest possible geographical extension of the plume. For the scope of this study, we collected suspended particles at surface at 12 stations during 2018 and 17 during 2021[33] (Fig. 2).

During both cruises, pressure, temperature, conductivity, and chlorophyll $a$ fluorescence were measured continuously by a SeaBird SBE-911+ attached to a rosette equipped with 10 L Niskin (2018) or flow (2021) bottles. In 2021, additional photosynthetically active radiation (PAR), turbidity, and SeaBird SUNA Nitrate sensors were attached to the rosette. The Brunt-Väisälä or buoyancy frequency ($N^2$) was derived using the potential density and its primary maximum was used for defining the depth of the mixed layer at each cast.

## Dissolved inorganic nutrients

Water samples for nutrient analysis were taken from Niskin/flow bottles fired at 5 to 12 depths through the water column. Inorganic nutrients (phosphate, silicate, nitrite, and nitrate+nitrite in both cruises, and ammonium in 2021) were analyzed on board colorimetrically according to Hansen and Koroleff[75] using a 4-channel Lachat Instruments QuikChem 8500 Series 2 autoanalyzer in 2018 and a QuAAtro, Seal Analytical continuous segmented flow analyzer in 2021. In 2018, the limits of detection of the instrument were 0.01 µM for phosphate, 0.02 µM for silicate, 0.02 µM for nitrate and 0.01 µM for nitrite. In 2021, the limits of detection were 0.01 µM for phosphate, 0.3 µM for silicate, 0.02 µM for nitrate, 0.01 for nitrite, and 0.03 for ammonium. The depths of the nitracline (1 µM), phosphocline (0.1 µM), and silicacline (2 µM) were estimated by linear interpolation, or assumed to be zero when the concentrations at the surface exceeded the threshold.

## Habitats delineation

Habitats shaped by similar physicochemical forcing were defined following the approach developed by Weber et al.[36] and based on simple, commonly measured, and relevant environmental variables in our study system. The procedure is described in detail in Pham et al.[31] who made the classification of habitats in six cruises along the Amazon River plume from 2010 to 2021, including our 2018 and 2021 cruises. Briefly, a hierarchical cluster analysis, complemented by a principal component analysis for validating the clustering tendency, was performed on the depth of the Deep Chlorophyll Maximum (DCM), the depth of the Mixed Layer (MLD), Sea Surface Temperature (SST), Sea Surface Salinity (SSS) and a custom-defined nitrate availability index (NAI). This NAI was calculated following Weber et al.[36] as:

$$
NAI = \begin{cases}
\left[NO_x\right]_{surface}, & if \left[NO_x\right]_{surface} \geq 0.5\mu M \\
-Z_{[NO_x]} = 2\mu M, & if \left[NO_x\right]_{surface} < 0.5\mu M \\
-Z_{bottom}, & else
\end{cases}
$$

where $[NO_x]$surface is the concentration of nitrate+nitrite at surface, $Z_{[NOx]}$ represents the depth at which nitrate + nitrite concentration reaches 2 µM, and $Z_{bottom}$ is the bottom depth of the CTD profile.

Eight habitats were defined in the meta-analysis of six cruises made by Pham et al.[31]. During our cruises, we took seston samples in three (2018) and six (2021) of these habitats (Fig. 2), varying in plume influence and age, and named as Riverine Input (RI), Young Plume Core (YPC), Outer Plume Margin (OPM), Western Plume Margin (WPM), modified Oceanic Water (MOW) and Oceanic Water (OSW). The habitat with the freshest plume water was RI, WPM and MOW were the ones containing the oldest plume waters in apparent age (Supplementary Note 1), while OSW was the habitat with absence of plume influence.

## Microalgae community structure

Samples for the analysis of diagnostic microalgae pigments at the surface were transferred directly from Niskin/flow bottles into opaque polycarbonate bottles. Soon after collection, a volume between 50 mL and 4.3 L, according to the density of particles, was filtered under low vacuum through 25 mm GF/F (0.7 µm nominal pore size) filters in 2018 or 25 mm Advantec GF-75 (0.3 µm nominal pore size) filters in 2021. Filters were then flash-frozen and stored in liquid nitrogen (−196 °C) until analysis at NASA GSFC using an Agilent RR1200 with a programmable autoinjector (900 µl syringe head), refrigerated autosampler, degasser, and photo-diode array detector with deuterium and tungsten lamps. Diagnostic pigments were measured by High-Performance Liquid Chromatography (HPLC), as described in Van Heukelem and Thomas[76] and Hooker et al.[77]. Diagnostic pigments were assessed at all stations in this study except station 24.03 in 2018, where pigments where not sampled.

The composition of the surface microalgae community was determined by the concentrations of the HPLC diagnostic pigments and the ChemTax matrix factorization procedure proposed by Mackey et al.[52] in

version 1.9.5 of the software designed by Wright[78]. The groups selected for this analysis were based on existing literature and represented groups previously identified along the plume[60,61,79,80], also confirmed by the flow microscope images collected during 2021 as explained below. These previous studies in the region reported the presence of representatives of cryptophytes, cyanobacteria (in particular, *Trichodesmium, Synechococcus, Prochlorococcus*), dinoflagellates, diatoms, haptophytes and prasinophytes. The initial pigment ratios (Table 1) were averaged using relevant literature studies contained in Higgins et al.[58]. ChemTax was run three times using 60 random ratio matrices until the contribution of each group to total chlorophyll *a* converged. For the second and third runs, the initial ratio matrix for randomization consisted of the average of the six best matrices in the previous run. The final ratio matrix is also shown in Table 1. The contribution of each group to total chlorophyll *a* provided by ChemTax in µg chla L⁻¹ was converted to a relative percent contribution for better assessing the changes in the dominant groups along the Amazon River plume.

In selected stations during 2021 cruise, microplankton samples were collected at surface from the ship's clean water supply passing an unknown volume into a 10 µm plankton hand net, filtered afterwards by a 200 µm sieve to remove mesozooplankton, and imaged in vivo by a PlanktoScope v2.1[81] for qualitatively describing the composition of the pigmented microplankton community (*n* = 19). Individual cells were segmented from the full-field pictures using the built-in segmentation function of the instrument. Afterward, image data were assigned to taxonomic categories using random forest algorithms and validated by experts on the web-based platform Ecotaxa[82]. The classified image dataset is available at https://ecotaxa.obs-vlfr.fr/prj/6346.

A partial and qualitative validation of the output of ChemTax was conducted by combining different concurrent measurements during each cruise. In 2018, flow cytometry samples were used to determine the presence or absence of *Synechococcus* and *Prochlorococcus*, while 18S-RNA analysis was used to identify the presence of diatoms, prasinophytes, and dinoflagellates at stations where data were available[83]. Additionally, in 2021, the presence of diatoms, dinoflagellates, and *Trichodesmium* in the microplankton fraction was confirmed using a flow imaging microscope (i.e., PlanktoScope v 2.1[81]). Although the application of all these complementary approaches does not enable validation of the absolute contribution of each group to total chlorophyll *a*, it confirms the presence of the different groups, thereby supporting the structure of the pigmented plankton community found along the Amazon River plume. One possible caveat in our analysis is that the literature ratios for *Trichodesmium* and *Synechococcus* depend on contrasting ratios of the same pigment marker (zeaxanthin). However, validation with the alternative variables mentioned above suggests that ChemTax effectively resolved meaningful biomass of each group. For instance, ChemTax did not identify the presence of *Trichodesmium* at stations 19.06 and 24.01 in 2021 (Fig. 3), where this organism was not observed in the PlanktoScope image set. Additionally, ChemTax did not assign any contribution of *Synechococcus* to chlorophyll *a* of station 6.04 in 2018 (Fig. 3), where it was also absent in the flow cytometry data[83].

## Elemental analysis of bulk and amino acids stable isotopes of seston

Seston samples were collected at/near the surface, using Niskin bottles (2018) or a water pump deployed in the upper 0.5 m of the water column (2021), for the analysis of bulk and compound specific isotope analysis (CSIA-AA) of amino acids carbon ($^{13}C/^{12}C$) and nitrogen ($^{15}N/^{14}N$) stable isotopes at 29 stations: 12 during 2018, and 17 during 2021 (Fig. 2). In 2018, samples for the bulk analysis were collected after passing 2.5−19 L through precombusted (4 h, 450 °C) 47 mm GF/F (0.7 µm nominal pore) filters by a pressurized air system designed for large volume filtrations. Samples for CSIA-AA were collected by passing between 6.4 and 18.7 L through 47 mm 0.2 µm Nucleopore polycarbonate filters using the same pressurized system. In 2021, we decided to size-fractionate seston to separate picoplankton (<3 µm) from the combination of nano- and microplankton (3–200 µm) as well as the free living (i.e, captured in the <3 µm fraction) from the particle-

**Table 1 | Initial and final pigment ratios introduced in and yielded by the ChemTax analysis, respectively**

| Initial Ratios | Chl b | But Fuco | Hex Fuco | Allo | Fuco | Perid | Zea | DV Chl b | Chl c1 + c2 | Chl c3 | Lut | Neo | Viola | Pras | Chl a |
|---|---|---|---|---|---|---|---|---|---|---|---|---|---|---|---|
| Cyanobacteria1 | 0 | 0 | 0 | 0 | 0 | 0 | 0.105 | 0 | 0 | 0 | 0 | 0 | 0 | 0 | 1 |
| Cyanobacteria2 | 0 | 0 | 0 | 0 | 0 | 0 | 0.453 | 0 | 0 | 0 | 0 | 0 | 0 | 0 | 1 |
| Cyanobacteria4 | 0.495 | 0 | 0 | 0 | 0 | 0 | 0.224 | 0.495 | 0 | 0 | 0 | 0 | 0 | 0 | 1 |
| Prasinophytes3 | 0.368 | 0 | 0 | 0 | 0 | 0 | 0.033 | 0 | 0 | 0 | 0.013 | 0.051 | 0.035 | 0.145 | 1 |
| Cryptophytes | 0 | 0 | 0 | 0.359 | 0 | 0 | 0 | 0 | 0.145 | 0 | 0 | 0 | 0 | 0 | 1 |
| Diatoms1 | 0 | 0 | 0 | 0 | 0.711 | 0 | 0 | 0 | 0.074 | 0 | 0 | 0 | 0 | 0 | 1 |
| Haptophytes6 | 0 | 0.004 | 0.237 | 0 | 0.066 | 0 | 0 | 0 | 0.178 | 0.156 | 0 | 0 | 0 | 0 | 1 |
| Dinoflagellates1 | 0 | 0 | 0 | 0 | 0 | 0.541 | 0 | 0 | 0.197 | 0 | 0 | 0 | 0 | 0 | 1 |
| **Final Ratios** | **Chl b** | **But Fuco** | **Hex Fuco** | **Allo** | **Fuco** | **Perid** | **Zea** | **DV Chl b** | **Chl c1 + c2** | **Chl c3** | **Lut** | **Neo** | **Viola** | **Pras** | **Chl a** |
| Cyanobacteria1 | 0 | 0 | 0 | 0 | 0 | 0 | 0.101 | 0 | 0 | 0 | 0 | 0 | 0 | 0 | 1 |
| Cyanobacteria2 | 0 | 0 | 0 | 0 | 0 | 0 | 1.343 | 0 | 0 | 0 | 0 | 0 | 0 | 0 | 1 |
| Cyanobacteria4 | 0.432 | 0 | 0 | 0 | 0 | 0 | 0.203 | 0.363 | 0 | 0 | 0 | 0 | 0 | 0 | 1 |
| Prasinophytes3 | 0.605 | 0 | 0 | 0 | 0 | 0 | 0.035 | 0 | 0 | 0 | 0.013 | 0.047 | 0.035 | 0.082 | 1 |
| Cryptophytes | 0 | 0 | 0 | 0.322 | 0 | 0 | 0 | 0 | 0.129 | 0 | 0 | 0 | 0 | 0 | 1 |
| Diatoms1 | 0 | 0 | 0 | 0 | 0.609 | 0 | 0 | 0 | 0.098 | 0 | 0 | 0 | 0 | 0 | 1 |
| Haptophytes6 | 0 | 0.009 | 0.488 | 0 | 0.096 | 0 | 0 | 0 | 0.215 | 0.211 | 0 | 0 | 0 | 0 | 1 |
| Dinoflagellates1 | 0 | 0 | 0 | 0 | 0 | 0.579 | 0 | 0 | 0.236 | 0 | 0 | 0 | 0 | 0 | 1 |

Ratios were taken from Higgins et al.[58]. Note: Cyanobacteria1 contains fresh and seawater filamentous colonial cyanobacteria such as *Trichodesmium*, *Nodularia*, or *Nostoc*; Cyanobacteria2 contains mainly *Synechococcus*; Cyanobacteria4 contains mainly *Prochlorococcus*; Prasinophytes3 contains groups like Ostreococcus, *Micromonas* or *Prasinococcus*; Diatoms1 contains microplankton diatoms such as *Thalassiosira*, *Skeletonema* or *Chaetoceros*; Haptophytes6 contains mainly *Gephyrocapsa* (former *Emiliania*); Dinoflagellates1 represents peridinin containing dinoflagellates such as *Tripos* (former *Ceratium*), *Prorocentrum* or *Peridinium*. The pigments included in the analysis were Chlorophyll *b* (Chl *b*), 19′-Butanoyloxyfucoxanthin (ButFuco), 19′-Hexanoyloxyfucoxanthin (Hex Fuco), Alloxanthin (Allo), Fucoxanthin (Fuco), Peridinin (Perid), Zeaxanthin (Zea), Divinyl Chlorophyll *b* (DV Chl *b*), the sum of Chlorophyll *c1* and *c2* (Chl *c1* + *c2*), Chlorophyll *c3* (Chl *c3*), Lutein (Lut), Neoxanthin (Neo), Violaxanthin (Viola), Prasinoxanthin (Pras), and Chlorophyll *a* (Chl *a*).

attached components of the microbial community. Bulk samples were collected by vacuum filtration by sequentially passing 0.1−4 L of water through precombusted (4 h, 450 °C) 25 mm GF/D Whatman (2.7 µm) and 25 mm GF-75 Advantec (0.3 µm) filters under low vacuum pressure (<100 mmHg). Size-fractionated CSIA-AA samples were collected again using the pressurized air system by sequentially passing 0.5−20 L of water through 47 mm 3 µm and 47 mm 0.2 µm Nucleopore polycarbonate filters. All samples were immediately dried (24–72 h, 60 °C), frozen and stored at −20 °C until analysis in the laboratory ashore. Samples were not acidified prior to analysis to prevent changes in the signature of nitrogen.

The stable C isotopes of the 2018 bulk samples were analyzed using a Micromass Isoprime 100 continuous-flow isotope ratio mass spectrometer (IRMS) coupled to a Carlo Erba elemental analyzer (NA 2500) at the Georgia Institute of Technology. In 2021, bulk samples were analyzed by EA Isolink CN (Thermo scientific) elemental analyzer coupled via a ConFlo IV (Thermo scientific) interface to a Delta V advantage (Thermo scientific) isotope ratio mass spectrometer at the Leibniz-Institute for Baltic Sea Research Warnemuende. The stability of the instruments and the contribution of blanks to our measurements were checked using a series of elemental (methionine) and isotopic (peptone) standards in each analytical run[84]. We have carried out multiple intercalibrations of these two systems.

For both cruises, stable nitrogen isotope compositions of individual amino acids were measured using an isotope ratio mass spectrometer (IRMS, Thermo Finnigan GmbH, MAT 253 MS, Germany) connected via a ConFlo IV interface unit to a gas chromatograph combustion periphery (GC-C, Thermo Scientific Trace 1310 GC, Italy; Thermo Scientific, GC Isolink, Germany) at the Leibniz-Institute for Baltic Sea Research Warnemuende. After 24 h of acid hydrolysis (6 N HCl, 110 °C), amino acids were derivatized to trifluoro-acetylated isopropyl amino acid esters following Hofmann et al.[85] and purified according to Veuger et al.[86]. An external standard of 16 individual amino acids was derivatized in the lab and run with each set of six samples. This external standard consisted of a mixture of alanine (Ala), arginine (Arg), aspartic acid + asparagine (Asp), cysteine (Cys), glutamic acid + glutamine (Glu), glycine (Gly), isoleucine (Ile), leucine (Leu), lysine (Lys), methionine (Met), phenylalanine (Phe), proline (Pro), serine (Ser), threonine (Thr), tyrosine (Tyr), and valine (Val). Additionally, an internal standard (trans-4-cyclohexane carboxylic acid) was added to each sample to assess the overall performance of the derivatization/purification procedure in the laboratory. The separation column in the GC consisted of a non-polar column coated with 5% phenyl-polysilphenylenesiloxane (BPX5, 60 m, 0.32 mm inner diameter, film thickness of 1 µm; SEG Analytical Science, Ringwood, Victoria, Australia). For each run, 2 µL of sample was injected via a PTV injector in splitless mode. The temperature program was as follows: Initial temperature at 50 °C, heat up at 12 °C/min to 120 °C, hold for 17 min, heat up at 3 °C/min to 180 °C, linger for 10 min, heat up at 5 °C/min to 200 °C, hold for 6 min, heat up to 250 °C at 10 °C/ min and hold for 7 min. This procedure allows us to separate and analyze 13 amino acids (i.e., Ala, Asp, Glu, Ile, Gly, Leu, Lys, Phe, Pro, Tyr, Ser, Thr and Val). However, here we report only the results of 12 because in 2021 Gly in our seston samples from the Amazon River plume co-eluted with a substance unknown to the NIST (National Institute of Standards and Technology) mass spectral library as identified by parallel analysis of subsamples with a single quadrupole GC-MS (Thermo Scientific ISQ 7000). Samples were analyzed in triplicate. Reproducibility for individual amino acid values was typically better than 1‰.

Bulk and amino acid specific carbon and nitrogen stable isotope abundances were expressed in δ notation (‰) relative to Vienna Pee Dee Belemnite (VPDB) and to Air, respectively. A brief explanation of the fundamental concepts of isotope ecology, including the δ notation, is given in Supplementary Note 3.

### Trophic position and nitrogen isotopes end members

The trophic position (TP) is typically assessed by comparing the measured nitrogen isotopic signatures of a source amino acid with those of a trophic amino acid (Supplementary Note 3). The canonical amino acids are phenylalanine and glutamic acid + glutamine, with the combination of their nitrogen isotope values along the equation proposed by Chikaraishi et al.[24] defining trophoclines, that is, lines representing the same trophic position. In this study, the trophic position of seston was calculated according to the equations proposed by these authors based on the nitrogen isotopes of the source amino acid phenylalanine (Phe) and the trophic amino acids glutamic acid + glutamine ($TP_{Glu}$) or alanine ($TP_{Ala}$) as follows:

$$TP_{Glu} = \frac{\delta^{15}N_{Glu} - \delta^{15}N_{Phe} - 3.4}{7.6} + 1$$

$$TP_{Ala} = \frac{\delta^{15}N_{Ala} - \delta^{15}N_{Phe} - 3.2}{5.7} + 1$$

The canonical $TP_{Glu}$ was used for defining the trophic position of seston, due to the fact that accounts for the effect of autotrophy and osmotrophy. A $TP_{Glu}$ below 1.4 represents autotrophs, and a $TP_{Glu}$ between 1.4 and 1.6 is the range of mixotrophs combining autotrophy and osmotrophy. The discussion of environmental drivers and community composition is hence focused on $TP_{Glu}$ along the Amazon River plume. The alternative $TP_{Ala}$ is used for defining the phagotrophic activity of protozooplankton and mixoplankton because alanine allows to track protistan phagotrophic trophic steps[18].

We conducted a comprehensive review of the existing literature to define the different nitrogen source end members resulting from the varied feeding strategies in unicellular organisms which reflect in TP (Supplementary Data 1). The literature data included: (i) autotrophic microalgae growing on inorganic nitrogen (DIN end member)[24,65,66]; (ii) fungi, *Archaea* and bacteria growing on ammonium (DIN end member)[16]; (iii) fungi, *Archaea* and bacteria growing on casamino acids (DON end member)[16]; and (iv) eukaryotic strict unicellular phagotrophs (protozooplankton) growing on autotrophic preys (N from phagocytosis of preys end member)[18]. The approach facilitates the analysis of the degree of isotopic enrichment of the two trophic amino acids (Glu and Ala) with trophic transfers due to the metabolism of heterotrophs (osmotrophs and phagotrophs), providing insight into the potentially different fractionation patterns associated with mixoplankton that combine photoautotrophy with osmotrophy and phagotrophy (Fig. 1, Supplementary Data 1).

### Remote sensing products and graphics

The median sea surface salinity provided by the Copernicus Global Ocean Physics Analysis and Forecast[87] was calculated at each grid point for the duration of each cruise (09 to 29/05/2018 and 21/04 to 13/05/2021) in order to illustrate the average position of the plume (Fig. 2). The median of the geostrophic velocities retrieved for the Copernicus Global Ocean Gridded L4 Sea Surface Heights[88] were also calculated at each grid point over the duration of each cruise to show the dominant direction and speed of the currents in the region (Fig. 2). Daily data was used for producing movies[33] in order to illustrate the dynamism along the Amazon River plume. Statistical analysis and plots were made by scripting R v4.0.5[89] in R Studio v2022.02.0[90]. Package *ggplot2* v3.5.1 was used for plotting[91] and *ggpubr* v0.4.0 for exporting the plots[92]. Data from the netCDF files provided by the Copernicus Marine Service was extracted and imported into a dataframe using package *ncdf4* v1.24[93], medians were estimated using built-in functions in package dplyr v 1.1.4[94]. Vector plots were made using *metR* v0.18.2[95].

### Statistics and reproducibility

The different pigmented plankton communities along the Amazon River plume were separated by a hierarchical cluster analysis (HCA) on the chemotaxonomy by HPLC samples during 2018 and 2021. The HCA was done using all the samples available at surface during both cruises ($n = 55$). The clusters were separated by Manhattan distances due to the existence of outliers, and the clustering tendency was confirmed by a principal component analysis, both done using *factoextra* v1.0.7 and *FactoMineR* v2.4 packages[96,97]. The physical, chemical and biological drivers potentially

predicting the trophic position (i.e., $TP_{Glu}$ and $TP_{Ala}$) of our seston samples ($n = 46$) were assessed using the decision tree classification machine learning model C5.0[35] in package *C50* v0.1.6[98]. This model was used because of its ability for handling non-linear relations among variables in datasets containing missing values. All our seston samples were included in the analysis with the main goal of defining the variables explaining or predicting the trophic mode of seston. The predicting variables included in the analysis were the thickness of the plume, the maximum of the buoyancy frequency, the depth of the mixed layer, the depth of the subsurface fluorescence maximum, the depth of the nitracline (depth where concentration reached $1\,\mu M$), phosphocline (depth where concentration reached $0.1\,\mu M$) and silicacline (depth where concentration reached $2\,\mu M$) and, at the surface: dissolved oxygen, temperature, salinity, nitrate to phosphate ratio, nitrate to silicate ratio, nitrate to nitrite ratio, the average size of the pigmented community estimated using pigment markers according to Bricaud et al.[99], the concentration of chlorophyll *a* and the relative contribution to chlorophyll *a* of all the groups identified by chemotaxonomy as mentioned above.

### Reporting summary

Further information on research design is available in the Nature Portfolio Reporting Summary linked to this article.

### Data availability

All data supporting the findings of this study are available within the paper and its Supplementary Information or deposited in open access repositories. Literature data are provided in Supplementary Data 1, along with the original references. Environmental data, trophic positions and $\delta^{15}N$ of glutamic acid, alanine and phenylalanine are provided in Supplementary Data 2. The apparent age of the waters along the Amazon River plume is provided in Supplementary Data3. The Movies illustrating daily changes in salinity and geostrophic velocities along the Amazon River plume are available in FigShare[33]. The nitrogen and carbon isotopes of the bulk samples, the nitrogen isotopes and mol% of all amino acids analyzed, the percent contribution of microalgae groups to total chlorophyll *a* as well as the environmental variables are available in Pangaea[100–102]. Raw HPLC pigments are available in Subramaniam[103] and Umbricht et al.[104]. The PlanktoScope v2.1 images used for validating the classification made by CHEMTAX can be accessed in https://ecotaxa.obs-vlfr.fr/prj/6346.

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

## Acknowledgements

We would like to thank the captain, crew and science parties of cruises EN614 on board RV Endeavor and M174 on board RV Meteor for their invaluable support at sea. A special thanks dedicated to all people involved in M174 as well as to chief scientist Prof. Dr. MV for making a memorable first international long term cruise in a German vessel after the hard lockdown

due to the Covid pandemic. We are grateful to Prof. RNP for the radium isotopes during EN614 and EN640, Dr. AP and Dr. JU for vivid discussions about the habitats and community structure along the Amazon River plume, to our in-house technical experts CB, LK, and ES for providing state-of-the-art nutrient measurements, and to our student helpers JL, ILS and FR for their assistance in lab work. Lastly, we would like to acknowledge our funding agencies: Deutsche Forschungsgemeinschaft (DFG) grant GPF19-1-13 to NLW and MV, DFG grant LO 1820/8-1 to NLW, DFG grant VO 487/14-1 to MV, NSF project OCE-1737078 to JPM and AS, Beatriz Galindo fellowship BG22-00067 to AFC, and NSF grant OCE-1737128, NASA grant 80NSSC21K0439 and the LDEO Climate and Life Fellowship to AS.

## Author contributions

The contribution of the authors is described following the CRediT taxonomy. Conceptualization, Investigation, Data Curation and Writing—original draft was done by N.L.W. and A.F.C., these authors jointly supervised this work. N.C., D.W., and I.L. also contributed to Investigation and Data Curation. A.F.C. contributed to Formal Analysis and Visualization. All authors were responsible for the Methodology and Writing—review & editing. Funding Acquisition was done by N.L.W., M.V., J.P.M., and A.S.

## Funding

## Competing interests

The authors declare no competing interests.
