## [Transparent Peer Review file · Communications Biology]

Mixotrophy emerges as an optimal strategy in mature waters of the Amazon River plume

Corresponding Author: Dr Ana Fernández-Carrera

Version 0:

Reviewer comments:

Reviewer #1

(Remarks to the Author)
Fernández-Carrera et al.
OVERVIEW

This is for the most part a well written work on an interesting ecosystem and tackling a timely and interesting topic. Where it falls down is mainly on two grounds, and I would invite the authors to resolve these issues.

#1 There is something of a confusion throughout the work with terminologies. In part this reflects the development of the mixoplankton paradigm terminologies themselves. However, this has now settled so it is important to use the most recent forms even though this may outwardly create some tensions with terminologies in some of the cited papers.

Thus:

Phytoplankton – all (cyanobacteria and protist) are potentially mixotrophic via photo+osmo ‘trophy. Some can grow solely on osmotrophy (e.g. diatoms in biotechnology in total darkness). In consequence, identifying ‘autotrophic’ vs ‘mixotrophic’ plankton does not make sense; all the autotrophic species are mixotrophic (the degree to which they enact mixotrophy is a separate issue).

Mixoplankton – protist only, are potentially mixotrophic via photo+osmo+phago ‘trophy.

The text confuses activities vs organism descriptors. There is thus a problem with ‘Autotrophs’ (there is no such group in reality) vs ‘autotrophy’.

#2 There is a problem with the use of the stable isotopes that needs to be recognised. Before proceeding, I should say that I think this stable isotope amino acid approach is really nice from a technological point of view and the authors are to be congratulated on its deployment. They also do caveat (rather late on) that the conceptual basis for deployment in field work needs confirmation. So, to the problem – the issue is that in tightly recycling systems, such as that being studied here, stable isotopes (explicitly d15N) do not provide (actually, cannot provide) a robust prognostic tool. This is from the work of Flynn et al. 2018, in one of those papers that users of the d13C and d15N methods perhaps do not want to acknowledge! Now, it may be that this amino acid isotope fractionation is rather robust against this problem, but the problem at least needs to be recognised. It would also provide a prompt for further work on the method. We really need this, because if correct, if robust, this approach could offer a very important step forward.

Flynn, K. J., Mitra, A., & Bode, A. (2018). Toward a mechanistic understanding of trophic structure: inferences from simulating stable isotope ratios. *Marine Biology*, 165, 1-13.

SPECIFIC COMMENTS

Graphic abstract – as all autotrophs are osmotrophs; this needs to be labelled with activities and not with organism labels
L21 This has been known for >50yrs, but rarely labelled as ‘mixotrophy’.

L24 This is the wrong way around and misleading .. ancestral mixoplankton were intermediates between phagotrophs and (then) protist phytoplankton. See Mitra et al. (2024) Trait trade-offs in phagotrophic microalgae: the mixoplankton conundrum, *European Journal of Phycology*, 59, 51-70. <https://doi.org/10.1080/09670262.2023.2216259> .

L26 the problem with identifying photo-osmotrophy is as much to do with separating protist from bacterial osmotrophy - this has been an issue reported since the 1980s. Now we recognise that bacterivory is displayed by many formerly termed ‘phytoplankton’ (now ‘mixoplankton’) the analyses and interpretations are more nuanced.

L49 why should it decrease that efficiency? That makes no sense and is contrary to evidence. (see Mitra et al. 2024

EurJPhycol)

L67 Again, where is the evidence? This is a very old paper you cite here; it perhaps makes little sense set against recent considerations.

L76 While the method has uses for sure, the use of any stable isotope approach, especially in a tightly coupled mature system must be recognised as potentially highly problematic.

L94 This paragraph looks like it was written by someone else; it is dominated by 'phytoplankton' which is inconsistent with the terminology used in the previous paragraphs. The problem is then proliferated throughout the rest of the text.

L136 Likewise this text is dominated by 'phytoplankton' .. perhaps 'microalgae' would be a better term as a catch-all?

L160 onwards: It is really important to evidence that this methodology works in these complex systems. I am not disputing that mixotrophy is important in these waters, but the most that can perhaps be evidenced using this method just now is that 'the data are consistent with' mixotrophy?

L224 OK this provides the required evidence; this should be stated clearly in the Introduction, to justify the whole approach !!

L258 Note that the model of Ward & Follows has no mechanistic basis at all. You may wish to check that model structure very carefully before citing it to evidence this point.

Cf Mitra et al. (2016) Defining planktonic protist functional groups on mechanisms for energy and nutrient acquisition; incorporation of diverse mixotrophic strategies. *Protist* 167, 106–120. <http://dx.doi.org/10.1016/j.protis.2016.01.003>

L261 evidenced by what? (the isotope signature, presumably?)

L272 I think the origin of this is Flynn & Berry LS (1999) The loss of organic nitrogen during marine primary production may be overestimated significantly when estimated using ^{15}N substrates. *Proceedings of the Royal Society Lond. B* 266; 641-647.

L283 Again there is the problem with the 'phytoplankton' word.

L316 I suggest you cross reference to the Mixoplankton Database –

Mitra et al. (2023) The Mixoplankton Database – diversity of photo-phago-trophic plankton in form, function and distribution across the global ocean. *Journal of Eukaryote Microbiology* 70 <https://doi.org/10.1111/jeu.12972>

If you have species that you can evidence as being mixoplankton that are absent from the database, it would be useful to identify those species.

L328 this is a very important issue and assumption. Given the problems of isotope discrimination through the food web it is all the more important. It has been explored via models and shown that isotopes cannot be used reliably in this fashion (Flynn et al. 2018).

L338 This is really confusing. To clarify, all phytoplankton are potentially mixotrophic via photo+osmotrophy. Mixoplankton can additionally use phagotrophy. So 'all mixotrophs' using phagotrophy strictly means that you are talking only about mixoplankton. Is that what you mean? And if so, why not use 'mixoplankton'?

L343 I suggest it indicates the potential usefulness only - we know that bulk $\delta^{15}\text{N}$ data are not sufficiently robust to direct this conclusion, though this amino acid fractionation may do the trick. (hopefully!)

Methods – nicely documented; only a few minor typos.

Fig.3 I suggest that this is relabelled as activities and not organisms. Thus 'autotrophs' is 'autotrophy' etc . The reason is that all of the 'photoautotrophs' are mixotrophs (either phytoplankton as photo+osmo, or mixoplankton as photo+osmo+phago).

Fig.4 similar comment to that in Fig.3. Where is 'osmotroph'? And I suggest that these 'herbivore', 'omnivore' and 'carnivore' labels need revisiting. Is feeding on a mixoplankton herbivory or omnivory ?! Is a 'phagotroph' as labelled here only a protist zooplankton? Do these terms help at all?! What is an ARP autotroph (all of which will be photo+osmo trophic) vs a 'ARP mixotroph'? (noting also that 'ARP' – Amazon river plume")

Fig.5 Aside for the typographic error of 'Autrotroph', again there is this problem of autotrophy vs mixotrophy.

Reviewer #2

(Remarks to the Author)

This manuscript proposes a novel approach to uncover the relevance of mixotrophic feeding within the marine microplanktonic community. By comparing the composition of stable nitrogen isotopes in amino acids the authors are able to identify the dominance of mixotrophic organisms (i.e. trophic position = 1.5) in seston samples collected in habitats of different age along the Amazon River plume. The dominance of mixotrophy was further related to the succession of phytoplankton communities along the plume. The authors conclude that mixotrophy was a mechanism to support plankton growth in aged waters, where nutrient and oxygen concentrations were relatively low. The study was well designed and the manuscript contains most of the required information to support the interpretation. The proposed approach may stimulate further research to unveil the role of mixotrophy in marine ecosystems with major implications not only for our knowledge of nutrient cycling and carbon sequestration but also for the understanding of functional diversity in marine plankton. However, before final publication the authors must consider the following issues:

L 49 "Transfer of energy from primary producers...." = transfer efficiency. Consider the relationship between transfer efficiency, food chain length and productivity. While there seems to be an universal inverse relationship between food chain length and system productivity (e.g. Legendre and Rassoulzadegan, 1985) transfer efficiency may be also influenced by multiple other factors (Décima, 2022). Mixotrophy may be well one of these factors as indicated in the following text (L 61-62).

L 129 cite Benson & Krause (1984) with number as for other references

L 132-135 Inorganic nutrient concentrations are indicated but no reference or data are provided.

L 136-139. Describe the communities in the same order as in Fig. 2: community 1, 2, 3, and 4). Avoid unnecessary repetition of methods (L 140-141).

L 159. Some information about the composition, abundance and/or biomass of the non-pigmented plankton (heterotrophic protozoa, small metazoa) would be appropriate to further characterize the trophic structure of the analyzed seston (if

available).

L 164 and Fig. 3 How were the limits for mixotrophy (dotted lines) determined? Do they represent the error in $\delta^{15}\text{NAla}$ and $\delta^{15}\text{NGlu}$ determinations (i.e. $<1\%$)?

L 178-179. Avoid repetition of methods (L 571-574) in the results section. The same for the Discussion section (L 244-245).

L 247-248. Avoid repetition of results (L183-187)

L 326-328. In addition to laboratory experiments, the differential fractionation of $\delta^{15}\text{NAla}$ and $\delta^{15}\text{NGlu}$ was also shown in field studies including zooplankton (e.g. Décima & Landry, 2020) but also in a large range of marine consumers (Viana et al., 2023). Take into account that, even when the apparent isotopic fractionation of these two amino acids is significantly correlated across taxa, the resulting trophic positions by applying group-specific fractionation values may lead to differences in the contribution of the microbial-protzoan food web (Viana et al. 2023). In fact, most of the samples identified as mixotrophs in Fig. 4 are above the regression line (but also a high number of autotrophs!), suggesting a substantial contribution of microbial-protzoan trophic steps in most cases.

L 329 and 334: correct references to Figure 3 (not Fig. 4)

L 334-335. Phagotrophy and osmotrophy cannot be distinguished using only $\delta^{15}\text{NGlu}$ (ref. 18). So, why do you interpret "evidence of phagotrophy" in Fig. 4? Perhaps adding a plot of $\delta^{15}\text{NAla}$ vs. $\delta^{15}\text{NPhe}$ in Fig. 3 would help to better differentiate between pure autotrophs and mixotrophs.

L351-352. Consider estimating microbial-protzoan contributions as in other studies (ref. 76; Fernández-Urruzola et al., 2023) to illustrate the relative increase in food chain length.

L 586 References: revise journal style and provide enough information to identify the source (e.g. refs. 25, 27, 31, 44, 55, 65, 85)

Additional references:

Décima, M. Zooplankton trophic structure and ecosystem productivity. *Mar. Ecol. Prog. Ser.* 692, 23-42 (2022).

<https://doi.org/10.3354/meps14077>

Décima, M. & Landry, M. Resilience of plankton trophic structure to an eddy-stimulated diatom bloom in the North Pacific Subtropical Gyre. *Mar. Ecol. Prog. Ser.* 643, 33-48 (2020). <https://doi.org/10.3354/meps13333>

Fernández-Urruzola, I. et al. Trophic ecology of midwater zooplankton along a productivity gradient in the Southeast Pacific. *Front. Mar. Sci.* 10, 1057502 (2023). <https://doi.org/10.3389/fmars.2023.1057502>

Legendre, L. & Rassoulzadegan, F. Plankton and nutrient dynamics in marine waters. *Ophelia* 41, 153-172 (1995).

<https://doi.org/10.1080/00785236.1995.10422042>

Viana, I. G., García-Seoane, R. & Bode, A. The missing trophic link: Contribution of the microbial loop to the estimation of the trophic position of pelagic consumers. *Limnol. Oceanogr.* 68, 2587-2602 (2023). <https://doi.org/10.1002/lno.12445>

Version 1:

Reviewer comments:

Reviewer #1

(Remarks to the Author)

Review of Version 2

I have looked over the rebuttal document and make the following observations:

In general the authors have made good replies. However, I am concerned that some of them (especially concerning terminology) have not all been implemented. Thus, there remains the problems around 'mixotrophy' and 'phytoplankton' etc. (see comments on the new document); it is as if who ever wrote the rebuttal did not check what was actually done, or vice versa.

Concerning the response to the use of stable isotopes, whether the amino acid technique is really as robust as is claimed here requires reference to a modelling approach (similar to the Flynn et al. method) that actually track the fate of isotopes. I have to say, however, that I would be extremely surprised if a unique signature appears, not least because of the taxonomic range of mixoplankton and the range of prey items. The least that needs to be done is a rigorous laboratory study; from what I can see, this has not been done. The reason that lab and field methods are alone unsuitable for this test is that they assume a trophic level. In especially highly complex microbial systems, allocation of a trophic level is itself highly questionable.

Indeed, the authors do accept in their rebuttal (and also in the revised Discussion) that the approach needs benchmarking against better data sets.

The lack of good organism identification is a great loss to this study. Being able to assign osmotrophy as osmo' and/or phago is really very important in trophic level structuring. Only phagotrophy would (classically) be identified as a means to alter the trophic level. Osmotrophy would not be considered as a means to shift the TL. One then wonders how osmotrophy operates when one considers that much uptake of DOM may simply represent within-guild take up of leakage products.

I much appreciated the commentary made by Reviewer #2. It reaffirms my previous concerns on the whole use of stable isotopes, that the approach is helpful, but that ultimately (as per the Flynn et al (2018) conclusion), the method is confounded by the fact that the TL concept itself is problematic. That osmotrophy and phagotrophy may give the same signal is also highly problematic.

For the new document, I make the following observations:

Graphic abstract; top right. These images give an impression that organism identification has been performed, and/or that certain species dominate. That is not correct, as far as I can see. AND these are not all 'phytoplankton'. The middle one is almost certainly a mixoplankton, for example.

Graphic abstract; lower part. The presence here of the word 'phytoplankton' is inappropriate.

L24 This is not correct; they may be mixoplankton.

L25 I suspect there are no 'strict' autotrophs. This term is also ambiguous; does it suggest that they only use autotrophy, or that they must involve autotrophy? (the latter is likely common in mixoplankton, in any case, that they cannot grow solely heterotrophically).

L31 To secure this statement you need identifications and evidence of phagotrophy. I am not convinced you have that. The method (as claimed) only indicates a scale of photo + osmo/phago trophy. Note that these '-trophy' terms are not nouns; they are not identifying organism types.

L45 This is too strong; it is suggested that many can exploit phagotrophy. All these organisms do not 'rely' upon phagotrophy. Indeed, it is likely that very few mixoplankton actually 'rely' on it (exceptions notably being the non-constitutive mixoplankton).

L49 You mean relying on osmotrophy for heterotrophy

L98 This seems too strong here. For example, how does this differ with osmotrophic consumption of material that is leaked when mixoplankton toxins kill a prey and make it lyse? The issue of diel cycles also need to be resolved. Is osmotrophy of DFAA in darkness handled differently in the light? (Yes it is, at least in diatoms!!) How is the isotope ratio affected by the use of DFAA recovered in darkness by phytoplankton that actually leaked those same DFAA in the previous light phase? The whole topic needs to be grounded in lab work first, no?

L178 Cryptophytes are quite likely mixoplankton.

L200 This should read 'mixotrophy', not the noun 'mixotrophs'.

L206 While it may be significant statistically, the scatter even around points that are very robust, is to my mind too great.

L216 This needs to be made explicit as to the quantification. % of what, exactly? Biomass, numeric abundance?

L255 'pure' autotrophy is too strong a term; 'strongly', or 'predominantly' perhaps? The scatter from plots such as Fig.4 to my mind precludes strong terminology.

L263 I would not support such a view. Many mixoplankton feed on bacteria (not sure what TP that would be, but it would not be TP1, for sure), and many others feed on other mixoplankton. This crude TP assumption for mixoplankton cannot be considered as safe.

L287 What evidence is there for this? This also assumes that mixotrophy is to supply C (noting as an aside that your methods track N).

L298 This is not correct. The evidence is not strong enough for that comment. The evidence shows that there are members of these groups that are mixoplanktonic. More profoundly, however, all of these organisms have a mixotrophic capability. Diatoms are well known for it, for example (as you note below).

L371 This appraisal is honest, but ultimately it also leaves us with not being sure of anything. What was needed here was a microscope assessment of species and of the presence of clear evidence of mixoplankton grazing activity (short term incubations?).

L451 The absence of specific information on organism types (genus etc) is very unfortunate. In consequence we cannot compare the plankton composition with membership of the Mixoplankton Data Base :
<https://onlinelibrary.wiley.com/doi/10.1111/jeu.12972?msockid=147dcec09ffd6f8d12f3da399e016edb>

L589 This is ambiguous, whether the signal actually separates phototrophy, osmotrophy and phagotrophy, let alone the trophic signal. Is osmotrophy assuming leak-recovery detected? Is osmotrophy supporting a technically different trophic level?!

L591 Mixoplanktonic activity is quite different to phototrophy + phagotrophy, especially in the context of N-physiology. This is due to the fate of recycled ammonium. I remain concerned on this issue.

L920 (Fig.1) suggest this needs to read as 'photoautotrophy' and 'mixotrophy'. The isotopes are aimed to identify processes, not organism types, so nouns are inappropriate.

L922 (Fig.1) Contrary to the rebuttal, there still seems to be confusion here; all the photoautotrophs will be (photo-osmo) mixotrophic.

L938 (Fig2) This needs to say what the PCA is driven by, what the two dimensions are.

L951 (Fig.3) explain the TP1 .. TP2 lines ('trophic position').

L963 (Fig.4) If this plot is meant to inspire confidence in the approach, I am afraid it rather fails to do so. There are, to my mind, too much scatter here. You could use the signatures as confirmation of what other lines of research may suggest, but alone, I do not think so. Just compare the protozooplankton and the microalgae . Indeed, if you were given any of these data points for the protist and cyanobacteria, could you correctly identify its trophic location? I do not think so.

In summary, I am now much clearer in my own mind as to how this work contributes to plankton science. I think the value of the work is useful rather than important. It flags more what needs to be done, and it also flags the problems in applying this amino acid isotope technique. The work itself is critically hampered by the lack of either supporting laboratory data (which can be done) and/or the lack of rate measurements and explicit taxonomic information from the field (which cannot be rectified).

Reviewer #2

(Remarks to the Author)

The authors have made a great effort in clarifying all the issues raised by the reviewers. Particularly, the definition of the use of the terms autotrophs, heterotrophs, and mixotrophs in relation to end-point trophic position values in the introduction facilitates the understanding of the manuscript and strengthens its contribution. The brief justification of the enhanced tracer properties of d15N CSIA vs. bulk d15N is also welcome. The recalculation of Fig. 4 with Ala and Glu d15N values normalized to Phe (instead of the converse in the first version of this figure) allows for a better understanding of the gap filled by mixotrophs. Indeed the high correlation and the slope close to unity supports an equal contribution of phagotrophy and osmotrophy, on average, through the range between TP= 1 and TP=3.

Finally, I agree that there is still a large uncertainty in TDF values of the different amino acids, particularly for low trophic level organisms, requiring further attention in the future. I praise the efforts made by the authors in trying to estimate trophic steps with the current data. I acknowledge that, despite the general agreement between the two different estimates, the large number of cases with "negative" trophic steps indicate that these calculations need to be improved for low trophic levels. Overall, this is a nice contribution to the field that can be used to inform processes represented in ecosystem modelling and climate change projections.

Reviewer #1 (Remarks to the Author):

Fernández-Carrera et al.

OVERVIEW

This is for the most part a well written work on an interesting ecosystem and tackling a timely and interesting topic. Where it falls down is mainly on two grounds, and I would invite the authors to resolve these issues.

We would like to thank the Reviewer for such a detailed and thorough review. The suggestion to revise the text to clarify the terminology used and the limitations on the application of bulk isotopes has allowed us to explain our proposed approach more clearly.

#1 There is something of a confusion throughout the work with terminologies. In part this reflects the development of the mixoplankton paradigm terminologies themselves. However, this has now settled so it is important to use the most recent forms even though this may outwardly create some tensions with terminologies in some of the cited papers.

Thus:

Phytoplankton – all (cyanobacteria and protist) are potentially mixotrophic via photo+osmo 'troph'. Some can grow solely on osmotrophy (e.g. diatoms in biotechnology in total darkness). In consequence, identifying 'autotrophic' vs 'mixotrophic' plankton does not make sense; all the autotrophic species are mixotrophic (the degree to which they enact mixotrophy is a separate issue).

Mixoplankton – protist only, are potentially mixotrophic via photo+osmo+phago 'troph'. The text confuses activities vs organism descriptors. There is thus a problem with 'Autotrophs' (there is no such group in reality) vs 'autotrophy'.

We agree with the Reviewer's clarification of the current paradigm of mixoplankton, and acknowledge the confusion that was found throughout the text. However, our approach reflects the metabolic strategies of the organisms (see next comment for a more detailed explanation). Thus, plankton relying mostly on autotrophy have a TP close to 1 (autotrophs and base of the food web). Even though they are all capable of osmotrophy, hence mixotrophs, if this osmotrophy is marginal and most of the requirements are covered by autotrophy, this will reflect in their amino acid signatures, and we would call these organisms autotrophs (or dominant autotrophs) for the scope of the manuscript. While plankton combining autotrophy with osmotrophy or osmo-phagotrophy, should deviate to the region around the trophocline of TP1.5, which would be occupied by active mixotrophs coupling heterotrophy with autotrophy. This is why we would still use this mixotrophy/autotrophy (autotroph/mixotroph) dichotomy in the text, although by definition all phytoplankton are potentially mixotrophs. Hence, we would like to reflect the dominant mode

defined by our CSIA AA approach using these terms in the text. We then agree that the text needs clarification, and a clear statement of the terminology is required, so we have made it in the introduction for ease of discussion (text below). However, we are open to further suggestions and modifications of the terminology if this is not fully appropriate.

L127-135: “The organisms will also be named based on their dominant activity identified using our amino acid isotope approach. Therefore, when autotrophy is the dominant mode in seston (TP1), the organisms will be referred to as ‘autotrophs’, whereas those with a TP of 1.5 will be referred to as ‘mixotrophs’, reflecting osmotrophy or phagotrophy in addition to autotrophy. The term mixoplankton will be applied to refer to organisms fulfilling the new paradigm (i.e., coupling osmo-phago-phototrophy), while phytoplankton refers only to photosynthetic cyanobacteria and diatoms. We are aware that the previous terminology could conflict with evidence of widespread osmotrophy (i.e., mixotrophy) rather than autotrophy in microalgae, but this approach is taken for ease of discussion.”

#2 There is a problem with the use of the stable isotopes that needs to be recognised. Before proceeding, I should say that I think this stable isotope amino acid approach is really nice from a technological point of view and the authors are to be congratulated on its deployment. They also do caveat (rather late on) that the conceptual basis for deployment in field work needs confirmation. So, to the problem – the issue is that in tightly recycling systems, such as that being studied here, stable isotopes (explicitly $d^{15}N$) do not provide (actually, cannot provide) a robust prognostic tool. This is from the work of Flynn et al. 2018, in one of those papers that users of the $d^{13}C$ and $d^{15}N$ methods perhaps do not want to acknowledge! Now, it may be that this amino acid isotope fractionation is rather robust against this problem, but the problem at least needs to be recognised. It would also provide a prompt for further work on the method. We really need this, because if correct, if robust, this approach could offer a very important step forward.

*Flynn, K. J., Mitra, A., & Bode, A. (2018). Toward a mechanistic understanding of trophic structure: inferences from simulating stable isotope ratios. *Marine Biology*, 165, 1-13.*

Thank you very much for this important comment. Flynn et al. (2018) state that “Determining the exact TL and SIR over a time course of trophic interactions in nature, as required to rigorously test the relationship between the two, is not possible. The test also needs to be conducted for food webs of different complexity and dynamics.” We agree that the bulk nitrogen isotope approach is complicated by the origin, movement, and transformation of nitrogen in the upper water column (Fry & Quiñones, 1994; Goering et al., 1990; Layman et al., 2012), as well as by potential decoupling of bulk $\delta^{15}N$ signatures given the different N turnover times (life spans) of autotrophs and consumers (Martínez del Rio et al., 2009; Montoya, 2007; Tiselius & Fransson, 2015).

Indeed, amino acid nitrogen stable isotope fractionation solves much of these problems: The strength of CSIA lies in providing information on both TL (or better TP for trophic position) and

N sources from a single organism/sample, which is achieved with a simple comparison of the $\delta^{15}\text{N}$ values of a so-called trophic amino acid like glutamic acid (Glu) or alanine (Ala) and the so-called source amino acid phenylalanine (Phe) (McClelland & Montoya, 2002; Mompeán et al., 2016). While Glu is enriched in ^{15}N by $\sim 8.0\%$ per trophic transfer (Chikaraishi et al., 2009), the $\delta^{15}\text{N}$ of Phe remains nearly unchanged when the amino acid (AA) is transferred through the food web and thus reflects the isotopic composition of the primary producers (N-source measure, Chikaraishi et al., 2010). This approach largely eliminates potential sources of error in TP estimates associated with temporal and physiological decoupling between a consumer and its diet, and has been refined and confirmed in numerous field- and lab-based trophic studies over the last decade (reviewed by Glibert et al., 2019 and Ohkouchi et al., 2017, Loick-Wilde et al. 2019, Weber et al. 2021). Here we applied this approach for the first time to the microplankton compartment in the field and identified that mixotrophy seems to be a systematic trait in certain parts of the Amazon River Plume. In order to acknowledge the CSIA-based improvements compared to bulk stable nitrogen isotope approaches in trophic ecology, we stated the following in the introduction in L74-78:

“The seminal work of McClelland and Montoya (2002)²⁰ demonstrated the potential of nitrogen isotopes in amino acids to track food web dynamics that eliminate much of the problems associated with the traditional bulk nitrogen isotope approaches concerning the origin, movement, and transformation of nitrogen in the upper water column, as well as by potential decoupling of bulk $\delta^{15}\text{N}$ signatures due to the different N turnover times (life spans) of autotrophs and consumers^{21,22}.”

The explanation above may be sufficient for addressing the concerns of the Reviewer. However, we would also like to extend the clarification of the soundness of this approach on its metabolic basis in the following lines, in case it was not clear enough for supporting our point. There are also experimental studies comparing TP estimated with CSIA AA and bulk SIA, supporting the Reviewer’s comment. For example, Bowes and Thorp (2015) made a comparison in laboratory feeding experiments with four trophic levels. The authors suggest that while nutritional condition has an effect on the $\delta^{15}\text{N}$ in individual amino acids (which we would say very likely will reflect in their bulk signature), this effect may be comparable between amino acids, and hence the difference between them is unchanged. And this is the key of this approach, this difference between trophic and source amino acids varies in narrow ranges.

The signature of source amino acids like phenylalanine reflects the source of nitrogen supporting primary producers at the base, and it should be possible to back-calculate the base using:

$$\delta^{15}\text{N}_{\text{base}} = \delta^{15}\text{N}_{\text{phe}} - 0.4 \times (\text{TP} - 1)$$

$\delta^{15}\text{N}_{\text{phe}}$ will hence change if strong recycling exists. However, amino acid metabolism is highly regulated due to their biological relevance, and while it is unclear what are the exact fractionation values associated to each enzyme involved in deamination or transamination reactions, evidence

suggest that there is a constraint variation in the final fractionation of the overall pool of each individual amino acid. Hence, the difference between the whole pool of the different types of amino acids varies in a narrow range, e.g., the difference between phenylalanine and glutamic acid varies between 3 and 4%. A recent revision by Ohkouchi (2023) explains this and the basis for different applications in more detail, further supporting the explanation by O'Connell (2017) that we also cited in the main text.

We acknowledge that the application of this method is very recent in the field, from the seminal work of McClelland and Montoya (2002), which stimulated further research in a few groups in Japan, Europe and the US, widening the application of isotopes in amino acids beyond trophic ecology (e.g., references in MS). Still, the community has not yet focused on mixoplankton. This is why we say the conceptual basis needs confirmation. There are no available data to validate what seems to be a strong relationship pointing to a "gap" in Figure 4. This gap would be occupied by mixoplankton.

The conceptual basis builds on the high regulation and consistency in apparent fractionation found in lower trophic levels to date. This is mainly one of the reasons why we explored the apparent isotopic fractionation of source amino acid phenylalanine relative to trophic amino acids alanine and glutamic acid (+glutamate) in literature values, looking for a space where mixoplankton should fit in terms of isotopic signatures. This apparent isotopic fractionation is formally more correct than the difference between absolute signatures of the amino acids, though in general the values are very close with differences in the decimal figures. We selected only literature plankton data for testing this relation. Among the reasons, because whole samples are measured, it is relevant to our own samples, and there is evidence supporting that in upper trophic levels, the trophic discrimination factor (TDF) of trophic amino acids seems to decrease with each trophic step, which makes the definition of TP for grouping the literature values a bit more challenging. Again, Ohkouchi (2023) goes through studies accounting this latter issue on the TDFs. Also, it should be noted that most microalgae data in literature are derived from cultured strains, while zooplankton come from field samples. Therefore, we mixed culture and field values in this revision and still found a significant relationship, which illustrates the tight regulation of the amino acids pool. This finding supports the potential of this approach for detecting mixotrophy/mixoplankton in the field.

Alanine and glutamic acid were chosen because, as explained in the main text, are relevant for our functional groups, especially in relation to phagotrophic vs osmotrophic protists. As we said, the apparent isotopic fractionation should vary in a narrow range and be consistent with their TP because of the high metabolic regulation of the pool of amino acids, and the TDFs should be similar in plankton trophic levels because of similarities in their overall N metabolism. It is clearly shown in our literature revision in Fig 4 (please see the explanation for changes in the comment related to Fig 4 below). Even though in all databases we found "phytoplankton", we have relabeled the dots as microalgae, following the Reviewer's suggestion. In fact, a quick revision of one outlier (inside a blue circle below) suggests that some mixoplankton were cultivated as

osmotrophs, but used as “phytoplankton” primary producers (i.e., autotrophs) in previous studies. This outlier belongs to a *Chlorella* spp used in feeding experiments for assessing CSIA on its consumers. It was grown in a medium supplemented with amino acids (Lee et al. 2020). Of course, we cannot confirm it was growing as an osmo-phago-mixotroph likely grazing bacteria (maybe present in the semi continuous setting used here), but the apparent isotopic fractionation of both amino acids suggests so, as the TP Glu/Phe gives a value of 1.6 and there is an obvious enrichment in $\epsilon_{\text{Ala/Phe}}$.

In our literature analysis, all microalgae were assumed TP1, as all literature references identified them as primary producers, the only apparent deviation from them being TP1 is the *Chlorella* mentioned above. Mesozooplankton from natural samples were labeled according to their estimated TP on Glu and Phe, hence herbivore = TP2, omnivore = TP 2.5, carnivore = TP 3. The phagotrophs are data from cultures of strict phagotrophs used by Gutierrez-Rodriguez et al. (2014) and Décima et al. (2017) for assessing the role of alanine as integrator of phagotrophy, activity that according to the authors does not reflect on the signature of glutamic acid.

Given the statistically significant relationship with an R^2 of 0.82, we hypothesize that the area where our ARP TP1.5 seston distributes is the space that mixoplankton would occupy due to their combined feeding strategies, similar to what happens with omnivores. But, as we said, no evidence in literature yet, as this has not been explored by the community. Though, we recently submitted a proposal for testing this ourselves in cultures, and hope to be able to fill the gap soon.

- Bowes, R. E., and J. H. Thorp. 2015. Consequences of employing amino acid vs. bulk-tissue, stable isotope analysis: a laboratory trophic position experiment. *Ecosphere* 6. doi:10.1890/es14-00423.1
- Chikaraishi, Y., Ogawa, N. O., Kashiyama, Y., Takano, Y., Suga, H., Tomitani, A., et al. (2009). Determination of aquatic food-web structure based on compound-specific nitrogen isotopic composition of amino acids. *Limnology and Oceanography: Methods*, 7(11), 740–750. <https://doi.org/10.4319/lom.2009.7.740>
- Chikaraishi, Y., Ogawa, N. O., & Ohkouchi, N. (2010). Further evaluation of the trophic level estimation based on nitrogen isotopic composition of amino acids. In N. Ohkouchi, I. Tayasu, & K. Koba (Eds.), *Earth, life, and isotopes* (Vol. 415, pp. 37–51). Kyoto University Press.
- Décima, M., M. R. Landry, C. J. Bradley, and M. L. Fogel. 2017. Alanine $\delta^{15}\text{N}$ trophic fractionation in heterotrophic protists. *Limnol. Oceanogr.* 62: 2308–2322. doi:<https://doi.org/10.1002/lno.10567>
- Flynn, K. J., Mitra, A., & Bode, A. (2018). Toward a mechanistic understanding of trophic structure: inferences from simulating stable isotope ratios. *Marine Biology*, 165, 1-13.
- Fry, B., & Quiñones, R. B. (1994). Biomass spectra and stable isotope indicators of trophic level in zooplankton of the northwest Atlantic. *Marine Ecology Progress Series*, 112, 201–204. <https://doi.org/10.3354/meps112201>
- Glibert, P. M., Middelburg, J. J., McClelland, J. W., & Jake Vander Zanden, M. (2019). Stable isotope tracers: Enriching our perspectives and questions on sources, fates, rates, and pathways of major elements in aquatic systems. *Limnology & Oceanography*, 64(3), 950–981. <https://doi.org/10.1002/lno.11087>
- Goering, J., Alexander, V., & Haubenstock, N. (1990). Seasonal variability of stable carbon and nitrogen isotopic ratios of organisms in a North Pacific bay. *Estuarine, Coastal and Shelf Science*, 30, 239–260. [https://doi.org/10.1016/0272-7714\(90\)90050-2](https://doi.org/10.1016/0272-7714(90)90050-2)
- Gutierrez-Rodriguez, A., M. Decima, B. N. Popp, and M. R. Landry. 2014. Isotopic invisibility of protozoan trophic steps in marine food webs. *Limnol. Oceanogr.* 59: 1590–1598. doi:10.4319/lo.2014.59.5.1590
- Layman, C. A., Araujo, M. S., Boucek, R., Hammerschlag-Peyer, C. M., Harrison, E., Jud, Z. R., et al. (2012). Applying stable isotopes to examine food-web structure: An overview of analytical tools. *Biological Reviews*, 87(3), 545–562. <https://doi.org/10.1111/j.1469-185x.2011.00208.x>
- Lee, M.-C., H. Choi, J. C. Park, and others. 2020. A comparative study of food selectivity of the benthic copepod *Tigriopus japonicus* and the pelagic copepod *Paracyclops nana*: A genome-wide identification of fatty acid conversion genes and nitrogen isotope investigation. *Aquaculture* 521: 734930. doi:<https://doi.org/10.1016/j.aquaculture.2020.734930>
- Loick-Wilde, N., Fernández-Urruzola, I., Eglite, E., Liskow, I., Nausch, M., Schulz-Bull, D., et al. (2019). Stratification, nitrogen fixation, and cyanobacterial bloom stage regulate the planktonic food web structure. *Global Change Biology*, 25(3), 794–810. <https://doi.org/10.1111/gcb.14546>
- Martínez del Río, C., Wolf, N., Carleton, S. A., & Gannes, L. Z. (2009). Isotopic ecology ten years after a call for more laboratory experiments. *Biological Reviews*, 84(1), 91–111. <https://doi.org/10.1111/j.1469-185X.2008.00064.x>
- McClelland, J. W., & Montoya, J. P. (2002). Trophic relationships and the nitrogen isotopic composition of amino acids in plankton. *Ecology*, 83, 2173–2180. [https://doi.org/10.1890/0012-9658\(2002\)083\[2173:TRATNI\]2.0.CO;2](https://doi.org/10.1890/0012-9658(2002)083[2173:TRATNI]2.0.CO;2)

- Mompeán, C., Bode, A., Gier, E., & McCarthy, M. D. (2016). Bulk vs. amino acid stable N isotope estimations of metabolic status and contributions of nitrogen fixation to size-fractionated zooplankton biomass in the subtropical N Atlantic. *Deep Sea Research Part I: Oceanographic Research Papers*, 114, 137–148. <https://doi.org/10.1016/j.dsr.2016.05.005>
- Montoya, J. P. (2007). Natural abundance of ^{15}N in marine planktonic ecosystems. In R. Michener, & K. Lajtha (Eds.), *Stable isotopes in ecology and environmental science* (2nd ed., pp. 176–201). Blackwell.
- O’Connell, T. C. 2017. ‘Trophic’ and ‘source’ amino acids in trophic estimation: a likely metabolic explanation. *Oecologia* 184: 317–326. doi:10.1007/s00442-017-3881-9
- Ohkouchi, N., Chikaraishi, Y., Close, H. G., Fry, B., Larsen, T., Madigan, D. J., et al. (2017). Advances in the application of amino acid nitrogen isotopic analysis in ecological and biogeochemical studies. *Organic Geochemistry*, 113, 150–174. <https://doi.org/10.1016/j.orggeochem.2017.07.009>
- Ohkouchi, N. 2023. A new era of isotope ecology: Nitrogen isotope ratio of amino acids as an approach for unraveling modern and ancient food web. *Proc. Japan Acad. Ser. B* 99: 131–154. doi:10.2183/pjab.99.009
- Tiselius, P., & Fransson, K. (2015). Daily changes in $\delta^{15}\text{N}$ and $\delta^{13}\text{C}$ stable isotopes in copepods: Equilibrium dynamics and variations of trophic level in the field. *Journal of Plankton Research*, 38(3), 751–761. <https://doi.org/10.1093/plankt/fbv048>
- Weber, S. C., Loick-Wilde, N., Montoya, J. P., Bach, M., Doan-Nhu, H., Subramaniam, A., et al. (2021). Environmental regulation of the nitrogen supply, mean trophic position, and trophic enrichment of mesozooplankton in the Mekong River plume and southern South China Sea. *Journal of Geophysical Research: Oceans*, 126, e2020JC017110. <https://doi.org/10.1029/2020JC017110>

SPECIFIC COMMENTS

Graphic abstract – as all autotrophs are osmotrophs; this needs to be labelled with activities and not with organism labels

Following the suggestion by the reviewer we have relabeled the symbols using activities, as it is more appropriate and makes the graphical abstract consistent. Now it reads: autotrophy, phagotrophy, osmotrophy and diazotrophy in the legend.

Succession of Functional Types in April/May

Potential impact of Phagotrophy and Osmotrophy on trophic amino acids

L21 This has been known for >50yrs, but rarely labelled as 'mixotrophy'.

L24 this is the wrong way around and misleading .. ancestral mixoplankton were intermediates between phagotrophs and (then) protist phytoplankton. See Mitra et al. (2024) Trait trade-offs in phagotrophic microalgae: the mixoplankton conundrum, European Journal of Phycology, 59, 51-70. <https://doi.org/10.1080/09670262.2023.2216259> .

L26 the problem with identifying photo-osmotrophy is as much to do with separating protist from bacterial osmotrophy - this has been an issue reported since the 1980s. Now we recognise that bacterivory is displayed by many formerly termed 'phytoplankton' (now 'mixoplankton') the analyses and interpretations are more nuanced.

The three comments above are related to the Abstract, and are related to issue #1 as well as other comments found in other sections of the manuscript below, that required some fundamental changes in the text. Hence, for addressing all them, we have edited the whole abstract. The new text reads as follows:

L21-35: “Phytoplankton, namely diatoms and cyanobacteria, are mixotrophs, combining photoautotrophy and the uptake of dissolved organic matter (osmotrophy). All other unicellular photosynthetic eukaryotes, except diatoms, are potentially phagotrophs and are currently classified as mixoplankton. This functional group occupies a unique position between strict autotrophs and heterotrophs in planktonic food webs, producing a greater carbon stock and offering higher-quality food for metazoans than phytoplankton do. However, field studies remain challenging due to the difficulty of distinguishing their sole activity in seston containing a mixture of all functional groups. During our April/May 2018 and 2021 cruises in the Amazon River plume, we examined seston using compound-specific stable isotope analysis of amino acids (CSIA AA) to determine its trophic dynamics. Based on the comparison of glutamic acid and alanine with phenylalanine, we found a dominance of mixoplankton in the Outer Plume Margin, a region of mature waters around 27 days old. Osmo-phago-phototrophy appears to be the optimal growth strategy in these heterogeneous margins as part of the succession of microalgae functional diversity along the plume. Our study underscores the urgent need to study mixoplankton in situ within a multidisciplinary framework and paves the way for a novel application of CSIA AA in field research.”

L49 why should it decrease that efficiency? That makes no sense and is contrary to evidence. (see Mitra et al. 2024 EurJPhycol)

We were referring to the classical concept of food chains, where every new trophic steps decreases the energy channeled up the chain. However, given the limit of words, and that our study focuses on showing the potential of the tool for detecting a dominance of mixoplankton in field samples, not on the potential transfer to other trophic levels, we have removed this sentence from the introduction. The main reason is that it was clearly not well explained, both reviewers found it

inaccurate, and would require a longer explanation that does not contribute to the actual scope of the paper mentioned above.

L67 Again, where is the evidence? This is a very old paper you cite here; it perhaps makes little sense set against recent considerations.

Agreed, we have removed this sentence from the introduction.

L76 While the method has uses for sure, the use of any stable isotope approach, especially in a tightly coupled mature system must be recognised as potentially highly problematic.

Following the explanations given above and the recommendation of the reviewer in L224, we have modified this part of the introduction to briefly explain why the CSIA AA can be reliably used even in tightly coupled mature system, circumventing the issues associated to bulk SIA. The new text reads as follows:

L74-98: “The seminal work of McClelland and Montoya (2002)²⁰ demonstrated the potential of nitrogen isotopes in amino acids to track food web dynamics that eliminate much of the problems associated with the traditional bulk nitrogen isotope approaches concerning the origin, movement, and transformation of nitrogen in the upper water column, as well as by potential decoupling of bulk $\delta^{15}\text{N}$ signatures due to the different N turnover times (life spans) of autotrophs and consumers^{21,22}. They showed that trophic and source amino acids exhibit distinct isotopic behaviors: trophic amino acids, such as glutamic acid or alanine, become enriched in $\delta^{15}\text{N}$ due to metabolic processing with each trophic transfer, while source amino acids, such as phenylalanine, retain their isotopic signatures from primary producers^{19,23}. Building on this foundation, subsequent studies, notably by Chikaraishi and coworkers, formalized the first equations that allow precise trophic position calculations²⁴, making CSIA AA a widely accepted approach in marine ecology. The potential of CSIA AA to address this complexity is evident in its ability to track both nitrogen assimilation pathways and metabolic processes associated with mixotrophy. Gutierrez-Rodriguez et al. (2014)¹⁷ and Décima et al. (2017)¹⁸ used cultures of strict phagotrophs to show that alanine reflects protist grazer activity while glutamic acid does not, highlighting the use of alanine as an integrator of additional protist grazer trophic steps in the planktonic food web, which is not obvious when glutamic acid is used to estimate trophic hierarchies. For instance, cultures of *Oxyrrhis marina* grazing on *Dunaliella tertiolecta* showed a similar difference between glutamic acid and phenylalanine in both phagotroph and prey (3.3 to 5.2‰ and 4.1 to 5.1‰, respectively), while alanine had a larger difference in the phagotroph (5.5 to 8‰)¹⁷. In contrast, Yamaguchi et al. (2017)¹⁶ showed that osmotrophic uptake of amino acids by prokaryotic and eukaryotic microbes in culture leaves an imprint on glutamic acid similar to that of herbivores, reflecting the heterotrophic metabolic effect on this amino acid isotopes. Hence, the combination of glutamic acid and alanine isotopic signatures has the potential to unequivocally detect mixoplankton activity in natural samples.”

L94 This paragraph looks like it was written by someone else; it is dominated by 'phytoplankton' which is inconsistent with the terminology used in the previous paragraphs. The problem is then proliferated throughout the rest of the text.

Agreed. In initial versions of the MS, we used “pigmented plankton” instead of phytoplankton, because our classification is based on pigment markers, and seemed more appropriate. However, it seemed to be confusing for some colleagues, and it is our perception that many researchers working with “phytoplankton” are not fully aware of the new paradigm, and still apply this term when referring to photosynthetic prokaryotes and eukaryotes, even if they are mixoplankton. Following the suggestion of the Reviewer below (L136) we have replaced “phytoplankton” by “microalgae” in the main text and the supplements, and added an explanation in the closing paragraph of the intro as follows:

L125-127: “For the purpose of this study, we will refer to the pigmented community containing both phytoplankton and mixoplankton ‘microalgae’.”

L136 Likewise this text is dominated by 'phytoplankton' .. perhaps 'microalgae' would be a better term as a catch-all?

Agreed (see above).

L160 onwards: It is really important to evidence that this methodology works in these complex systems. I am not disputing that mixotrophy is important in these waters, but the most that can perhaps be evidenced using this method just now is that 'the data are consistent with' mixotrophy?

This comment refers to the “Amino acids nitrogen isotopes and trophic mode” in the Results section. We hope our explanation above supports the feasibility of using stable isotopes of amino acids for detecting mixotrophy and mixoplankton in mature and complex systems, circumventing this way the issues associated to bulk isotopes.

Reflecting on this comment and that in L342, we have modified the discussion in L382-384 as follows to acknowledge the uncertainties related to the lack of literature data on mixoplankton in our approach:

“Nevertheless, our study demonstrates the potential usefulness of combining the nitrogen stable isotopes from different amino acids in seston to detect mixoplankton activity in the field, provided that samples primarily reflect the signature of microalgae.”

L224 OK this provides the required evidence; this should be stated clearly in the Introduction, to justify the whole approach!!

As stated above, we have explained the basis of the CSIA AA approach in L74-98 in the introduction for supporting the application of the approach (revised introduction text above).

L258 Note that the model of Ward & Follows has no mechanistic basis at all. You may wish to check that model structure very carefully before citing it to evidence this point.

Cf Mitra et al. (2016) Defining planktonic protist functional groups on mechanisms for energy and nutrient acquisition; incorporation of diverse mixotrophic strategies. Protist 167, 106–120. <http://dx.doi.org/10.1016/j.protis.2016.01.003>

Thanks for explaining the limitations of the model we cited (we were not aware of this issue), and also for providing a useful reference that we could consult. We have modified the text accordingly as follows:

L286-289: “Moreover, mixotrophy dominance was observed at stations with higher chlorophyll *a* relative to similar locations ($> 0.171 \mu\text{g L}^{-1}$), likely reflecting the production of a larger carbon stock by mixoplankton relative to phytoplankton suggested by Mitra et al. (2016)⁴⁸ ...”

L261 evidenced by what? (the isotope signature, presumably?)

Indeed. We have modified the text for stating that in our stable isotope results, there seems to be a dominance of mixotrophy in stations where O₂ is at saturation and waters columns are less stable.

L292-294: “Interestingly, our stable isotopes approach shows a dominance of mixotrophy at stations 6.04 and 24.03 during 2018, with waters close to oxygen saturation and a relatively less stable water column (Brunt Väisälä $N^2 \leq 0.006 \text{ s}^{-2}$, Table S1).”

L272 I think the origin of this is Flynn & Berry LS (1999) The loss of organic nitrogen during marine primary production may be overestimated significantly when estimated using 15N substrates. Proceedings of the Royal Society Lond. B 266; 641-647.

Thanks for pointing to this mis-citation in the Discussion. We have amended the reference using the appropriate source.

L283 Again there is the problem with the 'phytoplankton' word.

Changed.

L316 I suggest you cross reference to the Mixoplankton Database –

Mitra et al. (2023) The Mixoplankton Database – diversity of photo-phago-trophic plankton in form, function and distribution across the global ocean. Journal of Eukaryote Microbiology 70 <https://doi.org/10.1111/jeu.12972>

If you have species that you can evidence as being mixoplankton that are absent from the database, it would be useful to identify those species.

Unfortunately, we cannot identify species with our chemotaxonomy based on pigment markers. ChemTax allows us to define what is called by users as functional groups, based on their specific pigment composition. Several species share the same pigment markers within these functional groups. For defining the initial pigment ratios used in the matrix factorization, we selected relevant species from literature as mentioned in the caption of Table 1. But it is not possible to define species with our bulk measurements of pigments and seston.

We have modified slightly the paragraph for explaining this clearer:

L345-346: “These functional groups of unicellular eukaryotes are known to contained mixoplankton species, coupling osmo-phago-phototrophy^{1, 55}.”

L328 this is a very important issue and assumption. Given the problems of isotope discrimination through the food web it is all the more important. It has been explored via models and shown that isotopes cannot be used reliably in this fashion (Flynn et al. 2018).

We understand this comment follows the line of the #2 major issue raised by the Reviewer above. And we hope that our previous explanations clarified the reliability of CSIA AA for tracking the activity of phagotrophs and osmotrophs, as the isotopic signature of the amino acids is a result of the metabolic pathways, and was/is used in this fashion in many different studies and systems.

L338 This is really confusing. To clarify, all phytoplankton are potentially mixotrophic via phot+osmotrophy. Mixoplankton can additionally use phagotrophy. So 'all mixotrophs' using phagotrophy strictly means that you are talking only about mixoplankton. Is that what you mean? And if so, why not use 'mixoplankton'?

We agree that this part of the discussion focusing on the epsilons was really confusing. We have edited and reorder the text for readability, avoiding the repetition of osmotrophy in $\epsilon_{\text{Phe/Glu}}$ of the previous version before and after the line in the question raised by the Reviewer, which was obviously making the text more confusing. We hope that, in combination with the explanation of terminology in the introduction, the text would be easier to understand. The new paragraph reads as follows:

L367-375: “This combined osmo-autotrophy will result in the TP1.5 found in our mixotrophic samples along the Amazon River plume (Fig 3), with $\epsilon_{\text{Glu/Phe}}$ values consistent with the uptake of ambient amino acids (osmotrophy). However, the values of $\epsilon_{\text{Ala/Phe}}$ are also consistent with a contribution of phagotrophy in most mixotrophic samples, as Ala is more enriched than Phe in relation to organisms in TP1.0 (Fig 4). This evidence suggests that prasinophytes, haptophytes, dinoflagellates, and/or cryptophytes actively grazed on other microorganisms to some extent. Hence, it seems that all our mixotrophic samples along the Amazon plume contained organisms combining phagotrophy and osmotrophy as feeding strategies, that is, mixoplankton.”

L343 I suggest it indicates the potential usefulness only - we know that bulk d15N data are not sufficiently robust to direct this conclusion, though this amino acid fractionation may do the trick. (hopefully!)

Agreed. As explained above in the comment for L160 onwards, we have modified the text in L382-384: “Nevertheless, our study demonstrates the potential usefulness of combining the nitrogen stable isotopes from different amino acids in seston to detect mixoplankton activity in the field, provided that samples primarily reflect the signature of microalgae.”

Methods – nicely documented; only a few minor typos.

The comment is highly appreciated, we aimed to provide as much information as possible for making the methods clear and understandable.

Fig.3 I suggest that this is relabelled as activities and not organisms. Thus 'autotrophs' is 'autotrophy' etc. The reason is that all of the 'photoautotrophs' are mixotrophs (either phytoplankton as photo+osmo, or mixoplankton as photo+osmo+phago).

We have relabeled the figure using the numerical TPs of the organisms, following the general practice in this kind of plots, and modified the caption accordingly.

Figure 3. $\delta^{15}\text{N}_{\text{Phe}}$ (phenylalanine) vs $\delta^{15}\text{N}_{\text{Glu}}$ (glutamic acid + glutamate) of the seston samples collected along the Amazon River plume expressed in delta notation ($\delta^{15}\text{N}$ ‰, relative to atmospheric N_2). In 2018, total particles (combined square and triangle) were collected, while in 2021, seston was separated into two size fractions (triangles for $>3 \mu\text{m}$, squares for $0.2\text{--}3 \mu\text{m}$). The dashed lines represent the trophoclines of dominant autotrophy (TP1.0) and herbivory (TP2.0). The dotted lines delimit the trophic space defined for mixotrophs between TP1.4 and TP1.6. Colors represent the different habitats defined by Pham et al (2024) ordered by apparent age: Riverine Input (RI, gray), Young Plume Core (YPC, red), Outer Plume Margin (OPM, purple), Western Plume Margin (WPM, yellow), modified Oceanic Water (MOW, cyan) and Oceanic Water (OSW, blue).

Fig.4 similar comment to that in Fig.3. Where is 'osmotroph'? And I suggest that these 'herbivore', 'omnivore' and 'carnivore' labels need revisiting. Is feeding on a mixoplankton herbivory or omnivory?! Is a 'phagotroph' as labelled here only a protist zooplankton? Do these terms help at

all?! What is an ARP autotroph (all of which will be photo+osmo trophic) vs a 'ARP mixotroph'? (noting also that 'ARP' – Amazon river plume')

We have reviewed the labels according to their TP, instead of the organisms that would represent that TP for easing the discussion. In addition, we have estimated the apparent isotopic fractionation of Glu/Ala relative to Phe, instead of the previous calculation of Phe to Glu/Ala, because after carefully reading all the comments by both Reviewers we realized it was confusing, and less intuitive.

This new version is more consistent with the expected enrichment of trophic amino acids with every trophic step, so larger positive numbers now represent a larger enrichment of Glu/Ala relative to Phe. We have also added the data from strict phagotrophic protists provided in ref 17 to complement those of ref 18, as these are the only values of CSIA AA related to strict phagotrophs in literature.

As a result of this revision, we have modified the Methods for a new calculation of the epsilons in L580-586, and the linear regression equation in the results section (L205) and the figure below.

Figure 4. Apparent isotope fractionation of the two trophic amino acids (glutamic acid + glutamate, Glu; alanine, Ala) relative to the source amino acid (phenylalanine, Phe). Literature data for TP1 (autotrophic microalgae, dark green circles), TP2 (herbivorous metazooplankton, yellow squares), TP2.5 (omnivorous metazooplankton, orange squares) and TP3 (carnivorous metazooplankton, red squares) as well as strict unicellular eukaryotic phagotrophs in cultures in literature (Refs 17, 18, light green diamonds) are represented along with the TP1.0 (dark purple inverted triangles) and TP1.5 (light purple asterisks) seston collected along the Amazon River plume in 2018 and 2021.

Fig.5 Aside for the typographic error of 'Autroptroph', again there is this problem of autotrophy vs mixotrophy.

Apologies for the error. The typo was corrected. We have relabeled referring to the Dominant group defining the numerical TP.

Figure 5. Classification tree based on C5.0 showing the most likely predictors of the dominant trophic mode of seston and the thresholds defined by the model for each of them. Note the model misclassified one sample (17.17 mixotroph was classified as autotroph, accounting for a 2.2 % error). The stations classified in each branch are shown below the tree colored by the different habitats defined by Pham et al (2024) ordered by apparent age: Riverine Input (RI, gray), Young Plume Core (YPC, red), Outer Plume Margin (OPM, purple), Western Plume Margin (WPM, yellow), modified Oceanic Water (MOW, cyan) and Oceanic Water (OSW, blue).

Reviewer #2 (Remarks to the Author):

This manuscript proposes a novel approach to uncover the relevance of mixotrophic feeding within the marine microplanktonic community. By comparing the composition of stable nitrogen isotopes in amino acids the authors are able to identify the dominance of mixotrophic organisms (i.e. trophic position = 1.5) in seston samples collected in habitats of different age along the Amazon River plume. The dominance of mixotrophy was further related to the succession of phytoplankton communities along the plume. The authors conclude that mixotrophy was a mechanism to support plankton growth in aged waters, where nutrient and oxygen concentrations were relatively low. The study was well designed and the manuscript contains most of the required information to support the interpretation. The proposed approach may stimulate further research to unveil the role of mixotrophy in marine ecosystems with major implications not only for our knowledge of nutrient cycling and carbon sequestration but also for the understanding of functional diversity in marine plankton. However, before final publication the authors must consider the following issues:

We would like to thank the Reviewer for a thorough review and suggestions regarding the percentage of microbial steps. Rereading relevant literature references and reflecting on this calculation in our samples has helped us to better explain the applicability of the CSIA AA-based approach to mixoplankton.

L 49 “Transfer of energy from primary producers....” = transfer efficiency. Consider the relationship between transfer efficiency, food chain length and productivity. While there seems to be an universal inverse relationship between food chain length and system productivity (e.g. Legendre and Rassoulzadegan, 1985) transfer efficiency may be also influenced by multiple other factors (Décima, 2022). Mixotrophy may be well one of these factors as indicated in the following text (L 61-62).

We agree with the Reviewer that this sentence in the introduction is oversimplified, and relates to a vision of trophic ecology limited to food chains. This is why we started by “From the perspective of classical trophic ecology...”. However, considering that both Reviewers found it inaccurate, and that our study focuses on showing the potential of the tool for detecting a dominance of mixoplankton in field samples, not on the potential transfer to other trophic levels, we have removed this sentence from the introduction.

L 129 cite Benson & Krause (1984) with number as for other references

Reference format corrected.

L 132-135 Inorganic nutrient concentrations are indicated but no reference or data are provided.

We missed to reference again Table S1 here, thanks for pointing this out. We have also added “at surface” to point that only inorganic nutrients at the same depth as our samples are provided.

The new paragraph reads:

L163-167: “The significant nutrient load carried by the Amazon and Pará Rivers is evident in the elevated concentrations at surface of nitrate and silicate found in the brackish estuary (RI) compared to other habitats, with a sharp decrease at stations on the shelf (YPC), although values remained relatively high at around 2 μM (Table S1).”

L 136-139. Describe the communities in the same order as in Fig. 2: community 1, 2, 3, and 4). Avoid unnecessary repetition of methods (L 140-141).

The pigmented communities were described in the Results in the same order as they were labeled in Fig 2 from the river mouth to the ocean end. However, we realize that the arrangement of the figure, which may be intuitive for us because the plume spreads from south to north, was in fact really confusing for other readers, because the communities were labeled from the bottom to the top (community 1 at the bottom and 4 at the top). We appreciate the issue raised by the Reviewer and have rearranged figure 2 for a reading logical order from top to bottom in order to avoid confusion.

L 159. Some information about the composition, abundance and/or biomass of the non-pigmented plankton (heterotrophic protozoa, small metazoa) would be appropriate to further characterize the trophic structure of the analyzed seston (if available).

Indeed! We appreciate the suggestion by the Reviewer and strongly agree with them, as it would have been extremely relevant to gather this information in the scope of our findings.

Unfortunately, we only have samples for assessing the pigmented plankton community via HPLC and ChemTax. Even though we have images of small metazoan in the stations imaged by the PlanktoScope, our sampling was purely qualitative and combined samples from the ship's flow through system, niskin bottles or hand-net tows at surface. We were testing the instrument for the first time ever as a potential complement to HPLC, with no plans for complementing our own dataset along the Amazon River Plume for publication. But realized while treating data that it was a useful validation tool for ChemTax. We will definitely include the sampling, identification and enumeration of non-pigmented plankton in the design of future cruises.

L 164 and Fig. 3 How were the limits for mixotrophy (dotted lines) determined? Do they represent the error in $d15N_{Ala}$ and $d15N_{Glu}$ determinations (i.e. <1%)?

We set the middle trophocline for osmo-mixotrophy at TP1.5, mirroring the typical midpoint for omnivory at TP2.5. Taking into account the usual error of 1‰ in determining $\delta^{15}N$ in amino acids, already mentioned by the reviewer, which can result in an error of ± 0.2 TP units, and considering the range of TPs in our dataset is from 0.8 to 1.6, we set the upper limit of dominant autotrophy at values below 1.4. Therefore, the range of TPs representing a balanced contribution of osmotrophy and phototrophy to meet the organisms' requirements was set between 1.4 and 1.6 around a canonical midpoint of TP1.5 (Table S1). We had explained this briefly in the Methods and in the caption of Fig 3, as well as in the figure itself.

L578-579, methods: “The range of TP representing dominance autotrophy in seston was set below a TP of 1.4, while the TP of mixotrophs, relying on osmotrophy, was between 1.4 and 1.6.”

Figure 3. $\delta^{15}\text{N}_{\text{Phe}}$ (phenylalanine) vs $\delta^{15}\text{N}_{\text{Glu}}$ (glutamic acid + glutamate) of the seston samples collected along the Amazon River plume expressed in delta notation ($\delta^{15}\text{N}$ ‰, relative to atmospheric N_2). In 2018, total particles (combined square and triangle) were collected, while in 2021, seston was separated into two size fractions (triangles for $>3 \mu\text{m}$, squares for $0.2\text{--}3 \mu\text{m}$). The dashed lines represent the trophoclines of dominant autotrophy (TP1.0) and herbivory (TP2.0). **The dotted lines delimit the trophic space defined for mixotrophs between TP1.4 and TP1.6.** Colors represent the different habitats defined by Pham et al (2024) ordered by apparent age: Riverine Input (RI, gray), Young Plume Core (YPC, red), Outer Plume Margin (OPM, purple), Western Plume Margin (WPM, yellow), modified Oceanic Water (MOW, cyan) and Oceanic Water (OSW, blue).

L 178-179. Avoid repetition of methods (L 571-574) in the results section. The same for the Discussion section (L 244-245).

Apologies for the repetition in the discussion. We have decided to leave the reminder of the machine learning algorithm here in the Result section, because it is very likely that most readers are not aware of its application and may not read the methods before the results. But it was removed from the lines in the Discussion.

L 247-248. Avoid repetition of results (L183-187)

We appreciate the comment by the Reviewer, but we would like to keep the enumeration of predictors in this part of the discussion as a reminder, which also leads the discussion in the following sentences. We have removed the repetition of methods as suggested in the previous comment.

L 326-328. In addition to laboratory experiments, the differential fractionation of $\delta^{15}\text{NAla}$ and $\delta^{15}\text{NGlu}$ was also shown in field studies including zooplankton (e.g. Décima & Landry, 2020) but also in a large range of marine consumers (Viana et al., 2023). Take into account that, even when the apparent isotopic fractionation of these two amino acids is significantly correlated across taxa, the resulting trophic positions by applying group-specific fractionation values may lead to differences in the contribution of the microbial-protozoan food web (Viana et al. 2023). In fact, most of the samples identified as mixotrophs in Fig. 4 are above the regression line (but also a high number of autotrophs!), suggesting a substantial contribution of microbial-protozoan trophic steps in most cases.

L 334-335. Phagotrophy and osmotrophy cannot be distinguished using only $\delta^{15}\text{NGlu}$ (ref. 18). So, why do you interpret “evidence of phagotrophy” in Fig. 4? Perhaps adding a plot of $\delta^{15}\text{NAla}$ vs. $\delta^{15}\text{NPhe}$ in Fig. 3 would help to better differentiate between pure autotrophs and mixotrophs.

L351-352. Consider estimating microbial-protozoan contributions as in other studies (ref. 76; Fernández-Urruzola et al., 2023) to illustrate the relative increase in food chain length.

We will address these three comments together, because they are related. The first two belong to the discussion and the last to the summary paragraph.

First, we would like to apologize for the confusing way of laying Fig 4 out. After carefully reading both revision reports and discussing it, we have modified this plot for showing the apparent isotopic fractionation of Glu/Ala relative to Phe (instead of the fractionation of Phe relative to Glu/Ala in the previous plot). In this new plot shown in the next page, the more positive the number, the more enriched the trophic amino acids relative to Phe, which should make it more intuitive, as it represents the enrichment associated to trophic transfers better. In this new plot, points above the regression line actually represent samples in which Ala seems more enriched than Glu relative to Phe, as described in the works of Gutierrez-Rodriguez et al. (2014) and Décima et al (2017), and contrary to what happened in the previous plot, obviously confusing, in which points above the regression line represented samples in which Ala was less enriched.

The literature data for TP1 contains a mixture of cultures made with many different conditions. In reviewing the set, most of it compiled by McMahon et al. (2022), we realized that many TP1 were used as preys of herbivores, which were the actual focus of the publications. This implies a huge variety of media, light:dark cycles, and temperatures, not always using media containing only inorganics. For instance, there is one *Chlorella* spp culture on a medium supplemented with amino acids (a blue circle highlights this data point in new Fig 4 below), for which we calculated a $\text{TP}_{\text{Glu}/\text{Phe}}$ of 1.6, which suggests this mixoplankton was growing, at least, as osmo-phototroph. Though looking at the $\epsilon_{\text{Ala}/\text{Phe}}$, it would seem the discontinuous culture was not axenic and the mixoplankton was likely grazing on bacteria (no mention about the control of bacterial growth in the original paper, Lee et al. 2020).

Additionally, for increasing the data points of strict phagotrophic protists available in literature, we have added the data from *Oxyrrhis marina* cultures, a strict phagotrophic dinoflagellate, used by Gutierrez-Rodriguez et al. (2014) for assessing the role of Ala as integrator of protists trophic steps. These are the new light green diamonds in the plot (Refs 17 and 18 in legend), most of them overlapping with TP1 and below the linear regression line. As a result, we have modified the Methods and Results sections accordingly to account for the new calculation, data points and the linear regression equation in lines 205, 581-587.

Given that mixoplankton couples phagotrophy with osmotrophy and phototrophy, we suggest the use of these two amino acids for detecting the dominance of the coupled feeding strategies, as they should impact both Glu and Ala, contrary to what is shown in strict phagotrophs, which only impact Ala. We decided to use the apparent isotopic fractionation of trophic relative to source AA, because it is more appropriate formally than using the difference between the absolute $\delta^{15}\text{N}$ values (the values are similar but not exactly the same). We reckon this approach should currently complement the TP calculation using the canonical Glu/Phe. This canonical calculation will unequivocally account for osmotrophs using ambient amino acids along with phototrophy, after what Yamaguchi et al. (2017) showed. But for resolving if this mixotrophs are in fact mixoplankton, we need the comparison of $\epsilon_{\text{Glu/Phe}}$ (somehow accounted for in the $\text{TP}_{\text{Glu/Phe}}$ calculation) with $\epsilon_{\text{Ala/Phe}}$, as it is illustrated by our TP1.5 seston in the ARP, likely dominated by mixoplankton. In this regard, we have also found several samples along the ARP in which phagotrophs were clearly present, therefore, as mentioned by the Reviewer, they were samples with substantial contribution of microbial-protzoan trophic steps, invisible in the TP calculation based on Glu/Phe (these are the TP1 dark purple triangles inside the red circle in Fig 4 above).

We suspect that it is very likely that other amino acids may result useful for detecting mixoplankton and mixotrophs in the field. However, as we suggest in the text this will require further research with unicellular osmo-phototrophs and osmo-phago-phototrophs for accounting for specific trophic discrimination factors of these organisms as well as the β if we assume mixoplankton could outcompete photoautotrophs as base of the food web in mature systems, for instance. We also suspect that the cultivation of mixotrophs and mixoplankton done in a day:night cycles may result in different results from what is shown in Gutierrez-Rodriguez et al. (2014) and Décima et al. (2017), who used continuous light for the preys, and either continuous light or continuous dark for the grazers in all experiments, far from allowing diel cycles in the metabolisms of the organisms. All this results in an increase of the uncertainty for accounting for microbial steps specifically associated to mixoplankton.

The field studies mentioned by the Reviewer proved extremely useful for showing the presence of strict phagotrophs steps across different levels in marine food webs. Indeed, Viana et al. (2023) did an especially extensive literature analysis of a wide range of trophic levels and varied excretion metabolisms. But, also as mentioned by the Reviewer, these studies start primarily in metazoan zooplankton. Nothing is known about mixoplankton themselves. In our literature review, we found a seamless integration of trophic steps in the enrichment of trophic amino acids across planktonic trophic levels between TP2 and TP3 (see Fig. 4), with a continuous distribution of points with no gaps. Therefore, we would expect the hybrid metabolism of mixoplankton to result in an intermediate enrichment of both Glu and Ala, filling the gap between TP1 and TP2 in Fig. 4. This enrichment would occur right where the ARP TP1.5 seston distributes.

Considering all said above, we expect a high uncertainty in the calculation of microbial steps following the calculation proposed by Bode et al. (2021) and Viana et al. (2023):

$$\% \text{ of microbial steps} = \frac{TP_{Ala/Phe} - TP_{Glu/Phe}}{TP_{Ala/Phe}} \times 100$$

The main reason, obvious to the Reviewer, is that the TP calculation depends on the TDF (hence on TEF) and β chosen for estimating the TP based on Glu/Phe and Ala/Phe. We have done so with the values proposed by Chikaraishi et al. (2009) and Décima and Landry (2020), and plotted them below, where the range of the axes clearly illustrates the uncertainty associated to provide a unique numerical value for this %. As a reminder, the coefficients used were:

Chikaraishi et al. (2009)	β	TEF (TDF_{TrAA} – TDF_{Phe})
TP _{Glu/Phe}	3.4	7.6
TP _{Ala/Phe}	3.2	5.7
Décima and Landry (2020)	β	TEF (TDF_{TrAA} – TDF_{Phe})
TP _{Glu/Phe}	3.4	6.1
TP _{Ala/Phe}	3.2	4.5

As in the manuscript, the mixotrophs along the ARP in the figure are marked with an asterisk.

In summary, we appreciate the suggestion by the Reviewer, and agree that it is important to find a way for accurately account for both strict phagotrophic and mixoplanktonic steps in trophic ecology, as the complexity related to them at the base of food webs should determine the energy fueling upper trophic levels. Although their overall effect on this energy channeled up the food web may be different. However, we believe it is not possible to do it for the scope of our study, given the current uncertainty related to mixoplankton hybrid metabolism. Plainly, we were not looking for mixoplankton along the ARP, but mixoplankton found us. When looking for a potential explanations for our data, we found this consistency in the comparison of $\epsilon_{\text{Glu/Phe}}$ with $\epsilon_{\text{Ala/Phe}}$ in literature showing the potential space where mixoplankton distributes due to their combined osmo-phago-phototrophy. But we think that the main finding of our work is pointing to a promising direction for at least detecting the activity of mixoplankton in field samples that could complement other approaches currently developed or under development by the community. This is why we have deleted the sentence in L351-352 in the summary paragraph (“From our results, it is not possible to define the implications for the energy available to upper trophic levels, ...”), as it clearly requires, as suggested by the Reviewer, a numerical calculation of the % of microbial steps, which we do not believe it is possible to reliable do for our study, and hence moved the focus to the comparison of $\epsilon_{\text{Glu/Phe}}$ with $\epsilon_{\text{Ala/Phe}}$.

Given the additional confusion in section “Phagotrophy and osmotrophy potential impact on trophic amino acids” (comment for L333-334), which we realized due to both Reviewers comments, we have edited the text for clarity. In addition to what is explained above, we hope that the revised section in the main text explaining the potential of $\epsilon_{\text{Glu/Phe}}$ and $\epsilon_{\text{Ala/Phe}}$ is clearer, and

supports why the combination of these two apparent isotope fractionations revealed the activity of mixoplankton and strict phagotrophs.

- Bode, A., M. P. Olivar, C. López-Pérez, and S. Hernández-León. 2021. The microbial contribution to the trophic position of stomiiform fishes. *ICES J. Mar. Sci.* 78: 3245–3253. doi:10.1093/icesjms/fsab189
- Chikaraishi, Y., N. O. Ogawa, Y. Kashiyama, and others. 2009. Determination of aquatic food-web structure based on compound-specific nitrogen isotopic composition of amino acids. *Limnol. Oceanogr.* 7: 740–750. doi:10.4319/lom.2009.7.740
- Décima, M., and M. R. Landry. 2020. Resilience of plankton trophic structure to an eddy-stimulated diatom bloom in the North Pacific Subtropical Gyre. *Mar. Ecol. Prog. Ser.* 643: 33–48. doi:https://doi.org/10.3354/meps13333
- Décima, M., M. R. Landry, C. J. Bradley, and M. L. Fogel. 2017. Alanine $\delta^{15}\text{N}$ trophic fractionation in heterotrophic protists. *Limnol. Oceanogr.* 62: 2308–2322. doi:https://doi.org/10.1002/lno.10567
- Gutierrez-Rodriguez, A., M. Decima, B. N. Popp, and M. R. Landry. 2014. Isotopic invisibility of protozoan trophic steps in marine food webs. *Limnol. Oceanogr.* 59: 1590–1598. doi:10.4319/lno.2014.59.5.1590
- Lee, M.-C., H. Choi, J. C. Park, and others. 2020. A comparative study of food selectivity of the benthic copepod *Tigriopus japonicus* and the pelagic copepod *Paracyclopsina nana*: A genome-wide identification of fatty acid conversion genes and nitrogen isotope investigation. *Aquaculture* 521: 734930. doi:https://doi.org/10.1016/j.aquaculture.2020.734930
- McMahon, K. W., M. D. Ramirez, A. Besser, and S. D. Newsome. 2022. Primary producer amino acid nitrogen isotope values from published literature to examine beta variability in trophic position estimates. (Version 1) Version Date 2022-03-08. doi:doi:10.26008/1912/bco-dmo.870320.1
- Viana, I. G., R. García-Seoane, and A. Bode. 2023. A missing trophic link: Contribution of the microbial loop to the estimation of the trophic position of pelagic consumers. *Limnol. Oceanogr.* 68: 2587–2602. doi:https://doi.org/10.1002/lno.12445
- Yamaguchi, Y. T., Y. Chikaraishi, Y. Takano, N. O. Ogawa, H. Imachi, Y. Yokoyama, and N. Ohkouchi. 2017. Fractionation of nitrogen isotopes during amino acid metabolism in heterotrophic and chemolithoautotrophic microbes across Eukarya, Bacteria, and Archaea: Effects of nitrogen sources and metabolic pathways. *Org. Geochem.* 111: 101–112. doi:https://doi.org/10.1016/j.orggeochem.2017.04.004

L 329 and 334: correct references to Figure 3 (not Fig. 4)

This sentence in the “Phagotrophy and osmotrophy potential impact on trophic amino acids” section of the discussion referred to Figure 4. We realized that the text was really confusing and seemed to related to the Glu vs Phe space in Fig. 3.

As mentioned above, we have edited the text in this section of the discussion for providing a clearer explanation of the line of reasoning, and changed the labels in the legend of Fig 4 to

represent the TP of the different groups of points. This change in legend is also addressing concerns raised by Reviewer #1.

L358-367: “To our knowledge, this thesis was only studied in cultures of strict phagotrophs, but when mixoplankton are actively grazing on other microorganisms, a difference in $\epsilon_{\text{Ala/Phe}}$ similar to that of phagotrophs should occur and occupy the space of apparent fractionation of Ala towards the values of organisms in TP 2.0 in Figure 4. Additionally, Yamaguchi et al (2017)¹⁶ showed that different groups of Archaea, fungi and bacteria growing only on organic media (i.e, as osmotrophs) presented an apparent fractionation of Glu similar to that of herbivores (TP 2.0). Since all microalgae are capable of non-auxotrophic osmotrophy of amino acids, among other primary metabolites¹⁻³, we would expect mixotrophs, which rely on osmotrophy and autotrophy in equal measure, to also be distributed across the space of apparent fractionation of Glu between TP 1.0 and TP 2.0 (Fig. 4).”

L 586 References: revise journal style and provide enough information to identify the source (e.g. refs. 25, 27, 31, 44, 55, 65, 85)

Thanks for pointing to the wrong formatting of the references. We have reviewed the records in our citation software, and the style for making sure the style of the journal was respected.

Additional references:

Décima, M. Zooplankton trophic structure and ecosystem productivity. Mar. Ecol. Prog. Ser. 692, 23-42 (2022). <https://doi.org/10.3354/meps14077>

Décima, M. & Landry, M. Resilience of plankton trophic structure to an eddy-stimulated diatom bloom in the North Pacific Subtropical Gyre. Mar. Ecol. Prog. Ser. 643, 33-48 (2020). <https://doi.org/10.3354/meps13333>

Fernández-Urruzola, I. et al. Trophic ecology of midwater zooplankton along a productivity gradient in the Southeast Pacific. Front. Mar. Sci. 10, 1057502 (2023). <https://doi.org/10.3389/fmars.2023.1057502>

Legendre, L. & Rassoulzadegan, F. Plankton and nutrient dynamics in marine waters. Ophelia 41, 153-172 (1995). <https://doi.org/10.1080/00785236.1995.10422042>

Viana, I. G., García-Seoane, R. & Bode, A. The missing trophic link: Contribution of the microbial loop to the estimation of the trophic position of pelagic consumers. Limnol. Oceanogr. 68, 2587-2602 (2023). <https://doi.org/10.1002/lno.12445>

Rebuttal Letter Second Revision Report by Reviewer #1 to

Mixotrophy emerges as the optimal strategy in mature waters of the Amazon River plume

Reviewer #1 (Remarks to the Author):

Review of Version 2

We appreciate the time the reviewer dedicated to revising our manuscript. After reading the full revision report, we realized that the approach we used for amino acid nitrogen isotopes in our CSIA-AA measurements might not be obvious to experts without this particular area of expertise. Therefore, we have decided to frame the context differently because evidence in literature is robust, but was not explicitly presented.

We have included an explanation (L87-119) and new figure (Figure 1) in the introduction to clearly show the different nitrogen isotopes end members affecting the TP calculations, including autotrophic microalgae and microbes, osmotrophic microbes, and phagotrophic protozooplankton (McClelland et al. 2003; McCarthy et al. 2007; Chikaraishi et al. 2009; Yamaguchi et al. 2017; Décima et al. 2017). We have also used this figure in the Discussion to demonstrate step by step how our mixotrophic samples align with existing literature and highlight the possibility of identifying a predominance of osmo-, phago-, or photoautotrophy in seston samples, always provided they primarily represent the signature of microalgae (as explained in the manuscript).

In addition, we have added an explanation to the Supporting Information for readers without expertise in isotope ecology about the fundamentals, that we know it is useless for the reviewer, but hope will be useful for non-experts reaching our work.

Now, we will address each comment in detail in the text below.

I have looked over the rebuttal document and make the following observations:

In general the authors have made good replies. However, I am concerned that some of them (especially concerning terminology) have not all been implemented. Thus, there remains the problems around 'mixotrophy' and 'phytoplankton' etc. (see comments on the new document); it is as if who ever wrote the rebuttal did not check what was actually done, or vice versa.

We understand that the usage of terminology originating from carbon metabolism can be challenging when extended to the movement of nitrogen through food webs. Therefore, the Introduction provides definitions of the terms used in the context of the manuscript. The terms were used in accordance with the definitions provided, except for an error in the graphical abstract, where "phytoplankton" was not replaced by "microalgae".

However, to make the text easier to read, we have now included a table with an alphabetically ordered glossary of terms at the end of the introduction (shown below). We would like to restate

that our trophic position (TP) estimates are amino acid nitrogen-based. Hence, we have defined terms such as "autotroph" and "mixotroph" in relation to the TPs calculated in terms of nitrogen. Therefore, they are autotrophs if both, TP_{Ala} and TP_{Glu} have a value of 1.0, while they are "mixotrophs" if either, TP_{Ala} or TP_{Glu}, have a value of 1.5, with Ala reflecting phagotrophy and Glu osmotrophy (see reply to next comment below).

Term	Definition
Autotroph	An organism capable of generating its own food from dissolved inorganic nutrients and a trophic position of 1.0 for both, TP _{Glu} and TP _{Ala} .
Microalgae	A community of photosynthesizing unicellular plankton organisms that are pigmented and include phytoplankton and mixoplankton.
Mixoplankton	A unicellular, eukaryotic, planktonic organism (i.e., a protist) that combines osmo-, phago-, and photoautotrophy. This includes all protists except diatoms.
Mixotroph	An organism that grows by combining photoautotrophy, osmotrophy, and/or phagotrophy and a trophic position of 1.5 TP _{Glu} and/or TP _{Ala} .
Osmotrophy	A nutritional mode by which organisms absorb dissolved organic compounds from their environment, reflecting in a trophic position of 2.0 for TP _{Glu} and of 1.0 for TP _{Ala} , respectively, when it is their sole nutritional mode.
Phagotrophy	A nutritional mode by which organisms obtain nutrients by grazing and ingesting other organisms, reflecting in a trophic position of 2.0 for TP _{Ala} and of 1.0 for TP _{Glu} , respectively, when it is their sole nutritional mode.
Photoautotrophy	A nutritional mode by which organisms use light energy to create their own food from inorganic sources, reflecting in a TP of 1.0 for both TP _{Glu} and TP _{Ala} .
Phytoplankton	A unicellular prokaryotic or eukaryotic primary producer combining only osmo-photoautotrophy, namely cyanobacteria and diatoms, which are unable to phagocytose other organisms.
Trophic Position	The numerical position an organism occupies in a food web. In our study, trophic position (TP) is calculated based on nitrogen isotopes in the trophic amino acids glutamic acid (Glu) or alanine (Ala) versus the source amino acid phenylalanine (Phe), e.g., TP _{Glu} or TP _{Ala} .

Table 1. Glossary of terminology used in this manuscript. An alphabetical list of definitions for key terms in our manuscript.

Concerning the response to the use of stable isotopes, whether the amino acid technique is really as robust as is claimed here requires reference to a modelling approach (similar to the Flynn et al. method) that actually track the fate of isotopes. I have to say, however, that I would be

extremely surprised if a unique signature appears, not least because of the taxonomic range of mixoplankton and the range of prey items. The least that needs to be done is a rigorous laboratory study; from what I can see, this has not been done. The reason that lab and field methods are alone unsuitable for this test is that they assume a trophic level. In especially highly complex microbial systems, allocation of a trophic level is itself highly questionable.

Indeed, the authors do accept in their rebuttal (and also in the revised Discussion) that the approach needs benchmarking against better data sets.

Compound-specific isotope analysis of amino acids (CSIA-AA) is a relatively new scientific field, but it is steadily growing and finding new applications in resolving challenging questions in field research. Over the last 20 years, sufficient empirical evidence has emerged to support the robustness and greater accuracy of the method compared to bulk isotopes in trophic ecology for assessing trophic hierarchies. We can agree that a modeling approach similar to that suggested by the reviewer would be useful for complementing what is available in literature tracking metabolic pathways. However, there is ample empirical evidence demonstrating the robustness of CSIA-AA in trophic ecology and allocation of trophic levels, so we do not need to claim it ourselves. We have included an extended explanation of the metabolic basis (very likely independent of taxonomic variety) and references to the relevant empirical work in the field of CSIA-AA nitrogen isotopes in the *Supporting Information*.

Published empirical evidence illustrated the nitrogen isotopic differential enrichment when organisms are growing autotrophically, osmotrophically or phagotrophically, that is, there is already published evidence on the different end members contributing to mixoplankton activity. This relevant literature, referenced to in the previous version of the manuscript, is now graphically shown in new Figure 1 in the introduction (L87-119). The literature review contains microalgae grown on inorganic nitrogen (McClelland et al. 2003; McCarthy et al. 2007; Chikaraishi et al. 2009), the autotrophic and osmotrophic microbes in Yamaguchi et al. (2017), and protozooplankton (strict phagotrophs) in Décima et al. (2017). These data are also shown in a new supplementary table in excel (Table S2). The new plot shows the relation of the nitrogen isotopic signature of Glu and Ala relative to Phe in the trophic spaces to clearly illustrate how the different feeding strategies reflect in the TP calculations based on nitrogen isotopes.

We have also acknowledged in the text that there is a lack of specific data on mixoplankton growing osmo-phago-photoautotrophically. The main reason is that the mixoplankton paradigm is so recent that the CSIA-AA community has not yet taken an interest in it. However, there is published evidence illustrating the different end members.

The new text in the introduction and figures are presented below.

L87-119: “The graphical analysis of trophic position is done by comparing the $\delta^{15}\text{N}$ values of a trophic amino acid (e.g., glutamic acid, Glu, or alanine, Ala) with those of a source amino acid (e.g., phenylalanine, Phe). In this trophic space, it is possible to define trophoclines (Fig. 1), that is, lines representing the same trophic position which result from the combination of $\delta^{15}\text{N}$ values of the two amino acids along the aforementioned equations²⁴. Trophoclines can be labeled as

TP_{Glu} and TP_{Ala} for representing the trophic amino acid, glutamic acid + glutamate and alanine, respectively. The potential of CSIA-AA to address complexity in natural systems is evident in its ability to track both nitrogen assimilation pathways and metabolic processes associated with mixotrophy. Yamaguchi et al. (2017)¹⁶ compared the nitrogen isotope signatures of the amino acids of different groups of microbes (*Archaea*, fungi and bacteria) growing on ammonium or casamino acids. Those growing solely on ammonium as dissolved inorganic nitrogen (DIN) source were capable of de novo synthesis of their own amino acids, hence presented a difference in the nitrogen isotopes of Glu and Ala relative to Phe similar to that of autotrophs (Fig 1A, B), and overlapped with literature autotrophic microalgae end member around TP_{Glu} and TP_{Ala} 1.0. By contrast, the authors showed that the same microbes growing on dissolved organic nitrogen (DON) from a casamino acids media (i.e, as osmotrophs) presented a nitrogen isotope enrichment of Glu similar to that of herbivores, clearly distributing around the TP_{Glu} 2.0 (Fig 1A). That is, their TP based on amino acid nitrogen isotopes was consistent with herbivores. However, the impact on nitrogen isotope enrichment in Ala was not as clear as that of Glu with data distributing between the trophocline of autotrophs (TP_{Ala} 1.0, Fig 1B) to that of herbivores (TP_{Ala} 2.0, Fig 1B). The authors suggested that nitrogen isotopes in Glu integrated the heterotrophic processed related to osmotrophy better than nitrogen isotopes in other amino acids like Ala. By contrast, the unique role of Ala as integrator of protistan phagotrophic steps in the food web, invisible in nitrogen isotopes in Glu, was shown by Gutierrez-Rodriguez et al. (2014)¹⁷ and Décima et al. (2017)¹⁸ in diverse protozooplankton cultures (*Oxyrrhis marina*, *Heterocapsa triquetra*, and *Favella* spp.). When nitrogen from phagocytosis as feeding strategy is the sole nitrogen source of protists, that is, when they are the herbivorous trophic step in a food web, nitrogen isotopes in Glu do not enrich proportionally to this trophic step, hence TP_{Glu} is located also below 1.5 close to the autotrophic end member (Fig 1A). By contrast, TP_{Ala} of protozooplankton uniformly distributes around 2.0, clearly reflecting the trophic mode of these grazers (Fig 1B). Hence, given the distinct effect of osmotrophy and phagotrophy on their nitrogen isotopes, the combination of glutamic acid and alanine nitrogen isotopic signatures has the potential to detect mixoplankton activity in natural samples.”

Figure 1. **A)** $\delta^{15}\text{N}_{\text{Phe}}$ (phenylalanine) vs $\delta^{15}\text{N}_{\text{Glu}}$ (glutamic acid + glutamate) and **B)** $\delta^{15}\text{N}_{\text{Phe}}$ (phenylalanine) vs $\delta^{15}\text{N}_{\text{Ala}}$ (alanine) of the literature end members. Symbols represent the organisms: cultured fungi (square), cultured bacteria (diamond), cultured Archaea (triangle), protozooplankton (crossed circle), and cultured microalgae (circle). Colors represent the nitrogen source: dissolved inorganic nitrogen (green, DIN), dissolved organic nitrogen (purple, DON), nitrogen from phagocytosis (pink, N from preys). Trophoclines (dashed and dotted gray lines) with a slope of 1.0 and y-intercepts of 3.4‰, 7.2‰, and 11.0‰ for $\delta^{15}\text{N}_{\text{Glu}}$ and of 3.2‰, 6.0‰, and 8.9‰ for $\delta^{15}\text{N}_{\text{Ala}}$, respectively, represent different trophic positions (TPs = 1.0, 1.5, 2.0) according to Chikaraishi et al. (2009).

The lack of good organism identification is a great loss to this study. Being able to assign osmotrophy as osmo' and/or phago is really very important in trophic level structuring. Only phagotrophy would (classically) be identified as a means to alter the trophic level. Osmotrophy would not be considered as a means to shift the TL. One then wonders how osmotrophy operates when one considers that much uptake of DOM may simply represent within-guild take up of leakage products.

There is not a complete lack of identification, we used the available samples for taxonomy (i.e., specific pigment markers), using a commonly employed approach (i.e., ChemTax) for defining microalgae functional groups. Unfortunately, we lack genomics to address the composition of the entire community or quantitative microscopy to partially address the microplankton size fraction. However, we were able to provide the functional groups commonly used in marine ecology and remote sensing, for at least, pointing to the potential presence of mixoplankton in our samples, and the likeliness of their dominance based on the relative contribution to the community chlorophyll *a*.

Our TP is based on amino acid nitrogen isotopes of Glu and Phe. A trophic shift in amino acid nitrogen will reflect on the nitrogen isotopic signature of trophic amino acids, increasing their signature relative to the previous trophic step. The effect of osmotrophy in nitrogen isotopes was clearly demonstrated by Yamaguchi et al. (2017) and it is now shown in Figure 1 and explained in the Introduction. Mixotrophs will hold a nitrogen signature in their amino acids reflecting the partition between osmotrophy and autotrophy for covering their metabolic requirements. When autotrophy is more dominant than osmotrophy, the resulting TP will be close to the trophocline representing TP1 (it will be predominantly autotroph). While when osmotrophy is more dominant, the TP will be close to TP2, and this is when we should be able to detect a shift in the trophic level calculated by nitrogen isotopes. When organisms rely more on ambient nitrogen organic compounds than on self-produced nitrogen compounds, they are more heterotrophic. From the perspective of nitrogen isotopes, they will reflect a shift in TP.

I much appreciated the commentary made by Reviewer #2. It reaffirms my previous concerns on the whole use of stable isotopes, that the approach is helpful, but that ultimately (as per the Flynn et al (2018) conclusion), the method is confounded by the fact that the TL concept itself is problematic. That osmotrophy and phagotrophy may give the same signal is also highly problematic.

It is unclear to us what comment made by Reviewer #2 suggests that CSIA-AA is not useful or robust for trophic ecology studies and definition of TL. In the previous revision report, Reviewer #2 quoted in L326-338, L334-335 and L351-351 the importance of accounting for the so-called "protistan/microbial trophic steps" which related to the activity of protist phagotrophs (i.e., protozooplankton), which are invisible in the $\delta^{15}\text{N}$ signature of Glu but visible in the $\delta^{15}\text{N}$ of Ala (now shown in new Figure 1 and the text in L87-119). We agree with Reviewer #2 that this is an important issue requiring further research on both strict phagotrophs and mixoplankton. This will reduce the uncertainty in defining trophic hierarchies based on CSIA-AA in systems where this "invisible" trophic step could significantly impact the transfer of energy to upper trophic levels. As a result, we included this in the text, pointing to the need of combining TP_{Glu} and TP_{Ala} for defining the TP of microplankton in this context (L415-417 and L455-458).

As shown in the literature revision in Figure 1, osmotrophy and phagotrophy do not give the same $\delta^{15}\text{N}$ signal in Ala and Glu, and this is the basis of the approach we follow. More controlled experiments with mixoplankton cultures are a further step to learn about the regulation of alanine

turnover and/or catabolism in comparison to glutamic acid turnover and/or anabolism. However, laboratory studies show already that osmotrophy reflects on the $\delta^{15}\text{N}$ of Glu (Yamaguchi et al. 2017), while phagotrophy reflects on the $\delta^{15}\text{N}$ of Ala (Gutierrez-Rodriguez et al. 2014; Décima et al. 2017). As pointed by Reviewer #2 and our own previous reply, it is therefore necessary to combine the apparent $\delta^{15}\text{N}$ -enrichment of these two amino acids relative to the source amino acid keeping the information of the baseline, that is, to the $\delta^{15}\text{N}$ of phenylalanine (Phe). The strength of the CSIA-AA approach lies in the metabolic basis of the nitrogen isotopic fractionation of each amino acid, which depends on the turnover times and the biochemical routes they follow. We agree that more research focused on mixoplankton is necessary, but we think our field-based approach is the first important step to identify at times dominance of mixo-over autotrophy in the field. Knowing that the mature Amazon River plume in April/June is a hot spot and hot moment of mixotrophy allows us to go back there and to study mixotrophy and its biogeochemical consequences in more detail including research suggested by Reviewer #2.

As explained above, we added the underlying biochemistry and state of the art knowledge of amino acid nitrogen isotope-based trophic ecology in the supplements in the form of an “Isotope Ecology 101” in the Supporting Information.

For the new document, I make the following observations:

Please, note that to ease the discussion process for the text below the comments for the Graphical Abstract, we have added a line before each group of comments pointing to the section from which they come.

Graphic abstract; top right. These images give an impression that organism identification has been performed, and/or that certain species dominate. That is not correct, as far as I can see. AND these are not all 'phytoplankton'. The middle one is almost certainly a mixoplankton, for example.

Graphic abstract; lower part. The presence here of the word 'phytoplankton' is inappropriate.

We corrected “phytoplankton” to “microalgae”, thanks for noticing! When revising the graphical abstract and text in the first round, we have missed to amend “phytoplankton” by “microalgae” at that spot.

In relation to the pictures in the diagram showing the succession of functional types, we would like to clarify that they are the pictures by a PlanktoScope v2.1 and used for validating the chemotaxonomy (Methods). The set is open in EcoTaxa (the link is provided in the main text: <https://ecotaxa.obs-vlfr.fr/prj/6346>). As we explained in the methods, this sampling, only done in 2021, was merely qualitative, represents only the microplankton size fraction, and was used for validating the classification of the whole community done by HPLC pigments. This is the taxonomical classification that we can provide for the whole community of pigmented plankton.

We have removed the pictures from the graphical abstract. The new version is:

Potential impact of Phagotrophy and Osmotrophy on trophic amino acids

Comments related to the Abstract

L24 This is not correct; they may be mixoplankton.

L25 I suspect there are no 'strict' autotrophs. This term is also ambiguous; does it suggest that they only use autotrophy, or that they must involve autotrophy? (the latter is likely common in mixoplankton, in any case, that they cannot grow solely heterotrophically).

L31 To secure this statement you need identifications and evidence of phagotrophy. I am not convinced you have that. The method (as claimed) only indicates a scale of photo + osmo/phagotrophy. Note that these '-trophy' terms are not nouns; they are not identifying organism types.

These three comments belong to the Abstract and we will address them together. First, we would like to remind that given the limited word count of the abstract (exactly 200), it is very difficult to include many nuances in text.

It is unclear what is the inaccuracy in L24, when we say that “All other unicellular photosynthetic eukaryotes, except diatoms, are potentially phagotrophs and are currently classified as mixoplankton”. If they can potentially phagocytose other plankton, they must be mixoplankton, not phytoplankton.

We have removed “strict” from L25.

L31 is in relation to the issues raised for L98 and L371 below. We thus refer to the answers below, Figure 1 and the text in the introduction (L87-119) as well as the new section in the supplements for a detailed answer.

Comments related to the Introduction

L45 This is too strong; it is suggested that many can exploit phagotrophy. All these organisms do not 'rely' upon phagotrophy. Indeed, it is likely that very few mixoplankton actually 'rely' on it (exceptions notably being the non-constitutive mixoplankton).

We have replaced “rely” by “exploit” to accommodate the comment by the Reviewer, though when using “varying degrees” we meant the range that goes from not consuming other organisms at all to relying solely on phagotrophy.

L49 You mean relying on osmotrophy for heterotrophy

Indeed, they rely on osmotrophy as their only means of heterotrophy. We have modified the sentence as follows:

L48-50: “The impact of mixotrophs that rely on a combination of osmotrophy and photoautotrophy on biogeochemical cycling differs significantly from that of mixoplankton that use phagotrophy as well.”

L98 This seems too strong here. For example, how does this differ with osmotrophic consumption of material that is leaked when mixoplankton toxins kill a prey and make it lyse?

The issue of diel cycles also need to be resolved. Is osmotrophy of DFAA in darkness handled differently in the light? (Yes it is, at least in diatoms!!) How is the isotope ratio affected by the use of DFAA recovered in darkness by phytoplankton that actually leaked those same DFAA in the previous light phase? The whole topic needs to be grounded in lab work first, no?

To the best of our knowledge, there is no specific study about isotopic changes due to the processes mentioned by the Reviewer. However, the isotopic signatures of microalgae typically integrate over a period of a few days (O'Reilly et al. 2002). Hence, diel cycles are reflected in the overall signature of organisms as a weighted mean, showing the importance of each process that supplies or modifies nitrogenous compounds within the cell. The metabolic routes of ambient amino acids should be similar regardless of origin and therefore leave a similar imprint on the nitrogen isotopic signatures of the amino acids. We would also like to note that these nuances in origin of the amino acids or effect of diel cycling are beyond the scope of our study and the application of CSIA AA. Our data is the net result of the allocation of the nitrogen within cells by metabolic routes over at least 24h in microplankton communities. And in this sentence, we simply state that CSIA-AA has “the potential” of addressing the feeding strategies of mixoplankton.

We have modified the sentence as follows:

L117-119: “Hence, given the distinct effect of osmotrophy and phagotrophy on their nitrogen isotopes, the combination of glutamic acid and alanine nitrogen isotopic signatures has the potential to detect mixoplankton activity in natural samples.”

Comments related to the Results

L178 Cryptophytes are quite likely mixoplankton.

This sentence in the Results section addresses the composition of communities at the river's mouth, but does not comment on cryptophytes not being mixoplankton. It is unclear what are the modifications suggested by the reviewer here, because we are still not discussing who is mixoplankton or who is not. This is a description of the functional groups present in every community along the Amazon River plume. The fact that cryptophytes belong to mixoplankton is acknowledged in the closing paragraph of the section "Microalgae community succession along the Amazon River plume" in the Discussion in L378-383, as shown below.

L378-383: “As shown, prasinophytes, haptophytes, dinoflagellates and cryptophytes were present in all microalgae communities. These functional groups of unicellular eukaryotes are known to contained mixoplankton species, coupling osmo-phago-photoautotrophy.”

L200 This should read ‘mixotrophy’, not the noun ‘mixotrophs’.

In this sentence in the Results, we are describing the TPs shown in the glutamic vs phenylalanine space (Fig. 3). Hence, and following the terminology defined in L146-159, it should read “mixotroph” as we are referring to TP1.5, as also explained in the Glossary in Table 1.

L206 While it may be significant statistically, the scatter even around points that are very robust, is to my mind too great.

L206 refers to the slope and intercept not being significantly different from 1 and 0, but we have interpreted this comment as referring to the previous line where the results and significance of the linear regression are given.

We agree that about six points fall far above or below the microalgae cloud. It is also possible that a couple of points below 0‰ on each axis represent outliers. So, eight out of 39 microalgae points seem to be off the cloud. Therefore, it is reasonable to conclude that the relationship found in a summary of literature data, which mixes laboratory samples of microalgae and protozooplankton with field samples of metazooplankton, is not only statistically significant, but also robust because it explains 82% of the variation within the data (Sokal and Rohlf 1995). In other words, the model has strong explanatory power ($R^2 = 0.82$), and is statistically reliable robust ($p < 0.001$).

Taking those outliers out, the resulting relation is still robust.

However, as we have decided to illustrate the approach using the actual values of $\delta^{15}\text{N}$ of each amino acid (Glu, Ala and Phe) in relation to the trophoclines, this figure, the epsilons approach and the linear regression were removed from the text.

Even though this plot was removed from the current version of the manuscript, we would like to show the plot using the new literature revision for unicellular organisms and the different end members to demonstrate that the relation remains consistent. Below, we show a three-panel plot containing each $\epsilon_{\text{TrAA}/\text{Phe}}$ vs. its corresponding TP_{TrAA} (upper plots), and the $\epsilon_{\text{Ala}/\text{Phe}}$ and $\epsilon_{\text{Glu}/\text{Phe}}$ (lower plot). The upper plots show a very tight linear relation because the difference in ^{15}N

between trophic and source amino acids is proportional to the trophic step, and this difference determines the TP calculation. The lower plot still shows a consistent relation between $\epsilon_{\text{Ala/Phe}}$ and $\epsilon_{\text{Glu/Phe}}$, and presents a gap in literature between predominant autotrophs and predominant osmotrophs, where mixotrophs and mixoplankton would distribute based on the combination of feeding strategies that supply light amino acids (autotrophy) and feeding strategies that enrich the ^{15}N of amino acids (osmotrophy and phagotrophy).

L216 This needs to be made explicit as to the quantification. % of what, exactly? Biomass, numeric abundance?

We are describing in the Result the importance of each classifier separating the TP of the samples. In C5.0 is reported as the fraction of cases classified in each branch in %. This is explained at the beginning of the sentence (L215-216). The information is provided in the output of C5.0, and represents therefore, the % of samples classified in each branch of the tree shown in Fig 5. This is why the first classifier has a 100% because it can separate all cases in two main branches, but the last only a 21.7% because the rest of the cases were already separated by the other classifiers.

For making this cleared, we have modified the text as follows:

L241-244: “The relative attribute usage, or the percent of cases included in each branch of the classification tree for each classifier, was 100 % of cases for mixed layer depth, 73.9 % for surface oxygen concentration, 52.2 % for the maximum of buoyancy frequency, and 21.7 % for surface chlorophyll *a*.”

Comments related to the Discussion

L255 'pure' autotrophy is too strong a term; 'strongly', or 'predominantly' perhaps? The scatter from plots such as Fig.4 to my mind precludes strong terminology.

This sentence in the Discussion refers to a TP of 1.0, as an absolute value it indicates that the organism is purely autotrophic, and should fall on the trophocline of TP1 in figure 3. This sentence does not describe our data or literature data, but rather sets the starting point for any reading not familiar with trophic ecology. A TP of 1.0 indicates that autotrophy is the organism's sole mode. We believe it is important to set this here, therefore, “pure” is the appropriate adjective.

We have modified the text as follows for a clearer explanation:

L284-285: “Here an exact TP_{Glu} or TP_{Ala} of 1.0 indicates pure autotrophy, while deviations between 1.0 and 1.4 are interpreted as a predominance of autotrophy.”

L263 I would not support such a view. Many mixoplankton feed on bacteria (not sure what TP that would be, but it would not be TP1, for sure), and many others feed on other mixoplankton. This crude TP assumption for mixoplankton cannot be considered as safe.

This sentence in the Discussion is still explaining the framework for a non-specialist reader (the sentence reads: “Consequently, organisms combining both autotrophic and heterotrophic nutrition should occupy an intermediate position between TP 1.0 and TP 2.0”).

From the perspective of the nitrogen isotopes, and how the combined autotrophic and heterotrophic metabolisms reflect in the $\delta^{15}\text{N}$ of the trophic amino acids, this sentence is appropriate, so we have made no further modification. We refer to previous explanations, the

literature data in Figure 1 and the new section in the Supporting Information for supporting this decision.

L287 What evidence is there for this? This also assumes that mixotrophy is to supply C (noting as an aside that your methods track N).

This sentence in the Discussion is going through the C5.0 classifiers. We are trying to discuss a possible explanation for the undersaturated waters found in stations where seston had a clear TP of 1.5. This is based, on the one hand, on the empirical evidence provided by Wilken et al. (2014), who found that some mixoplankton can use the photosynthetic apparatus to provide energy with oxygen consumption by photorespiration rather than production by photosynthesis. Therefore, the photosynthetic apparatus contributes to oxygen consumption. On the other hand, surface waters should be at saturation by physicochemical processes alone. Oversaturation is possible when primary production is high, but undersaturation in our system suggests biological oxygen consumption. Hence, it is likely that mixoplankton dominate the community's activity by relying on photorespiration, which consumes oxygen.

We have modified the sentence to explain this cleared as follows:

L317-319: “If photoheterotrophy dominates, mixoplankton may use the photosynthetic apparatus for energy production rather than carbon fixation, thus contributing to oxygen consumption.”

L298 This is not correct. The evidence is not strong enough for that comment. The evidence shows that there are members of these groups that are mixoplanktonic. More profoundly, however, all of these organisms have a mixotrophic capability. Diatoms are well known for it, for example (as you note below).

It is unclear why this sentence in the context of the whole paragraph is not correct. This sentence in the Discussion attempts to explain a station in 2018 (i.e., total seston without fractionation) where almost 90% of the chlorophyll *a* of the community was explained by diatom pigments, but the TP was 1.5. Hence, our chemotaxonomy suggests that the pigmented community was dominated by phytoplankton. Mixoplankton were minor contributors (<10%). Then because the final isotopic signature is the result of a weighted average of each component of the community, the signature of diatoms very likely drives that of the community (Fry 2006).

We have modified the sentence to express this clearer:

L329-333: “Although all of the minor groups in this station, except *Trichodesmium*, are known members of the mixoplankton combining osmo-phago-photoautotrophy, it is very unlikely that they were able to override the potential dominance of diatoms in the nitrogen signatures of Glu and Ala of the whole sample, because the sum of all of them accounted for only 10% of the chlorophyll *a*.”

L371 This appraisal is honest, but ultimately it also leaves us with not being sure of anything. What was needed here was a microscope assessment of species and of the presence of clear evidence of mixoplankton grazing activity (short term incubations?).

In the previous version of the manuscript L371-372 read: “However, the values of $\epsilon_{\text{Ala/Phe}}$ are also consistent with a contribution of phagotrophy in most mixotrophic samples, as Ala is more enriched than Phe in relation to organisms in TP1.0 (Fig 4).”

In the new approach for explaining how we can detect autotrophs, osmotrophs and phagotrophs in the context of literature evidence (Figure 1, Table S2), it is clear that half of our TP_{Glu} 1.5 samples also presented the imprint of phagotrophy with a TP_{Ala} 1.5.

This section was modified to show the trophic spaces between Glu-Phe and Ala-Phe to better illustrate what samples presented a predominance of photoautotrophy, osmo-photoautotrophy, phago-photoautotrophy and osmo-phago-photoautotrophy in the community.

The new section, supported by the literature end members, reads as follows:

L385-421: “The heterotrophic reworking of amino acids acquired by osmotrophy or phagotrophy seems to reflect distinctively on the nitrogen isotope signatures of Glu and Ala, producing deviations from the signatures of autotrophs, and capturing the trophic shift. To define the ranges of variation in TP associated to each feeding strategy of unicellular plankton, we conducted a literature review of the different nitrogen isotopic end members (Fig 1, Table S2): i) microalgae, fungi, *Archaea* and bacteria growing autotrophically on inorganic nitrogen sources (DIN); ii) fungi, *Archaea* and bacteria growing heterotrophically on an organic nitrogen source (DON); and iii) strict protist phagotrophs preying autotrophic microalgae (N from preys). The canonical TP_{Glu} was used for discussing the main trophic position in the sections above. However, TP_{Glu} only accounts for mixotrophs combining osmotrophy and photoautotrophy. Whether these are also mixoplankton or not requires the comparison with TP_{Ala}. This way it would be possible to resolve the contribution of nitrogen derived from the phagocytosis of preys, hence allowing the subsequent identification of the trophic mode of mixoplankton combining osmo-phago-photoautotrophy.

If we compare the TP_{Glu} and TP_{Ala} of our seston samples (Fig 4) with the literature end members (Fig 1), we should be able to resolve the predominance of photoautotrophy, osmo-photoautotrophy, phago-photoautotrophy and osmo-phago-photoautotrophy in the community. Five of our mixotrophic seston samples showed a relatively balanced contribution of DIN and DON in their TP_{Glu}, which is raised to 1.5 (Fig 6A). However, their TP_{Ala} was still around 1.0, undistinguishable from the autotrophic end member in literature (microalgae and microbes) or our own autotrophic seston (Fig 6B). This suggests these five TP_{Glu} mixotrophic samples were mixotrophs combining only osmotrophy and photoautotrophy. By contrast, five of our autotrophic TP_{Glu} (1.0) samples, pointing to the absence of DON incorporation (Fig 7A), presented a clear distribution around TP_{Ala} 1.5. This suggests a combined contribution of the uptake of DIN and nitrogen derived from the phagocytosis of preys to their nitrogen isotopes (Fig 7B). The last five of our ARP seston samples around TP_{Glu} 1.5 (Fig 8A, Fig S3), also distributed around TP_{Ala} 1.5 (Fig 8B, Fig S3), which suggests a contribution of autotrophy (DIN), osmotrophy (DON) and phagocytosis (nitrogen from preys) to their signatures, that is, a

predominance of mixoplankton in our samples. However, given the lack of specific data on mixoplankton in literature, future studies addressing the impact of DON and N from phagocytosis incorporation on the nitrogen isotopes of different trophic amino acids in mixoplankton are still required to fully validate and extend the application of this amino acid nitrogen isotope approach in situ. For example, it is essential to determine the trophic discrimination factor of Ala and Glu in mixoplankton in order to accurately define TP based on the combination of these two amino acids, instead of only one, which is the most common practice. As shown above, the reason is that Glu only accounts for osmotrophy, and the impact on Ala was not specifically defined for mixoplankton. Nevertheless, our study demonstrates the potential usefulness of combining the nitrogen stable isotopes from different amino acids in seston to detect mixoplankton activity in the field, provided that samples primarily reflect the signature of microalgae.”

And the new figures are:

Figure 6. A) $\delta^{15}N_{Phe}$ (phenylalanine) vs $\delta^{15}N_{Glu}$ (glutamic acid + glutamate) of the literature end members and seston samples collected along the Amazon River plume. B) $\delta^{15}N_{Phe}$ (phenylalanine) vs $\delta^{15}N_{Ala}$ (alanine) of the literature end members and seston samples collected along the Amazon River plume. Symbols represent the organisms: cultured fungi (square),

cultured bacteria (diamond), cultured Archaea (triangle), protozooplankton (crossed circle), cultured microalgae (circle), and field seston (inverted triangles). Colors represent the nitrogen source: dissolved inorganic nitrogen from culture data (green, DIN), dissolved inorganic nitrogen from our seston data (dark green), dissolved organic nitrogen from culture data (purple), DIN + DON in our seston (dark red). The dashed and dotted gray lines represent the trophoclines like in Figure 1A, B.

Figure 7. A) $\delta^{15}\text{N}_{\text{Phe}}$ (phenylalanine) vs $\delta^{15}\text{N}_{\text{Glu}}$ (glutamic acid + glutamate) of the literature end members and seston samples collected along the Amazon River plume. B) $\delta^{15}\text{N}_{\text{Phe}}$ (phenylalanine) vs $\delta^{15}\text{N}_{\text{Ala}}$ (alanine) of the literature end members and seston samples collected along the Amazon River plume. Symbols represent the organisms: cultured fungi (square), cultured bacteria (diamond), protozooplankton (crossed circle), cultured microalgae (circle), and field seston (inverted triangles). Colors represent the nitrogen source: dissolved inorganic nitrogen in culture data (green), dissolved inorganic nitrogen in our seston data (dark green), DIN + N from phagocytosis (preys) in our seston (blue), N from phagocytosis in culture protozooplankton (pink). The dashed and dotted gray lines represent the trophoclines like in Figure 1A, B.

Figure 8. A) $\delta^{15}\text{N}_{\text{Phe}}$ (phenylalanine) vs $\delta^{15}\text{N}_{\text{Glu}}$ (glutamic acid + glutamate) of the literature end members and seston samples collected along the Amazon River plume. B) $\delta^{15}\text{N}_{\text{Phe}}$ (phenylalanine) vs $\delta^{15}\text{N}_{\text{Ala}}$ (alanine) of the literature end members and seston samples collected along the Amazon River plume. Symbols represent the organisms: cultured fungi (square), cultured bacteria (diamond), cultured Archaea (triangle), protozooplankton (crossed circle), cultured microalgae (circle), and field seston (inverted triangles). Colors represent the nitrogen source: dissolved inorganic nitrogen from culture data (green), dissolved inorganic nitrogen from our seston data (dark green), DIN + DON + N from phagocytosis (preys) in our seston (yellow), dissolved organic nitrogen from culture data (purple), and N from phagocytosis in culture protozooplankton (pink). The dashed and dotted gray lines represent the trophoclines like in Figure 1A, B.

Figure S3. **A)** $\delta^{15}\text{N}_{\text{Phe}}$ (phenylalanine) vs $\delta^{15}\text{N}_{\text{Glu}}$ (glutamic acid + glutamate), and **B)** $\delta^{15}\text{N}_{\text{Phe}}$ (phenylalanine) vs $\delta^{15}\text{N}_{\text{Ala}}$ (alanine) of the literature end members (Table S2) and seston samples collected along the Amazon River plume (Table S1). **C)** Comparison of TP_{Glu} and TP_{Ala} calculated according to the equations proposed by Chikaraishi et al. (2009) (Chikaraishi et al. 2009), the vertical and horizontal dashed line represent $\text{TP}_{\text{Glu}} 1.4$ and $\text{TP}_{\text{Ala}} 1.4$, respectively. Symbols represent the organisms: cultured fungi (square), cultured bacteria (diamond), cultured Archaea (triangle), cultured protozooplankton (crossed circle), cultured microalgae (circle), and

field seston (inverted triangle). Colors represent the nitrogen source end member: dissolved inorganic nitrogen (green and dark green, DIN), dissolved organic nitrogen (purple, DON), N from the phagocytosis of preys (pink), DIN+ DON (dark red), DIN + phagocytosis (blue), and DIN + DON + phagocytosis (yellow). It should be noted that two seston sample points overlap at $TP_{Glu} 1.4 + TP_{Ala} 1.1$, and two points overlap at $TP_{Glu} 1.4 + TP_{Ala} 1.2$ for a total of 5 seston samples on DIN +DON.

Comments related to the Methods

L451 The absence of specific information on organism types (genus etc) is very unfortunate. In consequence we cannot compare the plankton composition with membership of the Mixoplankton Data Base :

<https://onlinelibrary.wiley.com/doi/10.1111/jeu.12972?msocid=147dcec09ffd6f8d12f3da399e016edb>

We performed a classification based on pigment markers using ChemTax. This approach is commonly used for defining microalgal functional groups, and has been used for almost 30 years, since Mackey and collaborators proposed this inverse fitting technique in 1996 (Mackey et al. 1996). As mentioned in the introduction to the manuscript, this approach has its caveats, but it allows us to address the entire community of pigmented plankton, from pico- to micro-sized fractions. Using this approach, we were able to detect groups that potentially host known mixoplankton species, supporting the findings of our isotopic approach. Identification at species level was not possible and will not be possible, but we have provided information on upper-level taxa.

The goal of our cruise was not to investigate mixoplankton, we unexpectedly found the imprint of osmotrophy (Glu) and phagotrophy (Ala) in our seston. We would have done a quantitative and consistent sampling for the PlanktoScope and taken samples for microscopy if we were looking for mixoplankton. However, we would have been only able to address the micro size fraction by those means, leaving uncertainties in the pico and nano fraction, which would have required omics in the territorial waters of many American countries. Securing a permission for sampling genetic material in economic exclusive zones for basic research, as needed by field scientist, is extremely difficult if not almost impossible when the authorities tend to allow only local scientists involved in the project to take, handle and analyze those DNA/RNA samples (which is the case in most of the regions under the influence of the plume). Nevertheless, it's the way to go in the future, now that we learned where to look for a mixotrophy hot spot in the Amazon River plume.

L589 This is ambiguous, whether the signal actually separates phototrophy, osmotrophy and phagotrophy, let alone the trophic signal. Is osmotrophy assuming leak-recovery detected? Is osmotrophy supporting a technically different trophic level?!

This comment was in relation to the methods explaining the apparent isotopic fractionation of Glu and Ala relative to Phe (epsilons), and how they could be used for detecting the imprint of

osmotrophy or phagotrophy. We have decided to use a simpler but easier to understand approach using the actual $\delta^{15}\text{N}$ of the amino acids in relation to the different trophoclines both in our samples and the literature end members. And we have also shown the literature end members supporting the approach in the introduction (L87-119), Figure 1 and Table S2.

Hence, the new text in the methods read as follows:

L661-671: “We conducted a comprehensive review of the existing literature to define the different nitrogen source end members resulting from the varied feeding strategies in unicellular organisms which reflect in TP (Table S2). The literature data included: i) autotrophic microalgae growing on inorganic nitrogen (DIN end member); ii) fungi, *Archaea* and bacteria growing on ammonium (DIN end member); iii) fungi, *Archaea* and bacteria growing on casamino acids (DON end member); and iv) eukaryotic strict unicellular phagotrophs (protozooplankton) growing on autotrophic preys (nitrogen from phagocytosis of preys end member). The approach facilitates the analysis of the degree of isotopic enrichment of the two trophic amino acids (Glu and Ala) with trophic transfers due to the metabolism of heterotrophs (osmotrophs and phagotrophs), providing insight into the potentially different fractionation patterns associated with mixoplankton that combine photoautotrophy with osmotrophy and phagotrophy (Fig 1, Table S2).“

L591 Mixoplanktonic activity is quite different to phototrophy + phagotrophy, especially in the context of N-physiology. This is due to the fate of recycled ammonium. I remain concerned on this issue.

Ultimately, the isotopic signature is driven by metabolic routes rather than physiology. The effect of recycled ammonium is reflected in the source amino acid signature, representing the baseline, and is accounted for in CSIA-AA. We would like to note again that the literature dataset described in this section of the Methods represents the relative isotopic fractionation of the trophic amino acids (Glu and Ala) relative to the source amino acid, which directly reflects the baseline (Phe). This accounts for the source of nitrogen (N) to primary producers, regardless of its origin and degree of recycling, and the heterotrophic routes enriching the trophic amino acids relative to that of primary producers.

Comment related to Figures

L920 (Fig.1) suggest this needs to read as 'photoautotrophy' and 'mixotrophy'. The isotopes are aimed to identify processes, not organism types, so nouns are inappropriate.

The terminology used in this figure is consistent with that defined in L146-159, and the Glossary. It represents the TP calculated with the amino acid nitrogen stable isotopes. Hence, no change would be required.

L922 (Fig.1) Contrary to the rebuttal, there still seems to be confusion here; all the photoautorotrophs will be (photo-osmo) mixotrophic.

The terminology used in this figure is consistent with that defined in L146-159, and the Glossary. It represents the TP calculated with the amino acid nitrogen stable isotopes. Besides, the autotrophic sample in station 17.17 mentioned in this line had a TP_{Glu} very close to 1.0 pointing to a very low contribution of osmotrophy in this community.

L938 (Fig2) This needs to say what the PCA is driven by, what the two dimensions are.

We have modified this sentence in the caption of Fig 2 (now Fig 3) for clarity as follows:

L1023-1026: “The separation of the clusters is not random, as confirmed by the principal component analysis (PCA) on the top right panel. The data points in the PCA plot are colored according to the clusters in the dendrogram, and it is possible to see clear clusters distributing along the first two principal components (Dim 1 and Dim2).”

L951 (Fig.3) explain the TP1 .. TP2 lines ('trophic position').

It is unclear what is the issue raised by the Reviewer. The caption of the figure (L954-955) explains that the dashed lines represent trophoclines, which are lines that account for the same trophic position across the combination of $\delta^{15}N_{Phe}$ and $\delta^{15}N_{Glu}$ values at a given TP, as explained in the Introduction (L197).

However, as we have also included a panel with $\delta^{15}N_{Phe}$ and $\delta^{15}N_{Ala}$, the new caption was modified:

L1039-1048: “Figure 4. A) $\delta^{15}N_{Phe}$ (phenylalanine) vs $\delta^{15}N_{Glu}$ (glutamic acid + glutamate) of the seston samples collected along the Amazon River plume. B) $\delta^{15}N_{Phe}$ (phenylalanine) vs $\delta^{15}N_{Ala}$ (alanine) of the seston samples collected along the Amazon River plume. All values expressed in delta notation ($\delta^{15}N$ ‰, relative to atmospheric N_2). In 2018, total particles (combined square and triangle) were collected, while in 2021, seston was separated into two size fractions (triangles for >3 μm , squares for 0.2–3 μm). The gray dashed and dotted lines represent the trophoclines like in Figure 1 A, B. Colors represent the different habitats defined by Pham et al (2024) ordered by apparent age: Riverine Input (RI, gray), Young Plume Core (YPC, red), Outer Plume Margin (OPM, purple), Western Plume Margin (WPM, yellow), modified Oceanic Water (MOW, cyan) and Oceanic Water (OSW, blue).”

The explanation of the trophoclines in Figure 1 reads:

L995-998: “Trophoclines (dashed and dotted gray lines) with a slope of 1.0 and y-intercepts of 3.4‰, 7.2‰, and 11.0‰ for $\delta^{15}N_{Glu}$ and of 3.2‰, 6.0‰, and 8.9‰ for $\delta^{15}N_{Ala}$, respectively, represent different trophic positions (TPs = 1.0, 1.5, 2.0) according to Chikaraishi et al. (2009)”

L963 (Fig.4) If this plot is meant to inspire confidence in the approach, I am afraid it rather fails to do so. There are, to my mind, too much scatter here. You could use the signatures as confirmation of what other lines of research may suggest, but alone, I do not think so. Just compare the protozooplankton and the microalgae. Indeed, if you were given any of these data points for the protist and cyanobacteria, could you correctly identify its trophic location? I do not think so.

This figure was removed, and the literature end members were included in Fig 1. The equivalent to this plot would be now Fig S3 (shown above), which contains all the seston along the ARP and literature end members as well as the relation between TP_{Glu} and TP_{Ala}.

In summary, I am now much clearer in my own mind as to how this work contributes to plankton science. I think the value of the work is useful rather than important. It flags more what needs to be done, and it also flags the problems in applying this amino acid isotope technique. The work itself is critically hampered by the lack of either supporting laboratory data (which can be done) and/or the lack of rate measurements and explicit taxonomic information from the field (which cannot be rectified).

We do not fully agree with the assessment that our work is merely "useful." We believe it makes an important contribution to plankton science. Our method could be further developed to detect the dominance of mixoplankton and would complement existing methods for studying the ecological relevance of osmo- and phagotrophy in the field. The method, a proof of concept, is based on the metabolic basis of fractionation in trophic amino acids relative to source amino acids, which is consistent across other trophic levels and organisms, and was already demonstrated by empirical evidence in osmotrophs and phagotrophs separately. Our ability to account for the baseline and the trophic effect in a single sample is a big step forward in identifying the biogeography and environmental regulation of mixoplankton growth. However, we agree that more laboratory experiments on the diverse group of mixoplankton will help to identify the factors fostering dominance of mixotrophy as identified by CSIA-AA in the field.

References cited in this Rebuttal Letter

- Chikaraishi, Y., N. O. Ogawa, Y. Kashiyama, and others. 2009. Determination of aquatic food-web structure based on compound-specific nitrogen isotopic composition of amino acids. *Limnol. Oceanogr.* **7**: 740–750. doi:10.4319/lom.2009.7.740
- Décima, M., M. R. Landry, C. J. Bradley, and M. L. Fogel. 2017. Alanine $\delta^{15}\text{N}$ trophic fractionation in heterotrophic protists. *Limnol. Oceanogr.* **62**: 2308–2322. doi:<https://doi.org/10.1002/lno.10567>
- Fry, B. 2006. *Stable Isotope Ecology*, Springer New York, NY.
- Gutierrez-Rodriguez, A., M. Decima, B. N. Popp, and M. R. Landry. 2014. Isotopic invisibility of protozoan trophic steps in marine food webs. *Limnol. Oceanogr.* **59**: 1590–1598. doi:10.4319/lo.2014.59.5.1590
- Mackey, M., D. Mackey, H. Higgins, and S. Wright. 1996. CHEMTAX - a program for estimating class abundances from chemical markers: application to HPLC measurements of phytoplankton. *Mar. Ecol. Prog. Ser.* **144**: 265–283.
- McCarthy, M. D., R. Benner, C. Lee, and M. L. Fogel. 2007. Amino acid nitrogen isotopic fractionation patterns as indicators of heterotrophy in plankton, particulate, and dissolved organic matter. *Geochim. Cosmochim. Acta* **71**: 4727–4744. doi:10.1016/j.gca.2007.06.061

- McClelland, J. W., C. M. Holl, and J. P. Montoya. 2003. Relating low $\delta^{15}\text{N}$ values of zooplankton to N_2 -fixation in the tropical North Atlantic: insights provided by stable isotope ratios of amino acids. *Deep Sea Res. Part I Oceanogr. Res. Pap.* **50**: 849–861. doi:10.1016/s0967-0637(03)00073-6
- O'Reilly, C. M., R. E. Hecky, A. S. Cohen, and P.-D. Plisnier. 2002. Interpreting stable isotopes in food webs: Recognizing the role of time averaging at different trophic levels. *Limnol. Oceanogr.* **47**: 306–309. doi:https://doi.org/10.4319/lo.2002.47.1.0306
- Sokal, R. R., and F. J. Rohlf. 1995. *Biometry: The Principles and Practice of Statistics in Biological Research*, W. H. Freeman and Company.
- Wilken, S., J. M. Schuurmans, and H. C. P. Matthijs. 2014. Do mixotrophs grow as photoheterotrophs? Photophysiological acclimation of the chrysophyte *Ochromonas danica* after feeding. *New Phytol.* **204**: 882–889. doi:https://doi.org/10.1111/nph.12975
- Yamaguchi, Y. T., Y. Chikaraishi, Y. Takano, N. O. Ogawa, H. Imachi, Y. Yokoyama, and N. Ohkouchi. 2017. Fractionation of nitrogen isotopes during amino acid metabolism in heterotrophic and chemolithoautotrophic microbes across Eukarya, Bacteria, and Archaea: Effects of nitrogen sources and metabolic pathways. *Org. Geochem.* **111**: 101–112. doi:https://doi.org/10.1016/j.orggeochem.2017.04.004